# Minimax Estimation of Conditional Moment Models

**Nishanth Dikkala**
MIT
nishanthd@csail.mit.edu

**Greg Lewis**
Microsoft Research
glewis@microsoft.com

**Lester Mackey**
Microsoft Research
lmackey@microsoft.com

**Vasilis Syrgkanis**
Microsoft Research
vasy@microsoft.com

## Abstract

We develop an approach for estimating models described via conditional moment restrictions, with a prototypical application being non-parametric instrumental variable regression. We introduce a min-max criterion function under which the estimation problem can be thought of as solving a zero-sum game between a modeler who is optimizing over the hypothesis space of the target model and an adversary who identifies violating moments over a test function space. We analyze the statistical estimation rate of the resulting estimator for arbitrary hypothesis spaces, with respect to an appropriate analogue of the mean squared error metric, for ill-posed inverse problems. We show that when the minimax criterion is regularized with a second moment penalty on the test function and the test function space is sufficiently rich, the estimation rate scales with the critical radius of the hypothesis and test function spaces, a quantity which typically gives tight fast rates. Our main result follows from a novel localized Rademacher analysis of statistical learning problems defined via minimax objectives. We provide applications of our main results for several hypothesis spaces used in practice such as: reproducing kernel Hilbert spaces, high dimensional sparse linear functions, spaces defined via shape constraints, ensemble estimators such as random forests, and neural networks. For each of these applications we provide computationally efficient optimization methods for solving the corresponding minimax problem (e.g., stochastic first-order heuristics for neural networks). In several applications, we show how our modified mean squared error rate, combined with conditions that bound the ill-posedness of the inverse problem, lead to mean squared error rates. We conclude with an extensive experimental analysis of the proposed methods.

## 1 Introduction

Understanding how policy choices affect social systems requires an understanding of the underlying causal relationships between them. To measure these causal relationships, social scientists look to either field experiments or quasi-experimental variation in observational data. Most observational studies rely on assumptions that can be formalized in moment conditions. This is the basis of the estimation approach known as generalized method of moments (GMM) [Hansen, 1982].

While GMM is an incredibly flexible estimation approach, it suffers from some drawbacks. The underlying independence (randomization) assumptions often imply an infinite number of moment

---

A very preliminary version of this work appeared as *Adversarial Generalized Method of Moments* (see https://arxiv.org/abs/1803.07164)

Associated code can be found in https://github.com/microsoft/AdversarialGMM

conditions. Imposing all of them is infeasible with finite data, but it is hard to know which ones to select. For some special cases, asymptotic theory provides some guidance, but it is not clear that this guidance translates well when the data is finite and/or the models are non-parametric. Given the increasing availability of data and new machine learning approaches, researchers and data scientists may want to apply adaptive non-parametric learners such as reproducing kernel Hilbert spaces, high-dimensional regularized linear models, neural networks, and random forests to these GMM estimation problems, but this requires a way of finding *solutions to the moment conditions within complex hypothesis classes imposed by the learner* and selecting moment conditions *that are adapted to the hypothesis class of the learner*.

Many theoretical developments in machine learning and statistics begin by formulating the target estimand as the minimizer of a population loss function (typically strongly convex with respect to the output of the hypothesis) over a hypothesis space and the estimation procedure as an $M$-*estimator* (Chapter 5, [Van der Vaart, 2000]) that minimizes an empirical estimate of the population loss. Framing learning as $M$-estimation with a strongly convex loss leads to many desirable properties: i) tight generalization bounds and mean squared error rates based on localized notions of statistical complexity can be invoked to provide tight and fast finite sample rates with minimal assumptions [Bartlett et al., 2005, Wainwright, 2019], ii) regularization can be invoked to make the estimation adaptive to the complexity of the true hypothesis space, without knowledge of that complexity [Lecué and Mendelson, 2018, 2017, Negahban et al., 2012], iii) the computational problem can be typically efficiently solved via first order methods that can scale massively [Agarwal et al., 2014, Rahimi and Recht, 2008, Le, 2013, Sra et al., 2012, Bottou et al., 2007]. This formulation is seemingly at odds with the method of moments approach to estimation, as often moment conditions do not correspond to the gradient of some loss function, and this problem is exacerbated in the case of non-parametric endogenous regression problems (i.e., when the instruments in the observational study does not coincide with the treatments). This leads to the main question of this work: *Can we develop an analogue of the modern statistical learning theory of $M$-estimators for non-parametric problems defined via moment restrictions?*

Our starting point is a set of conditional moment restrictions

$$\mathbb{E}[y - h(x) \mid z] = 0 \tag{1}$$

where $y$ is an outcome of interest, $x$ is a vector of treatments, and $z$ is a vector of instruments.

To obtain a criterion function, we first move to an unconditional moment formulation, where the moment restrictions are products of the moment conditions and test functions in the instruments. We then take as our criterion function the maximum moment deviation over the set of test functions, where the set of test functions is potentially infinite:

$$h_0 = \arg\inf_{h \in \mathcal{H}} \sup_{f \in \mathcal{F}} \mathbb{E}[(y - h(x))f(z)] =: \arg\inf_{h \in \mathcal{H}} \sup_{f \in \mathcal{F}} \Psi(h, f) \tag{2}$$

This formulation of the conditional moment problem turns it into an adversarial learning problem, reminiscent of adversarial approaches in machine learning, such as Generative Adversarial Networks (GANs). Similar to Wasserstein [Arjovsky et al., 2017] and MMD [Li et al., 2017] GANs, in our adversarial problem the learner is trying to find a model $h$ that satisfies all moment constraints, and the adversary is trying to identify moments that are violated for the chosen $h$. Unlike with GANs, we do not learn a generative model or impose a likelihood but rather only impose moment conditions that our model $h$ needs to satisfy. This can be thought of as a zero-sum game between the learner and the adversary where the adversary's payoff for a strategy pair $(h, f)$ is given by $\Psi(h, f)$ (2).

We show that, as long as the set of test functions $\mathcal{F}$ contains all functions of the form $f(z) = \mathbb{E}[h(x) - h'(x) \mid z]$ for $h, h' \in \mathcal{H}$, an estimator based on a regularized empirical analogue of the minimax criterion achieves a projected MSE rate that scales with the critical radius of the function classes $\mathcal{F}, \mathcal{H}$, and their tensor product class (i.e., functions of the form $f(z) \cdot h(x)$, with $f \in \mathcal{F}$ and $h \in \mathcal{H}$). Since the critical radius captures information theoretically optimal rates for many function classes of interest, our main theorem can be used to derive tight estimation rates for many hypothesis spaces. Moreover, if the regularization terms relate to the squared norms of $h$ and $f$ in their corresponding spaces, then the estimation error scales with the norm of the true hypothesis, even without knowledge of this norm. Finally, one important aspect of the regularization that we consider for our main theorem is that it contains a second moment penalty on the test functions with a weight that does not vanish to zero asymptotically.

We offer applications of our main theorems for several hypothesis spaces of practical interest including reproducing kernel Hilbert spaces (RKHS), sparse linear functions, functions defined via shape restrictions, neural networks, and random forests. For many of these estimators, we offer optimization algorithms with performance guarantees. As we illustrate in extensive simulation studies, different estimators are best in different regimes.

**Related work**   The non-parametric IV problem has a long history in econometrics [Newey and Powell, 2003, Blundell et al., 2007, Chen and Pouzo, 2012, Chen and Christensen, 2018, Hall et al., 2005, Horowitz, 2007, 2011, Darolles et al., 2011, Chen and Pouzo, 2009]. Arguably the closest to our work is that of Chen and Pouzo [2012], who consider estimation of non-parametric function classes and estimation via the method of sieves and a penalized minimum distance estimator of the form: $\min_{h \in \mathcal{H}} \mathbb{E}[\mathbb{E}[y - h(x) \mid z]^2] + \lambda R(h)$, where $R(h)$ is a regularizer. As we show in Appendix A, our estimator can be interpreted asymptotically as a minimum distance estimator, albeit our estimation method applies to arbitrary function classes and non just linear sieves. There is also a growing body of work in the machine learning literature on the non-parametric instrumental variable regression problem [Hartford et al., 2017, Bennett et al., 2019, Singh et al., 2019, Muandet et al., 2019, 2020]. Our work has several features that draw connections to each of these works, e.g. Bennett et al. [2019], Muandet et al. [2019, 2020] also use a minimax criterion and Bennett et al. [2019], Muandet et al. [2019] also impose some form of variance penalty on the test function. We discuss subtle differences in Appendix A. Moreover, Singh et al. [2019], Muandet et al. [2019] also study RKHS hypothesis spaces and Hartford et al. [2017], Bennett et al. [2019] also study neural net hypothesis spaces. None of these prior works provide finite sample estimation error rates for arbitrary hypothesis spaces and typically only show consistency for the particular hypothesis space analyzed (with the exception of Singh et al. [2019], who provide finite sample rates for RKHS spaces, under further conditions on the smoothness of the true hypothesis). In Appendix A we offer a more detailed exposition on the related work and how it relates to our main results.

## 2   Preliminary Definitions

We consider the problem of estimating a flexible econometric model that satisfies a set of conditional moment restrictions presented in (1) (see also Appendix B), where $z \in \mathcal{Z} \subseteq \mathbb{R}^d$, $X \in \mathcal{X} \subseteq \mathbb{R}^p$, $y \in \mathbb{R}$, $h \in \mathcal{H} \subseteq (\mathcal{X} \to \mathbb{R})$ for $\mathcal{H}$ a hypothesis space. For simplicity of notation we will also denote with $\psi(y; h(x)) = y - h(x)$. The truth is some model $h_0$ that satisfies all the moment restrictions.

We assume we have access to a set of $n$ i.i.d. sample points $\{v_i := (y_i, x_i, z_i)\}_{i=1}^n$ drawn from some unknown distribution $\mathcal{D}$ that satisfies the moment condition in Equation (1). We will analyze estimators that optimize an empirical analogue of the minimax objective presented in the introduction, potentially adding norm-based penalties $\Phi : \mathcal{F} \to \mathbb{R}_+$, $R : \mathcal{H} \to \mathbb{R}_+$:

$$\hat{h} := \arg\min_{h \in \mathcal{H}} \sup_{f \in \mathcal{F}} \Psi_n(h, f) - \lambda \, \Phi(f) + \mu \, R(h)$$

where $\Psi_n(h, f) := \frac{1}{n} \sum_{i=1}^n \psi(y_i; h(x_i)) \, f(z_i)$.

We assume that $\mathcal{H}$ and $\mathcal{F}$ are classes of bounded functions on their corresponding domains and, without loss of generality, their image is a subset of $[-1, 1]$. Similarly, we will also assume that $y \in [-1, 1]$. The results of this section hold for a general bounded range $[-b, b]$ via standard re-scaling arguments with an extra multiplicative factor of $b$. Moreover, we will assume that $\mathcal{F}$ is a symmetric class, i.e. if $f \in \mathcal{F}$ then $-f \in \mathcal{F}$. Moreover, we will assume that $\mathcal{H}$ and $\mathcal{F}$ are equipped with norms $\| \cdot \|_{\mathcal{H}}, \| \cdot \|_{\mathcal{F}}$. For any function class $\mathcal{G}$ we let $\mathcal{G}_B = \{g \in \mathcal{G} : \|g\| \leq B\}$, be the $B$ bounded norm subset of the class.

Our estimation target is good generalization performance with respect to the projected root-mean-squared-error (RMSE), defined as the RMSE projected onto the space of instruments:

$$\|T(\hat{h} - h_0)\|_2 := \sqrt{\mathbb{E}\left[ \left( \mathbb{E}[\hat{h}(x) - h_0(x) \mid z] \right)^2 \right]} \qquad \text{(Projected RMSE)}$$

where $T : \mathcal{H} \to \mathcal{F}$ is the linear operator defined as $Th := \mathbb{E}[h(X) \mid Z = \cdot]$. This performance metric is appropriate given the ill-posedness problem well known in this setting; imposing further conditions on the strength of the correlation between the treatments and instruments (instrument

strength) allows one to, translate bounds on the projected RMSE to bounds on the RMSE (see e.g. Chen and Pouzo [2012] and other references in the applications below).[1]

We start by defining some preliminary notions from empirical process theory that are required to state our main results. Let $\mathcal{G}$ a class of uniformly bounded functions $g : \mathcal{V} \to [-1, 1]$ from some domain $\mathcal{V}$ to $[-1, 1]$. The *localized Rademacher complexity* of the function class is defined as: $\mathcal{R}_n(\delta; \mathcal{G}) = \mathbb{E}_{\{\epsilon_i\}_{i=1}^n, \{v_i\}_{i=1}^n} \left[ \sup_{\substack{g \in \mathcal{G} \\ \|g\|_2 \leq \delta}} \left| \frac{1}{n} \sum_{i=1}^n \epsilon_i g(v_i) \right| \right]$, where $\{v_i\}_{i=1}^n$ are i.i.d. samples from some distribution $D$ on $\mathcal{V}$ and $\{\epsilon_i\}_{i=1}^n$ are i.i.d. Rademacher random variables taking values equiprobably in $\{-1, 1\}$. We will also denote with $\mathcal{R}_n(\mathcal{G})$, the un-restricted Rademacher complexity, i.e. $\delta = \infty$.

We denote with $\|\cdot\|_2$ the $\ell_2$-*norm* with respect to the distribution $D$, i.e. $\|g\|_2 = \sqrt{\mathbb{E}_{v \sim D}[g(v)^2]}$, and analogously we define the *empirical $\ell_2$-norm* as $\|g\|_{2,n} = \sqrt{\frac{1}{n} \sum_i g(v_i)^2}$. In our context, where $v = (y, x, z)$, when functions take as input subsets of the vector $v$, then we will overload notation and let $\|\cdot\|_2$ and $\|\cdot\|_{2,n}$ denote the population and sample $\ell_2$ norms with respect to the marginal distribution of the corresponding input, e.g., if $h$ is a function of $x$ alone and $f$ a function of $z$ alone, we write $\|h\|_2 = \sqrt{\mathbb{E}_x[h(x)^2]}$, $\|f\|_2 = \sqrt{\mathbb{E}_z[f(z)^2]}$, and $\|hf\|_2 = \sqrt{\mathbb{E}_{x,z}[h(x)^2 f(z)^2]}$.

A function class $\mathcal{G}$ is said to be *symmetric* if $g \in \mathcal{G} \implies -g \in \mathcal{G}$. Moreover, it is said to be *star-convex* if: $g \in \mathcal{G} \implies r\,g \in \mathcal{G}, \forall r \in [0, 1]$. The *critical radius* $\delta_n$ of the function class $\mathcal{G}$ is any solution to the inequality $\mathcal{R}_n(\delta; \mathcal{G}) \leq \delta^2$.

## 3    Main Theorems

We show that, if the function space $\mathcal{F}_U$ contains projected differences of hypothesis spaces $h \in \mathcal{H}_B$, with some benchmark hypothesis $h_* \in \mathcal{H}_B$, i.e. $T(h - h_*) \in \mathcal{F}_U$, then a regularized minimax estimator can achieve estimation rates that are of the order of the projected root-mean-squared-error of the benchmark hypothesis $h_*$ and the critical radii of (i) the function class $\mathcal{F}_{3U}$ and (ii) a function class $\mathcal{G}$ that consists of functions of the form: $q(x) \cdot Tq(z)$, for $q = h - h_*$. The projected root mean squared error of the benchmark class can be understood as the *approximation error* or *bias* of the hypothesis space $\mathcal{H}_B$, and the critical radius can be understood as the *sampling error* or *variance* of the estimate. If $h_0 \in \mathcal{H}_B$, then the *approximation error* is zero. We present a slightly more general statement, where we also allow for $\mathcal{F}_U$ to not exactly include $T(h - h_*)$, but rather functions that are close to it with respect to the $\ell_2$ norm. For this reason, we will need to define the following slightly more complex hypothesis space, in order to state our main theorem:

$$\hat{\mathcal{G}}_{B,U} := \{(x, z) \to r\,(h(x) - h_*(x))\,f_h^U(z) : h \in \mathcal{H} \text{ s.t. } h - h_* \in \mathcal{H}_B, r \in [0, 1]\} \quad (3)$$

where $f_h^U = \arg\min_{f \in \mathcal{F}_U} \|f - T(h - h_*)\|_2$. If $T(h - h_*) \in \mathcal{F}_U$, then this simplifies to the class of functions of the form: $(h - h_*)(x)\,T(h - h_*)(z)$.

**Theorem 1.** *Let $\mathcal{F}$ be a symmetric and star-convex set of test functions and consider the estimator:*

$$\hat{h} = \arg\min_{h \in \mathcal{H}} \sup_{f \in \mathcal{F}} \Psi_n(h, f) - \lambda \left( \|f\|_{\mathcal{F}}^2 + \frac{U}{\delta^2} \|f\|_{2,n}^2 \right) + \mu\|h\|_{\mathcal{H}}^2 \quad (4)$$

*Let $h_* \in \mathcal{H}$ be any fixed hypothesis (independent of the samples) and $h_0$ be any hypothesis (not necessarily in $\mathcal{H}$) that satisfies the Conditional Moment (1) and suppose that:*

$$\forall h \in \mathcal{H} : \min_{f \in \mathcal{F}_{L^2\|h - h_*\|_{\mathcal{H}}^2}} \|f - T(h - h_*)\|_2 \leq \eta_n \quad (5)$$

*Assume that functions in $\mathcal{H}_B$ and $\mathcal{F}_{3U}$ have uniformly bounded ranges in $[-1, 1]$ and that: $\delta :=$*
*$\delta_n + c_0 \sqrt{\frac{\log(c_1/\zeta)}{n}}$, for universal constants $c_0, c_1$, and $\delta_n$ an upper bound on the critical radii of $\mathcal{F}_{3U}$*
*and $\hat{\mathcal{G}}_{B, L^2 B}$. If $\lambda \geq \delta^2/U$ and $\mu \geq 2\lambda(4L^2 + 27U/B)$, then $\hat{h}$ satisfies w.p. $1 - 3\zeta$:*

$$\|T(\hat{h} - h_*)\|_2, \|T(\hat{h} - h_0)\|_2 \leq O\left(\delta + \eta_n + \|h_*\|_{\mathcal{H}}^2 \frac{\lambda + \mu}{\delta} + \|T(h_* - h_0)\|_2 + \frac{\|T(h_* - h_0)\|_2^2}{\delta}\right)$$

*If further $\lambda, \mu = O(\delta^2)$ and $\delta \geq \|T(h_* - h_0)\|_2$, then:*

$$\|T(\hat{h} - h_*)\|_2, \|T(\hat{h} - h_0)\|_2 \leq O\left(\delta \max\{1, \|h_*\|_{\mathcal{H}}^2\} + \eta_n + \|T(h_* - h_0)\|_2\right)$$

Observe that in (4), the regularization terms push for a norm-constrained solution. If the classes $\mathcal{H}, \mathcal{F}$ already are norm constrained, then the theorem directly applies to the estimator that solely penalizes the $\ell_{2,n}$ norm of $f$, i.e.:[2]

$$\hat{h} := \arg\min_{h \in \mathcal{H}} \sup_{f \in \mathcal{F}} \Psi_n(h, f) - \|f\|_{2,n}^2 \tag{6}$$

However, as we show below, imposing norm regularization as opposed to hard norm constraints leads to adaptivity properties of the estimator.

**Adaptivity of regularized estimator**   Suppose that we know that for $B, U = 1$, we have that functions in $\mathcal{H}_B, \mathcal{F}_U$ have ranges in $[-1, 1]$ as their inputs range in $\mathcal{X}$ and $\mathcal{Z}$ correspondingly. Then our theorem requires that we set: $\lambda \geq \delta^2$ and $\mu \geq 2\lambda(4L^2 + 27)$, where $\delta^2$ depends on the critical radius of the function class $\mathcal{F}_1$ and $\mathcal{G}_1$. Observe that none of these values depend on the norm of the benchmark hypothesis $\|h_*\|_{\mathcal{H}}$, which can be arbitrary and not constrained by our theorem (see also Appendix C.1).

For some function classes $\mathcal{H}$ that admit sparse representations, we can get an improved performance if instead of testing for classes of functions $\mathcal{F}$ that contain $T(h - h_*)$, we test functions whose linear span contains $T(h - h_*)$, i.e. that $T(h - h_*) = \sum_i w_i f_i$, assuming the weights required in this linear span have small $\ell_1$ norm. The reason being that the generalization error of linear spans with bounded $\ell_1$ norm can be prohibitively large to get fast error rates, i.e. the Rademacher complexity of the span of $\mathcal{F}$ can be much larger than $\mathcal{F}$, thereby introducing large sampling variance to our sup-loss objective. To state the improved result, we define for any function space $\mathcal{F}$: $\text{span}_\kappa(\mathcal{F}) := \{\sum_{i=1}^p w_i f_i : f_i \in \mathcal{F}, \|w\|_1 \leq \kappa, p \leq \infty\}$, i.e. the set of functions that consist of linear combinations of a finite set of elements $\mathcal{F}$, with the $\ell_1$ norm of the weights bounded by $R$. To get fast rates in this second result, we will require that the $\ell_2$-normalized $T(h - h_*)$ belongs to the span. We present the theorem in the well-specified setting, but a similar result holds in the case where $h_0 \notin \mathcal{H}_B$, with the extra modification of adding a second moment penalty on $f$.

**Theorem 2.** *Consider a set of test functions $\mathcal{F} := \cup_{i=1}^d \mathcal{F}^i$, that is decomposable as a union of $d$ symmetric test function spaces $\mathcal{F}^i$ and let $\mathcal{F}_U^i = \{f \in \mathcal{F}^i : \|f\|_{\mathcal{F}}^2 \leq U\}$. Consider the estimator:*

$$\hat{h} = \arg\min_{h \in \mathcal{H}} \sup_{f \in \mathcal{F}_U} \Psi_n(h, f) + \lambda \|h\|_{\mathcal{H}} \tag{7}$$

*Let $h_0 \in \mathcal{H}_B$ be any fixed (independent of the samples) hypothesis that satisfies the Conditional Moment (1). Let $\delta_{n,\zeta} := 2 \max_{i=1}^d \mathcal{R}_n(\mathcal{F}_U^i) + c_0 \sqrt{\frac{\log(c_1 d/\zeta)}{n}}$, for some universal constants $c_0, c_1$ and $B_{n,\lambda,\zeta} := \|h_0\|_{\mathcal{H}} + \delta_{n,\zeta}/\lambda$. Suppose that:*

$$\forall h \in \mathcal{H}_{B_{n,\lambda,\zeta}} : \frac{T(h - h_0)}{\|T(h - h_0)\|_2} \in \text{span}_\kappa(\mathcal{F}_U)$$

*Then if $\lambda \geq \delta_{n,\zeta}$, $\hat{h}$ satisfies for some universal constants $c_0, c_1$, that w.p. $1 - \zeta$:*

$$\|T(h_0 - \hat{h})\|_2 \leq \kappa \left(2(B+1)\mathcal{R}_n(\mathcal{H}_1) + \delta_{n,\zeta} + \lambda \left(\|h_0\|_{\mathcal{H}} - \|\hat{h}\|_{\mathcal{H}}\right)\right)$$

In Appendix C we provide further discussion related to our main theorems: i) we provide further discussion on the adaptivity of our estimators, ii) we provide connections between the critical radius and the entropy integral and how to bound the critical radius via covering arguments, iii) we provide generic approaches to solving the optimization problem, iv) we show how to combine our main theorem on the projected MSE with bounds on the ill-posedness of the inverse problem in order to achieve MSE rates, v) we offer a discussion on the optimality of our estimation rate.

# 4  Application: Reproducing Kernel Hilbert Spaces

In this section we describe how Theorem 1 applies to the case where $h_0$ lies in a reproducing kernel Hilbert space (RKHS) $\mathbb{H}_{K_{\mathcal{H}}}$ with kernel $K_{\mathcal{H}} : \mathcal{X} \times \mathcal{X} \rightarrow \mathbb{R}$ and $Th_0$ lies in another RKHS $\mathbb{H}_{K_{\mathcal{F}}}$ with kernel $K_{\mathcal{F}} : \mathcal{Z} \times \mathcal{Z} \rightarrow \mathbb{R}$. We choose $\mathcal{F}$ to be an RKHS norm ball of $K_{\mathcal{F}}$ and outline here the main ideas behind the three components required to apply our general theory. A complete discussion is deferred to Appendix E.

First we characterize the set of test functions that are sufficient to satisfy the requirement that $T(h - h_0) \in \mathcal{F}_U$. In Lemma 7, we show that $Th \in \mathbb{H}_{K_{\mathcal{F}}}$ whenever the conditional density function $z \mapsto p(x \mid z)$ resides in the RKHS $\mathbb{H}_{K_{\mathcal{F}}}$ for each $x$. Moreover, we show that under the stronger conditions (see Lemma 8) that $p(x \mid z) = \rho(x - z)$ and $K_{\mathcal{H}}(x, y) = k(x - y)$, for $k$ positive definite and continuous, then $Th \in \mathbb{H}_K$, i.e. $Th$ falls in the same RKHS as $h$. These two theorems give concrete guidance in terms of primitive assumptions on what RKHS should be used as a test function space to satisfy $T(h - h_0) \in \mathcal{F}_U$.

Second, by recent results in statistical learning theory [Wainwright, 2019, Lem. 13.6], the critical radius of any RKHS-norm constrained subset of an RKHS class with kernel $K$ and norm bound $B$, can be characterized as a function of the *eigen-decay of the empirical kernel matrix* **K** *defined as* $\mathbf{K}_{ij} = K(x_i, x_j)/n$. More concretely, $\delta_n$ is the smallest positive solution to $B\sqrt{\frac{2}{n}}\sqrt{\sum_{j=1}^n \min\{\lambda_j^S, \delta^2\}} \leq \delta^2$, where $\lambda_j^S$ are the empirical eigenvalues and, in the worst-case is of the order of $n^{-1/4}$. In the context of Theorem 1, the function classes $\mathcal{F}$ and $\mathcal{G}_B$ are norm balls of the reproducing kernel Hilbert spaces with kernels $K_{\mathcal{F}}$ and $K_{\times}((x, z), (x', z')) = K_{\mathcal{H}}(x, x') \cdot K_{\mathcal{F}}(z, z')$ respectively. Thus we can bound the critical radius required in the theorem as a function of the eigendecay of the corresponding empirical kernel matrices, which are data-dependent quantities.

Combining these two facts, we can then apply Theorem 1 to get a bound on the estimation error of the minimax or regularized minimax estimator. Moreover, we show that for this set of test functions and hypothesis spaces, *the empirical min-max optimization problem can be solved in closed form*. In particular, the estimator in Equation (4) takes the form:

$$\hat{h} = \sum_{i=1}^n \alpha_{\lambda_*,i} K_{\mathcal{H}}(x_i, \cdot) \qquad \alpha_\lambda := (K_{\mathcal{H},n} M K_{\mathcal{H},n} + 4\lambda\mu K_{\mathcal{H},n})^\dagger K_{\mathcal{H},n} M y$$

where $K_{\mathcal{H},n} = (K_{\mathcal{H}}(x_i, x_j))_{i,j=1}^n$ and $K_{\mathcal{F},n} = (K_{\mathcal{F}}(z_i, z_j))_{i,j=1}^n$ are empirical kernel matrices, and $M = K_{\mathcal{F},n}^{1/2}(\frac{U}{n\delta^2}K_{\mathcal{F},n} + I)^{-1}K_{\mathcal{F},n}^{1/2}$ (for $A^\dagger$ the Moore-Penrose pseudoinverse of $A$). Moreover, in Appendix E.3, we discuss how ideas from low rank kernel matrix approximation (such as the Nystrom method) can avoid the $O(n^3)$ running time for matrix inverse computation in the latter closed form. Finally, we show (see Appendix E.4) that if we make further assumptions on the rate at which the operator $T$ distorts the orthonormality of the eigenfunctions of the kernel $K_{\mathcal{H}}$, then our analysis also yields mean-squared-error rates.

# 5  Application: High-Dimensional Sparse Linear Function Spaces

In this section we deal with high-dimensional linear function classes, i.e. the case when $\mathcal{X}, \mathcal{Z} \subseteq \mathbb{R}^p$ for $p \gg n$ and $h_0(x) = \langle \theta_0, x \rangle$ (see Appendix F for more details). We will address the case when the function $\theta_0$ is assumed to be sparse, i.e. $\|\theta_0\|_0 := \{j \in [p] : |\theta_j| > 0\} \leq s$. We will be denoting with $S$ the subset of coordinates of $\theta_0$ that are non-zero and with $S^c$ its complement. For simplicity of exposition we will also assume that $\mathbb{E}[x_i \mid z] = \langle \beta, z \rangle$, though most of the results of this section also extend to the case where $\mathbb{E}[x_i \mid z] \in \mathcal{F}_i$ for some $\mathcal{F}_i$ with small Rademacher complexity. Variants of this setting have been analyzed in the prior works of [Gautier et al., 2011, Fan and Liao, 2014]. We focus on the case where the covariance matrix $V := \mathbb{E}[\mathbb{E}[x \mid z]\mathbb{E}[x \mid z]^\top]$, has a restricted minimum eigenvalue of $\gamma$ and apply Theorem 2. We note that without the minimum eigenvalue condition, our Theorem 1 provides slow rates of the order of $n^{-1/4}$, for computationally efficient estimators that replace the hard sparsity constraint with an $\ell_1$-norm constraint.

**Corollary 3.** *Suppose that* $h_0(x) = \langle \theta_0, x \rangle$ *with* $\|\theta_0\|_0 \leq s$ *and* $\|\theta_0\|_1 \leq B$ *and* $\|\theta_0\|_\infty \leq 1$. *Moreover, suppose that* $\mathbb{E}[x_i \mid z] = \langle \beta_0^i, z \rangle$, *with* $\beta_0^i \in \mathbb{R}^p$ *and* $\|\beta_0^i\|_1 \leq U$ *and that the co-variance matrix* $V$ *satisfies the following restricted eigenvalue condition:*

$$\forall \nu \in \mathbb{R}^p \ s.t. \ \|\nu_{S^c}\|_1 \leq \|\nu_S\|_1 + 2\delta_{n,\zeta} : \nu^\top V \nu \geq \gamma\|\nu\|_2^2$$

*Then let $\mathcal{H} = \{x \to \langle \theta, x \rangle : \theta \in \mathbb{R}^p\}$, $\|\langle \theta, \cdot \rangle\|_{\mathcal{H}} = \|\theta\|_1$, $\mathcal{F}_U = \{z \to \langle \beta, z \rangle : \beta \in \mathbb{R}^p, \|\beta\|_1 \leq U\}$ and $\|\langle \beta, \cdot \rangle\|_{\mathcal{F}} = \|\beta\|_1$. Then the estimator presented in Equation (7) with $\lambda \leq \frac{\gamma}{8s}$, satisfies that w.p. $1 - \zeta$:*

$$\|T(\hat{h} - h_0)\|_2 \leq O\left(\max\left\{1, \frac{1}{\lambda}\frac{\gamma}{s}\right\} \sqrt{\frac{s}{\gamma}} \left((B+1)\sqrt{\frac{\log(p)}{n}} + U\sqrt{\frac{\log(p)}{n}} + \sqrt{\frac{\log(p/\zeta)}{n}}\right)\right)$$

*If instead we assume that $\|\beta_0^i\|_2 \leq U$ and $\sup_{z \in \mathcal{Z}} \|z\|_2 \leq R$ then by setting $\mathcal{F}_U = \{z \to \langle \beta, z \rangle : \|\beta\|_2 \leq U\}$ and $\|\langle \beta, \cdot \rangle\|_{\mathcal{F}} = \|\beta\|_2$, then the later rate holds with $U\sqrt{\frac{\log(p)}{n}}$ replaced by $\frac{UR}{\sqrt{n}}$.*

Notably, observe that in the case of $\|\beta_0^i\|_2 \leq U$, we note that if one wants to learn the true $\beta$ with respect to the $\ell_2$ norm or the functions $\mathbb{E}[x_i \mid z]$ with respect to the RMSE, then the best rate one can achieve (by standard results for statistical learning with the square loss), even when one assumes that $\sup_{z \in \mathcal{Z}} \|z\|_2 \leq R$ and that $\mathbb{E}[zz^\top]$ has minimum eigenvalue of at least $\gamma$, is: $\min\left\{\sqrt{\frac{p}{n}}, \left(\frac{UR}{n}\right)^{1/4}\right\}$. For large $p \gg n$ the first rate is vacuous. Thus we see that even though we cannot accurately learn the conditional expectation functions at a $1/\sqrt{n}$ rate, we can still estimate $h_0$ at a $1/\sqrt{n}$ rate, assuming that $h_0$ is sparse. Therefore, the minimax approach offers some form of robustness to nuisance parameters, reminiscent of Neyman orthogonal methods (see e.g. Chernozhukov et al. [2018]).

In Appendix F.3 we also provide first-order iterative and computationally efficient algorithms with provable guarantees for solving the optimization problem. Moreover, we show that recent advances in online learning theory can be utilized to get fast iteration complexity, i.e. achieve error $\epsilon$ after $O(1/\epsilon)$ iterations (instead of the typical rate of $O(1/\epsilon^2)$ for non-smooth functions). Finally, in Appendix F.4, we also show if we assume that the minimum eigenvalue of $V$ is at least $\gamma$ and the maximum eigenvalue of $\Sigma = \mathbb{E}[xx^\dagger]$ is at most $\sigma$, then the same rate as the one presented in Corollary 3 holds for the MSE, multiplied by the constant $\sqrt{\sigma/\gamma}$.

# 6 Neural Networks

In this section we describe how one can apply the theoretical findings from the previous sections to understand how to train neural networks that solve the conditional moment problem. We will consider the case when our true function $h_0$ can be represented (or well-approximated) by a deep neural network function of $x$, for some given domain specific network architecture, and we will represent it as $h_0(x) = h_{\theta_0}(x)$, where $\theta_0$ are the weights of the neural net (see Appendix H for more details). Moreover, we will assume that the linear operator $T$, satisfies that for any set of weights $\theta$, we have that $Th_\theta$ belongs to a set of functions that can be represented (or well-approximated) as another deep neural network architecture, and we will denote these functions as $f_w(z)$, where $w$ are the weights of the neural net.

**Adversarial GMM Networks (AGMM)**  Thus we can apply our general approach presented in Theorem 1 (simplified for the case when $U = B = 1$, $\lambda = \delta^2$, $\mu = 2\delta^2(4L^2 + 27)$, where $L$ is a bound on the lipschitzness of the operator $T$ with respect to the two function space norms and $\delta$ is a bound on the critical radius of the function spaces $\mathcal{F}_3$ and $\hat{\mathcal{G}}_{1,L^2}$):

$$\hat{\theta} = \arg\min_\theta \sup_w \mathbb{E}_n[\psi(y_i; h_\theta(x_i))f_w(z)] - \delta^2\|f_w\|_{\mathcal{F}}^2 - \frac{1}{n}\sum_i f_w(z_i)^2 + c\,\delta^2\|h_\theta\|_{\mathcal{H}}^2$$

for some constant $c > 1$ that depends on the lipschitzness of the operator $T$. The AGMM criterion for training neural networks is closely related to the work of Bennett et al. [2019]. However, the regularization presented in Bennett et al. [2019] is not a simple second moment penalization, so as to emulate the optimally weighted GMM criterion Hansen [1982]. Here we show that such a more complex penalty is not required if one simply wants fast projected MSE rates (in Appendix H we provide further discussion). Moreover, in Appendix H.1, we show how to derive intuition from our RKHS analysis to develop an architecture for the test function network that under conditions is guaranteed to contain the set of functions of the form $Th$. This leads to an MMD-GAN style adversarial GMM approach, where we consider test functions of the form: $f(z) = \frac{1}{s}\sum_{i=1}^s \beta_i K(c_i, g_w(z))$, where $c_i$ are parameters that could also be trained via gradient descent. The latter essentially corresponds to adding what is known as an RBF layer at the end of the adversary neural net (denoted as KLayer-Trained in experiments). Finally, in Appendix H.2, we provide heuristic methods for solving the non-convex/non-concave zero-sum game, using first order dynamics.

## 7 Random Forests via a Reduction Approach

We will show that we can reduce the problem presented in (6) to a regression oracle over the function space $\mathcal{F}$ and a classification oracle over the function space $\mathcal{H}$ (see Appendix I for more details). We will assume that we have a regression oracle that solves the square loss problem over $\mathcal{F}$: for any set of labels and features $z_{1:n}, u_{1:n}$ it returns

$$\text{Oracle}_{\mathcal{F}}(z_{1:n}, u_{1:n}) = \arg\min_{f \in \mathcal{F}} \frac{1}{n} \sum_{i=1}^{n} (u_i - f(z_i))^2$$

Moreover, we assume that we have a classification oracle that solves the weighted binary classification problem over $\mathcal{H}$ w.r.t. the accuracy criterion: for any set of sample weights $w_{1:n}$, binary labels $v_{1:n}$ in $\{0, 1\}$ and features $x_{1:n}$:

$$\text{Oracle}_{\mathcal{H}}(x_{1:n}, v_{1:n}, w_{1:n}) = \arg\max_{h \in \mathcal{H}} \frac{1}{n} \sum_{i=1}^{n} w_i \Pr_{z_i \sim \text{Bernoulli}\left(\frac{1+h(x_i)}{2}\right)} [v_i = z_i]$$

**Theorem 4.** *Consider the algorithm where for $t = 1, \ldots, T$: let*

$$u_i^t = \frac{1}{2}\left(y_i - \frac{1}{t-1}\sum_{\tau=1}^{t-1} h_\tau(x_i)\right), \qquad f_t = Oracle_{\mathcal{F}}\left(z_{1:n}, u_{1:n}^t\right)$$

$$v_i^t = 1\{f_t(z_i) > 0\}, w_i^t = |f_t(z_i)| \qquad h_t = Oracle_{\mathcal{H}}\left(x_{1:n}, v_{1:n}^t, w_{1:n}^t\right)$$

*Suppose that the set $A = \{(f(z_1), \ldots, f(z_n)) : f \in \mathcal{F}\}$ is a convex set. Then the ensemble: $\bar{h} = \frac{1}{T}\sum_{t=1}^{T} h_t$, is a $\frac{8(\log(T)+1)}{T}$-approximate solution to the minimax problem in Equation (6).*

In practice, we will consider a random forest regression method as the oracle over $\mathcal{F}$ and a binary decision tree classification method as the oracle for $\mathcal{H}$ (which we will refer to as RFIV). Prior work on random forests for causal inference has focused primarily on learning forests that capture the heterogeneity of the treatment effect of a treatment, but did not account for non-linear relationships between the treatment and the outcome variable. The method proposed in this section makes this possible. Observe that the convexity of the set $A$ is violated by the random forest function class with a bounded set of trees. Albeit in practice this non-convexity can be alleviated by growing a large set of trees on bootstrap sub-samples or using gradient boosted forests as oracles for $\mathcal{F}$. Moreover, observe that we solely addressed the optimization problem and postpone the statistical part of random forests (e.g. critical radius) to future work (see also Appendix I).

## 8 Further Applications

In the appendix we also provide further applications of our main theorems. In Appendix D we show how our theorems apply to the case where $\mathcal{H}$ and $\mathcal{F}$ are growing linear sieves, which is a typical approach to non-parametric estimation in the econometric literature (see e.g. Chen and Pouzo [2012]). In Appendix G we analyze the case where $\mathcal{H}$ and $\mathcal{F}$ are function classes defined via shape constraints. We analyze the case of total variation bound constraints and convexity constraints. This applications provides analogues of the convex regression and the isotonic regression to the endogenous regression setting and draws connections to recent works in econometrics on estimation subject to monotonicity constraints Chetverikov and Wilhelm [2017].

## 9 Experimental Analysis

**Experimental Design.** We consider the following data generating processes: for $n_x = 1$ and $n_z \geq 1$

$$y = h_0(x[0]) + e + \delta, \qquad\qquad\qquad \delta \sim N(0, .1)$$
$$x = \gamma z[0] + (1 - \gamma) e + \gamma, \qquad z \sim N(0, 2\, I_{n_z}), e \sim N(0, 2), \gamma \sim N(0, .1)$$

While, when $n_x = n_z > 1$, then we consider the following modified treatment equation:

$$x = \gamma z + (1 - \gamma) e + \gamma,$$

We consider several functional forms for $h_0$ including absolute value, sigmoid and sin functions (more details in Appendix J) and several ranges of the number of samples $n$, number of treatments $n_x$, number of instruments $n_z$ and instrument strength $\gamma$. We consider as classic benchmarks 2SLS with

a polynomial features of degree 3 (2SLS) and a regularized version of 2SLS where ElasticNetCV is used in both stages (Reg2SLS).

In addition to these regimes, we consider high-dimensional experiments with images, following the scenarios proposed in Bennett et al. [2019] where either the instrument $z$ or treatment $x$ or both are images from the MNIST dataset consisting of grayscale images of $28 \times 28$ pixels. We compare the performance of our approaches to that of Bennett et al. [2019], using their code. A full description of the DGP is given in the supplementary material.

**Results.** The main findings are: i) for small number of treatments, the RKHS method with a Nystrom approximation (NystromRKHS), outperforms all methods (Figure 1), ii) for moderate number of instruments and treatments, Random Forest IV (RFIV) significantly outperforms most methods, with second best being neural networks (AGMM, KLayerTrained) (Figure 2), iii) the estimator for sparse linear hypotheses can handle an ultra-high dimensional regime (Figure 3), iv) neural network methods (AGMM, KLayerTrained) outperform the state of the art in prior work [Bennett et al., 2019] for tasks that involve images (Figure 4). The figures below present the average MSE across 100 experiments (10 experiments for Figure 4) and two times the standard error of the average MSE.

| | NystromRKHS | 2SLS | Reg2SLS | RFIV |
|---|---|---|---|---|
| abs | **0.045 ± 0.010** | 0.100 ± 0.035 | 1.733 ± 2.981 | 0.084 ± 0.007 |
| 2dpoly | 0.121 ± 0.014 | **0.036 ± 0.022** | 9.068 ± 16.071 | 0.379 ± 0.022 |
| sigmoid | **0.016 ± 0.003** | 0.071 ± 0.037 | 0.429 ± 0.244 | 0.044 ± 0.006 |
| sin | **0.023 ± 0.003** | 0.090 ± 0.042 | 0.801 ± 0.420 | 0.057 ± 0.007 |
| frequentsin | 0.129 ± 0.005 | 0.193 ± 0.040 | 0.145 ± 0.017 | **0.126 ± 0.010** |
| step | **0.035 ± 0.003** | 0.103 ± 0.043 | 0.497 ± 0.276 | 0.056 ± 0.007 |
| 3dpoly | 0.220 ± 0.037 | **0.004 ± 0.003** | 0.066 ± 0.014 | 0.687 ± 0.069 |
| linear | 0.019 ± 0.003 | 0.038 ± 0.021 | 0.355 ± 0.189 | 0.048 ± 0.005 |
| band | **0.059 ± 0.003** | 0.125 ± 0.051 | 0.085 ± 0.017 | 0.071 ± 0.008 |

Figure 1: $n = 300, n_z = 1, n_x = 1, \gamma = .6$

| | NystromRKHS | 2SLS | Reg2SLS | RFIV | AGMM | KLayerTrained |
|---|---|---|---|---|---|---|
| abs | 0.143 ± 0.005 | 10050.672 ± 13267.141 | 0.122 ± 0.011 | **0.049 ± 0.001** | 0.062 ± 0.003 | 0.127 ± 0.007 |
| 2dpoly | 0.595 ± 0.025 | 5890.128 ± 8261.553 | 4.510 ± 1.245 | 0.346 ± 0.014 | **0.099 ± 0.006** | 0.240 ± 0.014 |
| sigmoid | 0.045 ± 0.003 | 11712.144 ± 16799.716 | 0.091 ± 0.005 | **0.017 ± 0.001** | 0.040 ± 0.001 | 0.024 ± 0.001 |
| sin | 0.058 ± 0.003 | 13769.428 ± 20805.861 | 0.114 ± 0.006 | **0.029 ± 0.001** | 0.074 ± 0.002 | 0.057 ± 0.002 |
| frequentsin | 0.136 ± 0.004 | 12928.749 ± 19554.361 | 0.144 ± 0.004 | **0.120 ± 0.002** | 0.158 ± 0.002 | 0.128 ± 0.002 |
| step | 0.064 ± 0.003 | 12187.342 ± 17814.756 | 0.109 ± 0.004 | **0.027 ± 0.001** | 0.066 ± 0.002 | 0.050 ± 0.001 |
| 3dpoly | 0.648 ± 0.039 | 432.572 ± 596.731 | **0.061 ± 0.005** | 0.444 ± 0.029 | 0.426 ± 0.027 | 0.491 ± 0.029 |
| linear | 0.080 ± 0.002 | 6964.376 ± 9566.774 | 0.107 ± 0.006 | 0.016 ± 0.001 | 0.020 ± 0.001 | **0.013 ± 0.001** |
| band | 0.078 ± 0.004 | 20401.368 ± 29655.000 | 0.090 ± 0.004 | **0.049 ± 0.002** | 0.088 ± 0.003 | 0.074 ± 0.003 |

Figure 2: $n = 2000, n_z = 10, n_x = 10, \gamma = .6$

| $p =$ | 1000 | 10000 | 100000 | 1000000 |
|---|---|---|---|---|
| SpLin | 0.020 ± 0.003 | 0.021 ± 0.003 | - | - |
| StSpLin | 0.020 ± 0.002 | 0.023 ± 0.002 | 0.033 ± 0.002 | 0.050 ± 0.004 |

Figure 3: $n = 400, n_z = n_x := p, \gamma = .6, h_0(x[0]) = x[0]$

| | DeepGMM (Bennett et al. [2019]) | AGMM | KLayerTrained |
|---|---|---|---|
| MNIST$_z$ | 0.12 ± 0.07 | 0.04 ± 0.03 | 0.05 ± 0.02 |
| MNIST$_x$ | 0.34 ± 0.21 | 0.24 ± 0.08 | 0.36 ± 0.20 |
| MNIST$_{xz}$ | 0.26 ± 0.16 | 0.21 ± 0.07 | 0.26 ± 0.11 |

Figure 4: MSE on the high-dimensional DGPs

## Broader Impact

Our work presents a unifying framework for the classical problem of generalized method of moments (GMM). Our framework can be easily applied in a diverse range of scenarios and is efficient at yielding accurate results even in high-dimensional scenarios. Moreover, we provide a strong theoretical foundation for our framework which shows how to use regularization or constrained optimization to obtain a theoretical bound on the performance of the underlying estimator. In providing the theoretical upper bounds on the generalization error of the GMM estimator, we bring in a statistical learning view, which is novel and has many related directions to explore in future. A

number of recent works (Bennett et al. [2019], Hartford et al. [2017]) try to tackle the GMM problem using neural networks. We build on this direction and present many novel ways in which one can use kernel ideas combined with neural networks to produce different estimators. We believe this flexibility offered by our framework speaks for its generality and potential for future impact. On a more broader front, within the econometrics literature, the GMM problem arises in many scenarios where a policy decision which impacts humans is to be made. A better and theoretically sound estimation procedure would result in better decisions made. Moreover, given instances of the problem where the decisions downstream impact humans, quantities such as privacy and fairness could also be incorporated into our framework for GMM estimation. A better and theoretically sound estimation procedure would result in better decisions made. Moreover, given instances of the problem where the decisions downstream impact humans, quantities such as privacy and fairness could also be incorporated into our framework for GMM estimation.

## Acknowledgments and Disclosure of Funding

Nishanth Dikkala was supported by NSF Awards IIS-1741137, CCF1617730 and CCF-1901292.

## Footnotes

[1]For our main theorem we do not even require completeness, i.e. there can be many $h_* \in \mathcal{H}$ such that $\mathbb{E}[(y - h_*(x))|z] = 0$. Even without completeness we can still show convergence to an equivalence class of $h_0$ under the projected mean-squared error metric. This adds robustness to our main theorem, as compared to existing approaches in econometrics that typically assume completeness to provide any form of guarantee. Such robustness could prove useful in settings with weak instruments where point identifiability only holds asymptotically. Imposing further assumptions on completeness or the degree of ill-posedness can allow one to relate the projected metric to the standard root-mean-squared-error metric and show RMSE-convergence to $h_0$ (see Appendix C.4).

[2] By setting $\lambda = \delta^2/U$ and $\mu = 2\lambda\left(4L^2 + 27U/B\right)$, using an $\ell_\infty$ norm in both function spaces, and taking $U, B \to \infty$. Observe that we can also take $L = 1$, since $\|Th\|_\infty \leq \|h\|_\infty$ for any $T$.

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
