[Supplementary Material]

## Supplementary Material:
## Minimax Estimation of Conditional Moment Models

## Contents

## A  Further Discussion on Related Work

Identification and estimation of endogenous regression problems via the method of instrumental variables (IV) has a long history in econometrics and causal inference [Bowden and Turkington, 1990, Angrist and Krueger, 1991, Imbens and Angrist, 1994, Balke and Pearl, 1997, Angrist and Krueger, 2001, Stock and Trebbi, 2003], dating back to early works in empirical economics [Wright, 1928] (for more detailed exposition see [Angrist and Pischke, 2008, Pearl, 2009b,a, Imbens and Rubin, 2015]). The most prevalent approach for estimating endogenous regression models with instruments is assuming low-dimensional linear relationships, i.e. $h_0(x) = \langle x, \theta \rangle$ and invoking the two-stage-least-squares (2SLS) algorithm: the treatment is regressed on the instruments via a first stage linear regression, $x \sim z$ to learn a model $\hat{f}(z) = \langle \hat{\beta}, z \rangle$ and subsequently the outcome $y$ is regressed on the predicted treatments from the first stage linear regression model, i.e. $y \sim \hat{f}(z)$. The coefficient in the final regression is taken to be the estimate of $\theta$.

### A.1  Non-Parametric IV Regression in Econometrics and Statistics

Non-parametric and high-dimensional versions of the IV estimation problem have received great attention by the econometrics and statistics community in the past two decades [Newey and Powell, 2003, Blundell et al., 2007, Chen and Pouzo, 2012, Chen and Christensen, 2018, Hall et al., 2005, Horowitz, 2007, 2011, Darolles et al., 2011, Chen and Pouzo, 2009].

**Sieve-based 2SLS**  Newey and Powell [2003] consider a non-parametric analogue of the 2SLS regression, where the non-parametric model $h_0(x)$ is approximated by a linear function on a growing feature space, i.e. $h_0(x) \approx \langle \phi(x), \theta \rangle$ and subsequently, the conditional expectations $\mathbb{E}[\phi(x) \mid z]$ are also approximated via linear functions $f(z) \approx \langle \psi(z), \beta \rangle$. Then a 2SLS estimation method is applied on these transformed feature spaces. The authors show asymptotic consistency of the resulting estimator, assuming that the approximation error goes to zero.

**Sieve-based regularized minimum distance estimation** Arguably the closest to our work is that of Chen and Pouzo [2012] (in particular their Theorem 4.1), who consider estimation of non-parametric function classes and estimation via the method of sieves and a penalized minimum distance estimator of the form: $\min_{h \in \mathcal{H}} \mathbb{E}[\mathbb{E}[y - h(x) \mid z]^2] + \lambda R(h)$, where $R(h)$ is a regularizer. The authors approximate the function class $\mathcal{H}$ by linear functions in a growing feature space. Subsequently, they also estimate the function $m(z) = \mathbb{E}[y - h(x) \mid z]$ based on another growing sieve.

Though it may seem at first that the approach in that paper and ours are quite distinct, the population limit of our objective function coincides with theirs. To see this, consider the simplified version of our estimator presented in (6), where the function classes are already norm-constrained and no norm based regularization is imposed. Moreover, for a moment consider the population version of this estimator, i.e.

$$\min_{h \in \mathcal{H}} \max_{f \in \mathcal{F}} \Psi(h, f) - \|f\|_2^2 = \min_{h \in \mathcal{H}} \max_{f \in \mathcal{F}} \mathbb{E}[(y - h(x))f(z) - f(z)^2]$$

Observe that if $\mathcal{F}$ is expressive enough (if $T(h_0 - h) \in \mathcal{F}$), then the maximizing test function is $\frac{1}{2}\mathbb{E}[y - h(x) \mid z] = \frac{1}{2}\mathbb{E}[h_0(x) - h(x) \mid z]$. Then by the law of iterated expectations, the population criterion becomes:

$$\min_{h \in \mathcal{H}} \mathbb{E}\left[(y - h(x))\frac{1}{2}\mathbb{E}[y - h(x) \mid z] - \frac{1}{4}\mathbb{E}[y - h(x) \mid z]^2\right] = \min_{h \in \mathcal{H}} \frac{1}{4}\mathbb{E}\left[\mathbb{E}[y - h(x) \mid z]^2\right]$$

Thus in the population limit and without norm regularization on the test function $f$, our criterion is equivalent to the minimum distance criterion analyzed in Chen and Pouzo [2012]. Another point of similarity is that we prove convergence of the estimator in terms of the pseudo-metric, the projected MSE defined in Section 4 of Chen and Pouzo [2012] - and like that paper we require additional conditions to relate the pseudo-metric to the true MSE.

The present paper differs in a number of ways: (i) the finite sample criterion is different; (ii) we prove our results using localized Rademacher analysis which allows for weaker assumptions; (iii) we consider a broader range of estimation approaches than linear sieves, necessitating more of a focus on optimization.

Digging into the second point, Chen and Pouzo [2012] take a more traditional parameter recovery approach which requires several minimum eigenvalue conditions and several regularity conditions to be satisfied for their estimation rate to hold (see e.g. their Assumptions 3.1, 3.2, 3.3, 4.1 and C.1). This is analogous to a mean squared error proof in an exogenous linear regression setting, that requires the minimum eigenvalue of the feature co-variance to be bounded away from zero. Moreover, such parameter recovery methods seem limited to the growing sieve approach, since only then one has a clear finite dimensional parameter vector to work on for each fixed $n$.

In contrast we work with infinite dimensional parameter spaces directly and our analysis makes no further assumptions other than boundedness of the random variables and the conditional moment restriction in order to provide a projected MSE rate. We do not require that the hypothesis space be a convex set, nor that the moment is path-wise differentiable with respect to $h$. Relaxing these assumptions is important, since they are violated in three of our leading examples: linear hypothesis spaces with hard sparsity constraints or for neural network spaces or for tree based regressors. Another benefit of the localized Rademacher analysis is that we do not require a preliminary proof of consistency, which is typical of more classical approaches to MSE rates. Such proofs typically require that $n$ be larger than some constant before the convergence rate kicks in, so that the estimator is within some small ball around the truth. This constant can sometimes be prohibitively large. Our convergence rate is global and holds without any lower bound condition on $n$. The sieve method is most closely related to our RKHS section (and the expository sieve Appendix D), where essentially we consider infinite dimensional linear function spaces. However, unlike the sieve method, we do not clip the eigenfunctions to a finite set that is growing, but rather impose an RKHS penalty. We show that this approach has advantages in auto-tuning to the ill-posedness of the problem. Finally, we do not require a bound on the ill-posedness of the problem in order to prove convergence rates in terms of the pseudo-metric - this bound is only needed in post-processing to relate the pseudo-metric to the MSE. By contrast Chen and Pouzo [2012] use the bounded ill-posedness condition (Assumption 4.1) to prove convergence in the pseudo-metric.

As a concrete example of the differences in the analysis, we apply our main Theorem 1 for the case where $\mathcal{H}$ and $\mathcal{F}$ are growing sieves, equipped with the parameter $\ell_2$ norms, i.e. $\mathcal{H} =$

$\left\{ \langle \theta, \phi_n(\cdot) \rangle : \theta \in \mathbb{R}^{k_n} \right\}$, $\mathcal{F} = \left\{ \langle \beta, \psi_n(\cdot) \rangle : \beta \in \mathbb{R}^{m_n} \right\}$, $\| \langle \theta, \phi_n(\cdot) \rangle \|_{\mathcal{H}} = \|\theta\|_2$, $\| \langle \beta, \psi_n(\cdot) \rangle \|_{\mathcal{F}} = \|\beta\|_2$, for some fixed and growing feature maps $\phi_n(\cdot)$, $\psi_n(\cdot)$. In that case $\eta_n$ will correspond to the approximation error of the sieve $\psi_n$ that is used for the test function space and, if we choose $h_* = \arg\min_{h \in \mathcal{H}} \|h_* - h_0\|_2$, then $\|T(h_* - h_0)\|_2 \leq \|h_* - h_0\|_2 =: \epsilon_n$, will correspond to the approximation error of the sieve $\phi_n$ that is used for approximating the model $h_0$. In that case, Theorem 1 gives a bound of $O\left( \delta_n \|\theta_*\|_2 + \eta_n + \epsilon_n \right)$, where $\theta_*$ is the $\ell_2$ norm of the parameter of the projection of $h_0$ on the sieve space for the model, i.e $\arg\min_{\theta \in \mathbb{R}^{k_n}} \|\langle \theta, \phi(\cdot) \rangle - h_0\|_2^2$. Moreover, $\delta$ is a bound on the critical radius of $\mathcal{F}_U$ and $\mathcal{G}_{B,U}$. Since both are finite dimensional linear functions, via standard covering arguments (see Corollary 5), we can bound $\delta = O\left( \sqrt{\frac{\max\{k_n, m_n\} \, \log(n)}{n}} \right)$.[3]

Combined with ill-posedness conditions provided in [Chen and Pouzo, 2012], our results can thus give an alternative proof to the results in [Chen and Pouzo, 2012] that i) do not make minimum eigenvalue conditions, ii) provide adaptivity to $\|\theta_*\|_2$, without knowledge of it, thereby justifying theoretically the use of the regularization term $R(h)$, that was mostly proposed for experimental improvement in [Chen and Pouzo, 2012]. We provide a more thorough exposition of how our main theorem applies to the case of growing sieves in Appendix D.

The localized Rademacher analysis also allows us to consider hypothesis spaces that are not linear sieves, such as neural nets and random forests. This introduces some new optimization difficulties, as the estimator cannot be written in closed form (as it can for linear sieves). Our work gives several solutions for these difficulties, via iterative first order algorithms. Intuitively, our optimization algorithms gradually and iteratively make gradient steps towards solving both optimization problems (of regressing $y - h(x)$ on $z$ and minimizing $\mathbb{E}[\mathbb{E}[y - h(x) \mid z]^2]$ over $\mathcal{H}$), as opposed to calculating full solutions of either problem. This formulation allows us to work with arbitrary hypothesis spaces and not just linear sieves.

**Tikhonov regularized minimum distance estimation**  The work of [Hall et al., 2005, Darolles et al., 2011, Horowitz, 2007, 2011], considers a Tikhonov regularized minimum distance estimator, as opposed to a sieve-based approach. In particular, they consider the population criterion:

$$\min_{h \in \mathcal{H}} \|T(y - h)\|_2^2 + \lambda \|h\|_2^2 \equiv \mathbb{E}\left[ \mathbb{E}[y - h(x) \mid z]^2 \right] + \lambda \|h\|_2^2$$

This is equivalent to the minimum distance criterion of Chen and Pouzo [2012], albeit with an added and crucial $\ell_2$ regularization penalty on the hypothesis. The $\ell_2$ or Tikhonov regularization achieves two objectives: i) it regularizes the estimate to avoid overfitting (as in the case of an exogenous regression setting), ii) it protects against the ill-posedness of the inverse problems, by avoiding estimates that put a lot of weight on the non-smooth eigenfunctions of the singular value decompsition of the operator $T$.

Darolles et al. [2011] consider the closed form solution to the minimization problem, which takes the form:

$$h^\lambda = \left( \lambda\, I + T^* T \right)^{-1} T^* r$$

where $r = Ty = \mathbb{E}[y \mid z]$ and $T^*$ is the adjoint operator of $T$, defined as $(T^* \phi)(x) = \mathbb{E}[\phi(z) \mid x]$. The authors further make minimal regularity assumptions that imply that the operator $T$ admits a singular value decomposition $(\sigma_i, \phi_i, \psi_i)_{i \in I}$ for some countable set $I$, with $1 = \sigma_0 \geq \sigma_1 \geq \ldots$, $\phi_i : \mathcal{X} \to \mathbb{R}$ and $\psi_i : \mathcal{Z} \to \mathbb{R}$, i.e. $T\phi_i = \sigma_i \psi_i$ and $T^* \psi_i = \sigma_i \phi_i$. Moreover, one can express the $T$ operator as:

$$Th = \sum_{i \in I} \sigma_i \langle h, \phi \rangle \psi_i$$

where the $\langle, \rangle$ is the inner product associated with the corresponding $\ell_2$ metric spaces, i.e. $\langle h, \phi \rangle = \mathbb{E}[h(x)\phi(x)]$ and $\langle f, \psi \rangle = \mathbb{E}[f(z)\psi(z)]$. Under these assumptions the optimal solution to the population criterion can also be written as:

$$h^\lambda = \sum_{i \in I} \frac{\sigma_i}{\lambda + \sigma_i^2} \langle r, \psi_i \rangle \phi_i$$

Intuitively, the functions $\phi_i, \psi_i$ correspond roughly to the sieve functions that are used in the sieve methods, e.g. Chen and Pouzo [2012], albeit instead of clipping the function $h$ to be supported on the

first $k_n$ eigenfunctions of the singular value decomposition, they impose a Tikhonov regularization, which penalizes the large eigenfunctions in a smoother manner.

Darolles et al. [2011] take the latter approach to estimation by first estimating the conditional operators $T$ and $T^*$ from samples and the conditional expectation $\mathbb{E}[y \mid z]$, in a first stage. They achieve this using non-parametric kernel density estimation methods to estimate the densities $p(x, z)$, $p(z, x)$, $p(y, z)$, $p(z)$ and $p(x)$. Subsequently they estimate the operators $T, T^*$ in a plug-in manner, e.g. $\hat{T}h = \int_{\mathcal{X}} h(x) \frac{\hat{p}(x,z)}{\hat{p}(z)} dx$. Finally, they consider the plug-in estimate: $\hat{h}^\lambda = (\lambda I + \hat{T}^* \hat{T})^{-1} \hat{T}^* \hat{r}$.

The crucial assumption in this line of work (see e.g. Hall et al. [2005], Darolles et al. [2011]) is what is known as the *source condition*, where the true hypothesis $h_0$ is assumed to be smooth in the metric space defined by the eigenfunctions of the operator $T$. More concretely:

$$\sum_{i \in I} \frac{\langle h, \phi_i \rangle^2}{\sigma_i^{2\beta}} < \infty \qquad \text{(source condition)}$$

This assumption has the crucial implication that $\|h^\lambda - h\|_2^2 = O\left(\lambda^{\min\{\beta, 2\}}\right)$, thus providing a control on the bias of the regularized estimate with respect to the MSE metric.

The decay of the singular values $\sigma_i$ of $T$, is related to the rate of decay $\tau_m$ that we analze in E.4. For instance, if the operator $T$ has as right eigenfunctions the eigenfunctions of the kernel $K$, then observe that: $Te_i = \lambda_i \psi_i$ and $\mathbb{E}[\mathbb{E}[e_i(x) \mid z]\mathbb{E}[e_j(x) \mid z]] = \mathbb{E}[(Te_i)(z)(Te_i)(z)] = \lambda_i^2 \mathbb{E}[\phi_i(z)\phi_j(z)]$. Thus we get that $\mathbb{E}[\mathbb{E}[e_i(x) \mid z]\mathbb{E}[e_i(x) \mid z]] = \lambda_i^2$ and for $i \neq j$, $\mathbb{E}[\mathbb{E}[e_i(x) \mid z]\mathbb{E}[e_j(x) \mid z]] = 0$. Thus the conditions in Lemma 11 are satisfied with $\tau_m = \sigma_m^2$ and $c = 0$. Thus by imposing an RKHS norm penalty and assuming the latter eigendecomposition, we can get rates of the form:

$$\|\hat{h} - h_*\|_2 = \min_{m \in \mathbb{N}_+} \left( \frac{4\delta^2}{\sigma_m^2} + B\lambda_{m+1} \right)$$

Moreover, observe that: $\langle h, e_i \rangle = a_i$ and that if $\|h\|_K \leq B$, then $\sum_{i \in I} \frac{a_i^2}{\lambda_i} \leq B$ and hence: $\sum_{i \geq m} a_i^2 \leq \lambda_m B$, where $\lambda_i$ are the eigenvalues of the kernel. In this notation, the source condition is equivalent to: $\sum_{i \in I} \frac{a_i^2}{\sigma_i^{2\beta}} < \infty$. An RKHS norm bound implies that the function is smooth in the metric space defined by the eigenfunctions of $T$ and $B$ is a level of smoothness. However, the level of smoothness is also governed by the eigendecay of the kernel $K$. In some sense, the eigendecay of the kernel governs the numerator of the source condition, while the eignedecay of the operator $T$ governs the denominator. Our estimator adapts to these two quantities and automatically and optimally balances them, by imposing an RKHS norm penalty. However, the two conditions are slightly incomparable, even though they capture similar constraints.

## A.2 High Dimensional IV Regression

Instrumental variable estimation with high dimensional sparse linear models was analyzed in [Gautier et al., 2011, Fan and Liao, 2014]. [Gautier et al., 2011] proposes a Dantzig selector analogue for endogenous regression. Our work on sparse linear hypotheses provides a minimax formulation alternative to the Dantzig selector. [Fan and Liao, 2014] propose a variant of the optimally re-weighted generalized method of moments in high-dimensions with hard sparsity. Our results apply to the setting analyzed in [Fan and Liao, 2014] and unlike [Fan and Liao, 2014] our estimation algorithm is computationally efficient (see F for more details).

## A.3 Non-Parametric IV Regression and Machine Learning

There is also a growing body of work in the machine learning literature on the non-parametric instrumental variable regression problem [Hartford et al., 2017, Bennett et al., 2019, Singh et al., 2019, Muandet et al., 2019, 2020].

**Neural networks** The seminal work of Hartford et al. [2017] provided a methodology for training neural networks that solve the instrumental variable problem by taking a non-parametric analogue of the two stage least squares method. A preliminary version of this work [Lewis and Syrgkanis, 2018] proposed a minimax criterion for training neural networks that solve the IV problem (albeit, crucially,

with no regularization on the test functions and no formal guarantees on estimation rates). It however, does not provide statistical guarantees of the resulting estimator apart from a fully non-parametric rate that grows exponentially with dimension. A crucial difference of our work from [Lewis and Syrgkanis, 2018] is that they don't penalize the objective with the norm of the test function which is the key idea that enables our fast rates (based on critical radius of $\mathcal{F}$). Finally, [Lewis and Syrgkanis, 2018] only provides experimental results for neural nets, while here we provide experimental and theoretical results for many other function classes of interest. Bennett et al. [2019] also considered a minimax criterion with a variance penalty. Albeit the variance penalty they impose is not the second moment of the test functions and depends on a preliminary estimate of the true model. Moreover, they only show asymptotic consistency of their estimate and not finite sample rates and primarily focus on neural network applications (see Section 6 for more details).

**Reproducing kernel function spaces**   Singh et al. [2019] consider a RKHS analogue of Hartford et al. [2017], where the hypothesis space $h$ fall in an RKHS and the conditional distribution of $X$ conditional on $Z$ is represented via a conditional kernel mean embedding. They offer very strong finite RKHS-norm rates on the estimated $h$, which typically imply sup-norm rates of the recovered function. Albeit, we focus on projected MSE and MSE rates and achieve faster rates as a function of the eigendecay of the kernel and the degree of ill-posedness. Moreover, the work of Singh et al. [2019] makes several stronger *prior* assumptions, that control the smoothness of the function within the kernel, assumptions that are typical of RKHS norm guarantees in kernel ridge regression [Caponnetto and De Vito, 2007], but which are not required for the weaker MSE metric. In essence, the prior condition imposes that the RKHS norm of error of the projection of the true function on the top $J$ eigenfunctions $\|(I - P_J)f_0\|_{\mathcal{H}}^2 = \sum_{j>J} \frac{a_j^2}{\lambda_j} \to 0$ as $J$ goes to infinity. This is not the case for all functions in the RKHS, which only implies that the $\ell_2$-norm of the error of these projections vanishes, i.e. $\|(I - P_J)f_0\|_2^2 = \sum_{j>J} a_j^2 \stackrel{J \to \infty}{\to} 0$.

**RKHS and neural network training**   Muandet et al. [2019] also propose a method that is very related to the second moment penalized method that we propose, albeit the motivation stems from a different dual formulation of the two-stage-least-squares problem presented in [Hartford et al., 2017] and similar to [Bennett et al., 2019] only offer asymptotic consistency of the estimator and only focus on RKHS function spaces. Finally, Muandet et al. [2020] consider the version of the minimax criterion that does not impose the second moment penalty on $f$, and make the important observation that for RKHS function spaces, the internal maximization takes a closed form, leading to a pairwise sample criterion (see Equation (12) and Equation (15)). Moreover, they focus primarily on hypothesis testing as opposed to estimation. The un-penalized criterion can have sub-optimal convergence guarantees, as it does not posses the property that as the hypothesis of the learner gets close to the truth, then the adversary is testing smaller functions in terms of variance. The inability to achieve the fast rates attained via the critical radius was the main reason why we introduced the second moment penalty. The suboptimality of the un-penalized kernel based criterion was also proven in the context of hypothesis testing by Balasubramanian et al. [2017], who also show that a form of second moment penalization can yield hypothesis tests with optimal power, when the alternative is very close to the null. Moreover, for RKHS, we show that the penalized method still admits a closed form solution, albeit now the closed form depends on the inverse of a kernel matrix, which makes it less amenable to gradient training as we discuss in 6.

# B   Beyond the IV Moments

Our results easily extend to arbitrary moments that are linear in $h$, which can capture several other problems in econometrics and causal inference, but for simplicity of exposition we focus on the case of moments of the form $y - h(x)$. Moreover, our results can also be extended to non-linear and non-smooth moments $\psi(y; h(x))$, albeit in that case our convergence rates will be with respect to the distance metric: $d(\hat{h}, h) = \sqrt{\mathbb{E}[\mathbb{E}[\psi(y; \hat{h}(x)) - \psi(y; h(x)) \mid z]^2]}$ as opposed to the projected MSE distance. For instance, in the case of $\alpha$-quantile IV regression: $\psi(y; h(x)) = a - 1\{y \leq h(x)\}$ and the distance metric corresponds to: $d(\hat{h}, h_0) = \sqrt{\mathbb{E}[\mathbb{E}[1\{y \leq h(x)\} - \alpha \mid z]^2]}$.

## C  Supplementary Discussion of Main Theorems

### C.1  Adaptivity of Regularized Estimator

Suppose that we know that for $B, U = 1$, we have that functions in $\mathcal{H}_B, \mathcal{F}_U$ have ranges in $[-1, 1]$ as their inputs range in $\mathcal{X}$ and $\mathcal{Z}$ correspondingly. Then our Theorem requires that we set: $\lambda \geq \delta^2$ and $\mu \geq 2\lambda(4L^2 + 27)$, where $\delta^2$ depends on the critical radius of the function class $\mathcal{F}_1$ and $\mathcal{G}_1$. Observe that none of these values depend on the norm of the benchmark hypothesis $\|h_*\|_{\mathcal{H}}$, which can be arbitrary and not constrained by our theorem. For instance, if we knew that the true model $h_0 \in \mathcal{H}$ and $T(h - h_0) \in \mathcal{F}_{L^2\|h - h_0\|_{\mathcal{H}}^2}$, then we can apply the latter theorem to get rates of the form:

$$O\left(\delta \max\left\{1, \|h_0\|_{\mathcal{H}}^2\right\}\right)$$

with $\lambda = \delta^2$ and $\mu = 2\delta^2(4L^2 + 27)$. This hyperparameter tuning only requires knowledge of the critical radius of the function classes $\mathcal{F}_1$ adn $\mathcal{H}_1$ and the Lipschitz constant of the operator $T$, but does not require knowledge of the norm of the true model $\|h_0\|_{\mathcal{H}}$, nor upper bounds on it. If the true model does not fall in the hypothesis $\mathcal{H}$, then observe that we also require knowledge of the unconstrained approximation error, i.e. if we knew that:

$$\inf_{h \in \mathcal{H}} \|h - h_0\|_2 \leq \epsilon_n$$

and that $T(h - h_0) \in \mathcal{F}_{L\|h - h_0\|_{\mathcal{H}}}$, then we can choose $\delta \geq \epsilon_n$ to get rates of the form:

$$O\left(\delta \max\left\{1, \|h_*\|_{\mathcal{H}}^2\right\} + \epsilon_n\right)$$

where $h_* = \arg\inf_{h \in \mathcal{H}} \|h - h_0\|_2$. Again we do not require knowledge of the norm of the unconstrained projection, $\|h_*\|_{\mathcal{H}}$, just bounds on the approximation error of the unconstrained function space. Then the regularized estimator adapts to the norm of the projection of the true model on $\mathcal{H}$. These results are inline with recent work on statistical learning theory [Lecué and Mendelson, 2017, 2018] for square losses and extend these qualitative insights to the minimax objectives that we deal with.

### C.2  Critical Radius and Rademacher Complexity via Covering

The critical radius of a function class is characterized to within a constant factor by it's empirical localized Rademacher critical radius, which subsequently is chracterized by the empirical entropy integral. The empirical Rademacher complexity of a function class $\mathcal{G} : \mathcal{V} \to [-1, 1]$, for a given set of samples $S = \{v_i\}_{i=1}^n$ is defined as:

$$R_S(\delta; \mathcal{G}) = \mathbb{E}_{\{\epsilon_i\}_{i=1}^n}\left[\sup_{g \in \mathcal{G} : \|g\|_{2,n} \leq \delta} \frac{1}{n} \sum_i \epsilon_i g(v_i)\right]$$

The empirical critical radius is defined as any solution $\hat{\delta}_n$ to:

$$R_S(\delta; \mathcal{G}) \leq \delta^2$$

Proposition 14.1 of Wainwright [2019] shows that w.p. $1 - \zeta$,

$$\delta_n = O\left(\hat{\delta}_n + \sqrt{\frac{\log(1/\zeta)}{n}}\right). \tag{8}$$

Thus we can choose $\delta$ in our main theorems based on the empirical critical radius $\hat{\delta}_n$.

Moreover, an upper bound on the empirical critical radius can be obtained via the empirical covering integral defined as follows. An empirical $\epsilon$-cover of $\mathcal{G}$, is any function class $\mathcal{G}_\epsilon$, such that for all $g \in \mathcal{G}$, $\inf_{g_\epsilon \in \mathcal{G}_\epsilon} \|g_\epsilon - g\|_{2,n} \leq \epsilon$. We denote with $N(\epsilon, \mathcal{G}, S)$ as the size of the smallest empirical $\epsilon$-cover of $\mathcal{G}$. The empirical metric entropy of $\mathcal{G}$ is defined as $H(\epsilon, \mathcal{G}, S) = \log(N(\epsilon, \mathcal{G}, S))$. An empirical $\delta$-slice of $\mathcal{G}$ is defined as $\mathcal{G}_{S,\delta} = \{g \in \mathcal{G} : \|g\|_{2,n} \leq \delta\}$. Then the empirical critical radius of $\mathcal{G}$ is upper bounded by any solution to the inequality:

$$\int_{\delta^2/8}^{\delta} \sqrt{\frac{H(\epsilon, \mathcal{G}_{S,\delta}, S)}{n}} d\epsilon \leq \frac{\delta^2}{20} \tag{9}$$

Observe that a conservative upper bound on $\hat{\delta}_n$ comes from replacing $\mathcal{G}_{S,\delta}$ inside the integral with $\mathcal{G}$, i.e. when we do not restrict the function class to be in an empirical $\delta$-slice, when calculating it's empirical metric entropy. For many function classes (e.g. parametric $\ell_2$-balls, RKHS, high-dimensional sparse parametric spaces, VC-subgraph classes) this still yields tight results. For some other cases, such as $\ell_1$-balls centered around a sparse parameter, this can be loose.

When we make this relaxation, then observe that we can derive an upper bound on the critical radius of $\mathcal{G}_{B,U}$, as a function of the empirical metric entropy of $\mathcal{H}$ and $\mathcal{F}$. Observe that if $\mathcal{H}_\epsilon$ is an empirical $\epsilon$-cover of $\mathcal{H}$ and $\mathcal{F}_\epsilon$ is an empirical $\epsilon$-cover of $\mathcal{F}_U$, then since $\mathcal{H}$ contains functions uniformly bounded in $[-1, 1]$, we have that:

$$\inf_{h \in \mathcal{H}_\epsilon, f_\epsilon \in \mathcal{F}_\epsilon} \|(h_\epsilon - h)f_\epsilon - (h - h_*)f_h^U\|_{2,n} \le 2\|h_\epsilon - h_*\|_{2,n} + 2\|f_\epsilon - f_h^U\|_{2,n} \le 4\epsilon$$

Thus, the product of these two spaces is an $\epsilon$-cover of the function class $\mathcal{G}$ defined in Equation (3). Hence, the empirical metric entropy of $\mathcal{G}$ satisfies:

$$H(\epsilon, \mathcal{G}_{B,U}, S) \le H(\epsilon/4, \mathcal{H}_B, S) + H(\epsilon/4, \mathcal{F}_U, S)$$

Thus by applying Proposition 14.1 of Wainwright [2019] we get the following corollary.

**Corollary 5.** *Suppose that $\hat{\delta}_n$ satisfies the inequality:*

$$\int_{\delta^2/8}^{\delta} \sqrt{\frac{H(\epsilon/4, \mathcal{H}_{2B}, S) + H(\epsilon/4, \mathcal{F}_{3U}, S)}{n}} d\epsilon \le \frac{\delta^2}{20}$$

*Then w.p. $1 - \zeta$, $\delta_n \le O\left(\hat{\delta}_n + \sqrt{\frac{\log(1/\zeta)}{n}}\right)$, where $\delta_n$ is the maximum of the critical radii of $\mathcal{F}_{3U}$, $\mathcal{G}_{B,U}$ and $\hat{\mathcal{G}}_{B,U}$.*

For instance, if $\mathcal{H}$ and $\mathcal{F}$ is assumed to be a VC-subgraph class with constant VC dimension, then the above is satisfied for $\hat{\delta}_n = O\left(\sqrt{\frac{\log(n)}{n}}\right)$.

### C.3 Solving the Min-Max Optimization Problem

In this section we outline some strategies for addressing the empirical min-max problem required by the estimators described in Equations (4) and (7). In subsequent sections, we will present instances of these optimization approaches for each of the function classes that we consider.

First observe that if the hypothesis space can be parameterized as $h(x; \theta)$, such that the moment $\psi(y; h(x; \theta))$ is convex in $\theta$ and the inner optimization problem is solvable in closed form then we can solve the empirical problem via subgradient descent: i.e. letting

$$f_*(\cdot; h) := \underset{f \in \mathcal{F}}{\arg\sup} \, \Psi_n(h, f) - \lambda \Phi(f),$$
$$\theta_{t+1} := \theta_t - \eta \left(\mathbb{E}_n \left[f_*(z; h(x; \theta_t)) \nabla_\theta h(x; \theta_t)\right] + \mu \nabla_\theta R(h(\cdot; \theta_t))\right)$$

where $\Phi, R$ are the regularizers on $f$ and $h$ correspondingly. After $T$ iterations, the average parameter $\bar{\theta} = \frac{1}{T} \sum_{t=1}^{T} \theta_t$, will correspond to an $O\left(T^{-1/2}\right)$ approximate solution to the min-max problem. This approximate solution will satisfy the same guarantees as $\hat{h}$ presented in Theorem 1 and Theorem 2, augmented by an extra $O\left(T^{-1/2}\right)$ additive factor.

Many times, even if the hypothesis space is not parameterizable by a finite dimensional parameter vector $\theta$, universally, we can invoke characterizations (typically referred to as *representer theorems*), that prove that the empirical solution can always be expressed in terms of a finite set of parameters (many times of the order of the number of samples). This is for instance the case when $\mathcal{F}$ and $\mathcal{H}$ belong to a Reproducing Kernel Hilbert space, as we will see in Section 4. In such settings, we will see that even the overall min-max optimization problem can be expressed in closed form, involving only matrix inversions and mutliplications, with matrices of size of the order of $n^2$.

Since the min-max problem does not have a smooth gradient, one can also benefit by invoking algorithms that are tailored to saddle point problems. These improvements typically assume some

structure on the inner optimization problem. For instance, if the function $f$ can be parameterizedd as $f(\cdot; w)$ such that the inner maximization problem is concave in $w$ then faster than $T^{-1/2}$ optimization rates can be achieved. We will see examples of such settings in the high-dimensional linear function class setting in Section 5. The following set of papers provide examples of algorithms that achieve $T^{-1}$ approximation rates (see e.g. Nesterov [2005], Nemirovski [2004], Rakhlin and Sridharan [2013], Mokhtari et al. [2019]).

One simple such algorithm is the simultaneous optimistic mirror descent algorithm proposed in Rakhlin and Sridharan [2013] and also recently analyzed by several papers, both theoretically and empirically, in the context of non-convex optimization problems (see e.g. Daskalakis et al. [2017], Mertikopoulos et al. [2018]). In this algorithm, instead of fully solving the internal optimization problem, we only take gradient steps. However, it modifies the gradient descent algorithm to incorporate a notion of *optimism* (i.e. that the next gradient will look similar to the last gradient). In particular, if we use the short-hand notation $\Psi_n(\theta, w) := \Psi_n(h(\cdot; \theta), f(\cdot; w))$, then in the simplified setting where we have no regularization on $\theta, w$, the algorithm is described via the following update dynamics:

$$\theta_{t+1} = \theta_t - 2\eta \nabla_\theta \Psi_n(\theta_t, w_t) + \eta \nabla_\theta \Psi_n(\theta_{t-1}, w_{t-1})$$
$$w_{t+1} = w_t + 2\eta \nabla_w \Psi_n(\theta_t, w_t) - \eta \nabla_w \Psi_n(\theta_{t-1}, w_{t-1})$$

Convex constraints on $\theta$ and $w$ can be easily incorporated via projection steps and we defer to Rakhlin and Sridharan [2013] for the formal definition of the algorithm in that setting. Similarly, for the regularized versions one would simply replace $\Psi_n$ with its regularized counterparts.

Unlike the sub-gradient descent approach, the simultaneous optimistic gradient dynamics, with the regularized version of our estimator, can also be implemented in a stochastic gradient manner, where a mini-batch of samples are drawn at each step (with replacement), from the empirical set of samples and $\Psi_n$ is replaced with the empirical expectation over that sub-sample. This can enable applications where storing all the dataset in-memory is prohibitive. Moreover, this algorithm has variants that have been proven beneficial for neural nets (see, e.g. the Optimistic Adam algorithm of Daskalakis et al. [2017], also used in the related work of Bennett et al. [2019] in a generalized method of moments setup). Properties of simultaneous gradient dynamics in non-convex/non-concave settings have also been a topic of recent interest in the machine learning community and recent techinques from this line of work can be invoked to empirically solve the optimization problem (see e.g. Jin et al. [2019], Nouiehed et al. [2019], Thekumparampil et al. [2019], Yang et al. [2020], Lin et al. [2020]).

### C.4 From Projected MSE to MSE: Measure of Ill-Posedness

If we want to get a bound on the RMSE of $\hat{h}$, i.e. $\|h - h_0\|_2$, then we need to bound the quantity:

$$\tau^*(\delta) = \sup_{h \in \mathcal{H}_B : \|T(h - h_*)\|_2 \le \delta} \|h - h_*\|_2$$

In fact, it suffices to bound the measure of ill-posedness of the operator $T$ with respect to the function class $\mathcal{H}_B$, defined as:

$$\tau := \sup_{h \in \mathcal{H}_B} \frac{\|h - h_*\|_2}{\|T(h - h_*)\|}.$$

Both of these measures have been used in the literature on conditional moment models. For instance, Chen and Pouzo [2012] defines both of these measures for the case where $\mathcal{H}_B$ is a space of growing linear sieves. In that case, the second measure $\tau$ is typically referred to as the *sieve measure of ill-posedness*. Then observe that Theorem 1 implies that:

$$\|\hat{h} - h_*\|_2 \le \tau \|T(\hat{h} - h_*)\|_2 \le O\left(\tau \delta_n + \tau \|T(h_* - h_0)\|_2\right) \le O\left(\tau \delta_n + \tau \|h_* - h_0\|_2\right)$$

which by a triangle inequality also implies that:

$$\|\hat{h} - h_0\|_2 \le O\left(\tau \delta_n + (\tau + 1)\|h_* - h_0\|_2\right)$$

Choosing $h_* = \arg\min_{h \in \mathcal{H} : \|h\|_{\mathcal{H}} \le B} \|h_* - h_0\|$, yields the bound:

$$\|\hat{h} - h_0\|_2 \le O\left(\tau \delta_n + (\tau + 1) \inf_{h \in \mathcal{H} : \|h\|_{\mathcal{H}}} \|h - h_0\|_2\right)$$

Subsequently one can appropriately choose $\mathcal{H}$ and $B$ so as to trade-off the ill-posedness constant and the bias term.

Moreover, we show that when we have a bounded ill-posedness measure, then we can prove a more convenient version of Theorem 1, that only requires bounds on the critical radius of the centered function classes $\mathrm{star}(\mathcal{H}_B - h_*) = \{r(h - h_*) : h \in \mathcal{H}_B, r \in [0,1]\}$ and $\mathrm{star}(T(\mathcal{H} - h_*)) = \{T(h - h_*) : h \in \mathcal{H}_B, r \in [0,1]\}$, as opposed to the space $\mathcal{G}$ that contains products of these functions.

**Theorem 6.** *Let $\mathcal{F}$ be a symmetric and star-convex set of test functions and consider the estimator in Equation* (4). *Let $h_0$ be any hypothesis (not necessarily in $\mathcal{H}$) that satisfies the Conditional Moment* (1) *and suppose that $\mathcal{H}$ satisfies that:*

$$\inf_{h \in \mathcal{H}} \|h - h_0\|_2 \leq \epsilon_n$$

*and let $h_* = \arg\inf_{h \in \mathcal{H}} \|h - h_0\|_2$. Moreover, suppose that:*

$$\forall h \in \mathcal{H} : \min_{f \in \mathcal{F}_{L^2 \|h - h_*\|_{\mathcal{H}}^2}} \|f - T(h - h_*)\|_2 \leq \eta_n$$

*Assume that functions in $\mathcal{H}_B$ and $\mathcal{F}_{3U}$ have uniformly bounded ranges in $[-1, 1]$ and that:*

$$\delta := \delta_n + \eta_n + \epsilon_n + c_0 \sqrt{\frac{\log(c_1/\zeta)}{n}}$$

*for universal constants $c_0, c_1$, and $\delta_n$ an upper bound on the critical radii of the classes $\mathcal{F}_{3U}$ and*

$$\mathrm{star}(\mathcal{H}_B - h_*) := \{r(h - h_*) : h - h_* \in \mathcal{H}_B, r \in [0,1]\}$$
$$\mathrm{star}(T(\mathcal{H}_B - h_*)) := \{r f_h : h - h_* \in \mathcal{H}_B, r \in [0,1]\}$$

*where $f_h = \arg\min_{f \in \mathcal{F}_U} \|f - T(h - h_*)\|_2$. If $O(\delta^2) \geq \lambda \geq \delta^2/U$ and $O(\delta^2) \geq \mu \geq 2\lambda(4L^2 + 27U/B)$, then $\hat{h}$ satisfies w.p. $1 - 3\zeta$:*

$$\|h - h_0\|_2 = O\left(\tau^2 \delta \max\{1, \|h_*\|_{\mathcal{H}}\}\right)$$

### C.5 Minimax Optimality of Estimation Rate

In this section we take the viewpoint of establishing minimax optimal rates for the estimation problem of interest and discuss under which circumstances the upper bound we provide will typically be tight (i.e. achieving the statistically best possible projected RMSE). Suppose that the only prior assumptions we are willing to make about our data generating process is that it satisfies the moment condition, that $h_0 \in \mathcal{H}$ and that $T_0 \in \mathcal{T}$ for some function class $\mathcal{H}$ and linear operator class $\mathcal{T}$. Moreover, let $\mathcal{F} := \{Th : T \in \mathcal{T}, h \in \mathcal{H}\}$. What is the minimax estimation rate, with respect to the projected MSE norm, achievable in this setting? More concretely, let $D(h, T)$ be any distribution consistent with function $h$, linear operator $T$ and conditional moment condition $Th = E[y \mid z]$. Then for any estimator $\hat{h}$, that takes as input a training sample $S$ of size $n$, drawn i.i.d. from $D(h, T)$, and returns a function $\hat{h}_S$, we want to lower bound the minimax optimal rate:

$$\min_{\hat{h}} \max_{h_0 \in \mathcal{H}, T_0 \in \mathcal{T}} \mathbb{E}_{S \sim D(h_0, T_0)^n} \left[\|T_0(\hat{h}_S - h_0)\|_2^2\right]$$

If the space $\mathcal{T}$ contains the identity, then this is lower bounded by the RMSE rates of a non-parametric regression problem over hypothesis space $\mathcal{H}$. Thus by standard results on regression problems, the critical radius of $\mathcal{H}$ is insurmountable for many classes $\mathcal{H}$ of interest (see e.g. Massart [2000], Bartlett et al. [2005], Rakhlin et al. [2017].

Moreover, suppose that there exists a $T \in \mathcal{T}$ such that: for all $f$ there exists $h \in \mathcal{H}$, such that $Th = f$, i.e. $T$ is the worst mapping that allows one to span all of $\mathcal{F}$. Then even if we knew $T = T_0$, we could not bypass the critical radius of $\mathcal{F}$ for many classes $\mathcal{F}$ of interest (see e.g. Bartlett et al. [2005], Rakhlin et al. [2017]). More generally, we can lower bound the minimax risk as:

$$\max_{T_0 \in \mathcal{T}} \min_{\hat{h}} \max_{h_0 \in \mathcal{H}} \mathbb{E}_{S \sim D(h_0, T_0)^n} \left[\|T_0(\hat{h}_S - h_0)\|_2^2\right]$$

Let $\mathcal{F}_T = \{Th : h \in \mathcal{H}\}$. Then the above can be re-written:

$$\max_{T_0 \in \mathcal{T}} \min_{\hat{f} \in \mathcal{F}_T} \max_{f_0 \in \mathcal{F}} \mathbb{E}_{S \sim D(f_0)^n} \left[ \|\hat{f}_S - f_0\|_2^2 \right]$$

where $D(f_0)$ is any distribution that satisfies $\mathbb{E}[y \mid z] = f_0$. This is the minimax lower bound for the regression problem of predicting $y$ from $z$, assuming that $\mathbb{E}[y \mid z] \in \mathcal{F}_T$. Thus we have that the minimax rate is at least $\max_T \delta(\mathcal{F}_T)$. If we knew that there was a finite set of $k$ representative linear operators $T_1, \ldots, T_k$ in $\mathcal{T}$, such that $\mathcal{F} = \mathcal{F}_{T_1} \cup \ldots \cup \mathcal{F}_{T_k}$, then observe that the critical radius of $\mathcal{F}$ is at most $O(\log(k))$ more than the maximum critical radius of each of the $\mathcal{F}_{T_i}$. Thus the only case that remains open where our upper bound might not be providing tight results is when there is not such finite small set of representative operators in $\mathcal{T}$. In many of our settings, we will have that $\delta(\mathcal{F}) \sim \delta(\mathcal{H})$, which is achieved for the single identity operator $\mathcal{T} = I$. The case where our upper bound is loose, is essentially the case when knowing the operator, or some equivalence class of the operator, can significantly reduce the sample complexity of the problem. Potentially in such settings fitting a first stage model of $T$ to identify the equivalence class or a finite number of viable equivalence classes and focus only on a remaining set of $k$ candidate $\mathcal{F}_{T_1} \cup \ldots \cup \mathcal{F}_{T_i}$ in a second stage can be beneficial. However, in most of our applications this setting does not arise. One for instance can follow techniques similar to aggregation algorithms Rakhlin et al. [2017], that applies our minimax estimator on an $\epsilon$ partition of the original hypothesis $\mathcal{H}$ and then aggregates the resulting winning hypothesis from each partition. However, this would typically be a computationally inefficient algorithm.

# D   Application: Growing Linear Sieves

Consider the case where $\mathcal{H}$ and $\mathcal{F}$ are growing linear sieves, i.e.

$$\mathcal{H} = \mathcal{H}_n := \left\{ \langle \theta, \phi_n(\cdot) \rangle : \theta \in \mathbb{R}^{k_n} \right\},$$
$$\mathcal{F} = \mathcal{F}_n := \left\{ \langle \beta, \psi_n(\cdot) \rangle : \beta \in \mathbb{R}^{m_n} \right\},$$

equipped with norms $\|\langle \theta, \phi_n(\cdot) \rangle\|_{\mathcal{H}} = \|\theta\|_2$, $\|\langle \beta, \psi_n(\cdot) \rangle\|_{\mathcal{F}} = \|\beta\|_2$, for some known and growing feature maps $\phi_n(\cdot), \psi_n(\cdot)$.

Moreover, we denote with $\eta_n$ the approximation error of the sieve $\psi_n$ that is used for the test function space, i.e. for all $h, h_* \in \mathcal{H}$:

$$\inf_{f \in \mathcal{F}} \|f - T(h - h_*)\|_2 \le \eta_n$$

and, let $\epsilon_n$ the approximation error of the sieve $\phi_n$ used for the model, i.e.:

$$\inf_{h \in \mathcal{H}} \|h - h_0\|_2 \le \epsilon_n$$

In that case, applying Theorem 1 with $h_* = \arg\inf_{h \in \mathcal{H}} \|h - h_0\|_2$, gives a bound w.p. $1 - \zeta$ of:

$$\|T(\hat{h} - h_0)\|_2 \le O\left( \left( \delta_n + \epsilon_n + \sqrt{\frac{\log(1/\zeta)}{n}} \right) \max\{1, \|\theta_*\|_2^2\} + \eta_n \right)$$

where $\theta_*$ is the $\ell_2$ norm of the parameter that corresponds to $h_*$.

Moreover, $\delta_n$ is a bound on the critical radius of $\mathcal{F}_U$ and $\mathcal{G}_{B,U}$. Since both are finite dimensional linear functions, via standard covering arguments (see Corollary 5), we can bound $\delta_n = O\left( \sqrt{\frac{\max\{k_n, m_n\} \log(n)}{n}} \right)$. We also now provide a more intricate argument that removes the $\log(n)$ from this rate. Observe that $\mathcal{F}_U$ is a simple linear model space and therefore existing results directly apply to show that the critical radius of $\mathcal{F}_U$ is at most $\sqrt{\frac{m_n}{n}}$ (see e.g. Example 13.5 of Wainwright [2019]). The function space $\mathcal{G}_{B,U}$ is a bit more subtle. We will in fact bound the critical radius of the following larger class:

$$\tilde{\mathcal{G}}_{B,U} = \{(x, z) \to \langle \theta - \theta_*, \phi_n(x) \rangle \langle \beta, \psi_n(z) \rangle : \theta \in \mathbb{R}^{k_n}, \beta \in \mathbb{R}^{m_n}, \|\theta - \theta_*\|_2 \le B, \|\beta\|_2 \le U\}$$

We will use the empirical covering integral bound on the critical radius, presented in Equation (9). Thus we need to bound the metric entropy of the function class $\tilde{\mathcal{G}}_{B,U}(\delta) = \{g \in \tilde{\mathcal{G}}_{B,U} : \|g\|_{2,n} \le \delta\}$.

Let $\Psi_n$ denote the $n \times k_n$ matrix whose $i$-th row corresponds to the vector $\psi_n(x_i)$ and similarly $\Phi_n$. Observe that the norm empirical $\ell_{2,n}$ norm can then be written as:

$$\|\langle \theta - \theta_*, \phi_n(\cdot)\rangle\langle \beta, \psi_n(\cdot)\rangle\|_{2,n} = \frac{\|\Psi_n(\theta - \theta_*)\|_2 \|\Phi_n \beta\|_2}{\sqrt{n}}$$

Thus $\ell_{2,n}$ defines a norm on the space defined by the Hadamard (coordinate-wise) product $v_1 \circ v_2$ of two vectors $v_1, v_2$ in range($\Psi_n$) and range($\Phi_n$), correspondingly, i.e. $\|v_1 \circ v_2\| = \frac{\|v_1\|_2 \|v_2\|_2}{\sqrt{n}}$. Moreover, $\tilde{\mathcal{G}}_{B,U}(\delta)$ is isomorphic to a $\delta$-ball in this space. Moreover, observe that the dimension of the space $\{v_1 \circ v_2 : v_1 \in \text{range}(\Psi_n), v_2 \in \text{range}(\Phi_n)\}$ is at most $\text{rank}(\Psi_n)\,\text{rank}(\Phi_n) \leq k_n \cdot m_n$. Therefore by the volumetric argument presented in Example 5.4 of Wainwright [2019], we get that for any set of samples $S$ of size $n$, $\log(H(\epsilon, \tilde{\mathcal{G}}_{B,U}(\delta), S) \leq k_n m_n \log\left(1 + \frac{2\delta}{\epsilon}\right)$. Moreover, observe that:

$$\int_0^\delta \log(H(\epsilon, \tilde{\mathcal{G}}_{B,U}(\delta), S)d\epsilon \leq \sqrt{\frac{k_n m_n}{n}} \int_0^\delta \sqrt{\log\left(1 + \frac{2\delta}{\epsilon}\right)} d\epsilon$$

$$\leq \delta\sqrt{\frac{k_n m_n}{n}} \int_0^1 \sqrt{\log\left(1 + \frac{2}{u}\right)} du = c\,\delta\sqrt{\frac{k_n m_n}{n}}$$

for some constant $c$. Thus Equation (9) is satisfied for $\delta = O\left(\sqrt{\frac{k_n m_n}{n}}\right)$. Combining all these we get a projected MSE rate w.p. $1 - \zeta$ of:

$$\|T(\hat{h} - h_0)\|_2 = O\left(\left(\sqrt{\frac{k_n m_n}{n}} + \eta_n + \epsilon_n + \sqrt{\frac{\log(1/\zeta)}{n}}\right)\max\{1, \|\theta_*\|_2^2\}\right)$$

Invoking standard bounds on the approximation error of classical sieves (e.g. wavelets) and optimally balancing $k_n, m_n$, yields concrete rates (see e.g. Chen and Pouzo [2012] for particular approximation rates of known sieves).

Combined with ill-posedness conditions provided in [Chen and Pouzo, 2012], our results can thus give an alternative proof to the results in [Chen and Pouzo, 2012] that i) do not make minimum eigenvalue conditions, ii) provide adaptivity to $\|\theta_*\|_2$, without knowledge of it, thereby justifying theoretically the use of the regularization term $R(h)$, that was mostly proposed for experimental improvement in [Chen and Pouzo, 2012]. For instance, one concrete ill-posedness condition is that $\lambda_{\min}\left(\mathbb{E}\left[\mathbb{E}[\phi_n(x) \mid z]\mathbb{E}[\phi_n(x) \mid z]^\top\right]\right) \geq \gamma_n$ and $\lambda_{\max}\left(\mathbb{E}\left[\psi_n(x)\psi_n(x)^\top\right]\right) \leq \sigma_n$. Then the ill-posedness constant is upper bounded by $\tau_n = \sigma_n/\gamma_n$. Moreover, if one assumes a bound on ill-posedness, then Theorem 6 requires $\delta$ to be an upper bound of simpler function spaces, that all correspond to simple linear function spaces in finite dimensions. Thus a smaller bound of $O\left(\sqrt{\frac{\max\{k_n, m_n\}}{n}}\right)$, suffices, leading to an error w.p. $1 - \zeta$ of the form:

$$\|\hat{h} - h_0\|_2 = O\left(\tau_n^2 \left(\sqrt{\frac{\max\{k_n, m_n\}}{n}} + \eta_n + \epsilon_n + \sqrt{\frac{\log(1/\zeta)}{n}}\right)\max\{1, \|\theta_*\|_2^2\}\right)$$

# E   Application: Reproducing Kernel Hilbert Spaces

In this section we deal with the case where $h_0$ lies in a reproducing kernel Hilbert space (RKHS) $\mathbb{H}_{K_\mathcal{H}}$ with kernel $K_\mathcal{H} : \mathcal{X} \times \mathcal{X} \to \mathbb{R}$ and $Th_0$ lies in another RKHS $\mathbb{H}_{K_\mathcal{F}}$ with kernel $K_\mathcal{F} : \mathcal{Z} \times \mathcal{Z} \to \mathbb{R}$. We present the three components required to apply our general theory.

First we characterize the set of test functions that are sufficient to satisfy the requirement that $T(h - h_0) \in \mathcal{F}_U$; *under non-parametric assumptions on the conditional density $p(x \mid z)$ then we can have $K_\mathcal{H} = K_\mathcal{F}$*. Second, by recent results in statistical learning theory, the critical radius of the function classes $\mathcal{F}$ and $\mathcal{G}$ can be characterized as a function of the *eigendecay of the kernel $K$ and the product kernel $K_\times((x,z),(x',z')) = K(x,x') \cdot K(z,z')$ and in the worst-case is of the order of $n^{-1/4}$*. Combining these two facts, we can then apply Theorem 1, to get a bound on the estimation error of the minimax or regularized minimax estimator. Finally, we show that for this set of test functions and hypothesis spaces, *the empirical min-max optimization problem can be solved in closed form*; in particular the inner maximization problem can be shown to correspond roughly to a regularized version of a pairwise metric of the form: $\sum_{i,j} \psi_i K(z_i, z_j)\psi_j$, where $\psi_i = \psi(y_i; h(x_i))$.

### E.1 Characterization of Sufficient Test Functions

In general, it suffices to assume that the linear operator $T$ is regular enough that it satisfies that for any $h \in \mathcal{H}$, we have that $Th \in \mathbb{H}_{K_{\mathcal{F}}}$ for some known kernel $K_{\mathcal{F}}$ and that it is an $L$-Lipschitz operator with respect to the pair of RKHS norms $\|\cdot\|_{\mathcal{H}}, \|\cdot\|_{K_{\mathcal{F}}}$. Then observe that we satisfy the requirement that $T(h - h_*) \in \mathcal{F}_{L^2 \|h - h_*\|_{\mathcal{H}}^2}$, if we take $\mathcal{F} = \mathbb{H}_{K_{\mathcal{F}}}$. We now present two complementary sets of sufficient conditions for which the aforementioned property holds.

The first set of conditions applies to a generic function class $\mathcal{H}$ and asks principally that $p(x|\cdot)$ belongs to a common RKHS for each $x$.

**Lemma 7.** *Suppose that, for each $x$, $p(x|\cdot)$ is an element of an RKHS $\mathbb{H}_{K_{\mathcal{F}}}$ and $h \in \mathcal{H}$ satisfies $|h(x)| \leq \kappa(x)\|h\|_{\mathcal{H}}$ for some $\kappa : \mathcal{X} \to \mathbb{R}$. If $L \triangleq \int \kappa(x)\|p(x|\cdot)\|_{K_{\mathcal{F}}} dx < \infty$, then $Th \in \mathbb{H}_{K_{\mathcal{F}}}$ with $\|Th\|_{K_{\mathcal{F}}} \leq L\|h\|_{\mathcal{H}}$.*

*Proof.* For any nonnegative $h$, Jensen's inequality implies that

$$\|Th\|_{K_{\mathcal{F}}} = \|\int h(x)p(x|\cdot)dx\|_K \leq \int |h(x)|\|p(x|\cdot)\|_{K_{\mathcal{F}}} dx. \tag{10}$$

The same result (10) holds for arbitrary signed $h$ due to the decomposition $h = h_+ - h_-$ for $h_+(x) = \max(h(x), 0)$ and $h_-(x) = \max(-h(x), 0)$, the identity $|h(x)| = |h_+(x)| + |h_-(x)|$, and the triangle inequality $\|Th\|_{K_{\mathcal{F}}} \leq \|Th_+\|_{K_{\mathcal{F}}} + \|Th_-\|_{K_{\mathcal{F}}}$.

Now consider any $h \in \mathcal{H}$ satisfying $|h(x)| \leq \kappa(x)\|h\|_{\mathcal{H}}$ for some $\kappa : \mathcal{X} \to \mathbb{R}$ By our inequality (10), we have

$$\|Th\|_{K_{\mathcal{F}}} \leq \|h\|_{\mathcal{H}} \int \kappa(x)\|p(x|\cdot)\|_{K_{\mathcal{F}}} dx = L\|h\|_{\mathcal{H}}.$$

$\square$

The second set of conditions applies when $h$ belongs to a translation-invariant RKHS and ensures that $Th$ belongs to the same RKHS. Suppose that the kernel $K_{\mathcal{H}}(x, y) = k(x - y)$. Moreover, suppose that $p(x \mid z) = \rho(x - z)$. Then the following lemma states that $Th \in \mathbb{H}_{K_{\mathcal{H}}}$ and hence also $T(h - h_*) \in \mathbb{H}_{K_{\mathcal{H}}}$ for any $h, h_* \in \mathbb{H}_{K_{\mathcal{H}}}$.

**Lemma 8.** *Suppose the conditional distribution of $X$ given $Z = z$ has continuous density $p(x|z) = \rho(x - z)$ and that $K_{\mathcal{H}}(x, y) = k(x - y)$ for $k$ positive definite and continuous. If the generalized Fourier transform of $k$ is continuous on $\mathbb{R}^d \backslash \{0\}$, then $Th \in \mathbb{H}_{K_{\mathcal{H}}}$ for all $h \in \mathbb{H}_{K_{\mathcal{H}}}$ with $\|Th\|_{K_{\mathcal{H}}} \leq L\|h\|_{K_{\mathcal{H}}}$ for $L = \|\hat{\rho}\|_{\infty}$.*

*Proof.* Fix any $h \in H_K$. By [Wendland, 2004, Thm. 10.21], $\|h\|_{K_{\mathcal{H}}} = \|\hat{h}/\sqrt{\hat{k}}\|_2 < \infty$. Moreover, since $\rho$ is in $L^1$, the Hausdorff-Young inequality implies that $\hat{\rho} \in L^{\infty}$. Hence, since $Th = h * \rho$,

$$\|Th\|_{K_{\mathcal{H}}}^2 = \int \widehat{Th}(\omega)^2/\hat{k}(\omega)d\omega = \int \hat{h}(\omega)^2\hat{\rho}(\omega)^2/\hat{k}(\omega)d\omega \leq \|\hat{\rho}\|_{\infty}^2\|\hat{h}/\sqrt{\hat{k}}\|_2^2 = L^2\|h\|_{K_{\mathcal{H}}}^2 < \infty,$$

so that $Th \in \mathbb{H}_{K_{\mathcal{H}}}$ by [Wendland, 2004, Thm. 10.21]. $\square$

Thus in Theorem 1 we can use $\mathcal{H} = \mathcal{F} = \mathbb{H}_K$ for $K = K_{\mathcal{H}}$. Moreover, we can set $B$ to be an upper bound on the squared RKSH norm of $h_0$, i.e. $\|h_0\|_{\mathcal{H}}^2 \leq B$ so that we can take $h_* = h_0$ and have $\|T(h_* - h_0)\|_2 = 0$, i.e. zero bias. Moreover, by Lemma 8 we also know that $\|Th_0\|_{\mathcal{F}}^2 \leq LB$ for some constant $L$. Thus we can set $U = 2LB$ in Theorem 1 and have that Equation (5) holds with $\eta_n = 0$. Thus by Theorem 1, we can get that the estimator in Equation (4) satisfies w.p. $1 - 3\zeta$:

$$\|T(\hat{h} - h_0)\|_2 \leq \delta_n + c_0\sqrt{\frac{\log(c_1/\delta)}{n}}$$

where $\delta_n$ is an upper bound on the critical radii of $\mathcal{F}_{6LB}$ and $\mathcal{G}_B$, which simplify to:

$$\mathcal{F}_{3U} := \left\{ f \in \mathbb{H}_K : \|f\|_K^2 \leq 6LB \right\}$$

$$\mathcal{G}_B := \left\{ (x, z) \to (h(x) - h_0(x)) \, T(h - h_0)(z) : h \in \mathbb{H}_K, \|h - h_0\|_K^2 \leq B \right\}$$

Similar rates can also be established for the regularized estimator analogue in Theorem 1, without explicit knowledge of $B$.

## E.2 Critical Radius of $\mathcal{F}_{3U}$ and $\mathcal{G}_B$

We now turn to analyze the critical radii of $\mathcal{F}_{3U}$ and $\mathcal{G}_B$. We first show that these function spaces are also RKHS with appropriate kernels and have bounded RKHS norms. This is trivial for $\mathcal{F}_U$. Moreover, observe that the space $\mathcal{G}_B$, contains the product of two functions $hf$, where $h : \mathcal{X} \to [-1, 1]$ and $f : \mathcal{Z} \to [-1, 1]$ and such that $h \in \mathcal{H}$ and $f = Th \in \mathcal{F}$. Thus the space $\mathcal{G}$, with inner product $\langle hf, h'f' \rangle_{\mathcal{G}} = \langle h, h' \rangle_{\mathcal{H}} \langle f, f' \rangle_{\mathcal{F}}$, also admits a reproducing kernel, defined as (see Proposition 12.2 of Wainwright [2019]):

$$K_{\mathcal{G}}((x; z), (x'; z')) = K_{\mathcal{H}}(x, x') \, K_{\mathcal{F}}(z, z')$$

Moreover, $\|hf\|_{\mathcal{G}} = \|h\|_{\mathcal{H}} \|f\|_{\mathcal{F}}$. Thus if $h$, satisfies $\|h\|_{\mathcal{H}}^2 \leq B$, then by Lemma 8, $\|Th\|_{\mathcal{F}}^2 \leq L\|h\|_K^2 \leq LB$ for some constant $L$ and $\|hf\|_{\mathcal{G}}^2 \leq LB^2$.

Assuming that the RKHS spaces $\mathcal{F}$ and $\mathcal{G}$, also have a sufficiently fast eigendecay then existing results in statistical learning theory also bound the generalization error Wainwright [2019]. In particular, Corollary 14.2 of Wainwright [2019], shows that for any RKHS $\mathbb{H}_K$, if we let

$$\mathbb{H}_K^B := \{h \in \mathbb{H}_K : \|h\|_K \leq B\},$$

then we can bound the localized Rademacher and empirical Rademacher complexity as:

$$\mathcal{R}(\delta; \mathbb{H}_K^B) \leq B\sqrt{\frac{2}{n}}\sqrt{\sum_{j=1}^{\infty} \min\{\lambda_j, \delta^2\}} \qquad \mathcal{R}_S(\delta; \mathbb{H}_K^B) \leq B\sqrt{\frac{2}{n}}\sqrt{\sum_{j=1}^{n} \min\{\lambda_j^S, \delta^2\}}$$

where $\lambda_j$ are the eigenvalues of the kernel and $\lambda_j^S$ are the empirical eigenvalues of the empirical kernel matrix $\mathbf{K}$ defined as $\mathbf{K}_{ij} = K(x_i, x_j)/n$. Moreover, the unrestricted Rademacher complexity is upper bounded as (see Lemma 26.10 of Shalev-Shwartz and Ben-David [2014]):

$$\mathcal{R}(\mathbb{H}_K^B) \leq O\left(B\sqrt{\frac{\max_{x \in \mathcal{X}} K(x, x)}{n}}\right)$$

Thus in the worst case we can take $\delta_n = O\left(\sqrt{B}\left(\frac{\max_{x \in \mathcal{X}} K(x,x)}{n}\right)^{1/4}\right)$, to get a non-parametric rate of convergence.[4] However, for many kernels, the eigendecay will be sufficiently fast, that $\delta^2$ will not be binding in the minimum. For instance, for the Gaussian kernel in one dimension on the domain $[0, 1]$, with bandwidth of 1, i.e. $K(x, x') = e^{-\frac{(x-x')^2}{2}}$, we have that $\delta_n = O\left(B\sqrt{\frac{\log(n+1)}{n}}\right)$ (see Example 14.4 of Wainwright [2019]).

**Data-adaptive estimation**  Moreover, by Equation (8), we can choose $\delta$ in Theorem 1 based on the empirical critical radius. Observe that the empirical eigenvalues are directly computable from the data and hence, we can calculate a data-adaptive quantity $\hat{\delta}_n$ and choose $\delta$ in Theorem 1, based on this data-adaptive quantity plus an $O\left(\sqrt{\frac{\log(1/\zeta)}{n}}\right)$ term. Moreover, if we use the regularized estimator, then we also do not require knowledge of $B$, which leads to a very data-adaptive estimation scheme. The only thing required is knowledge of an upper bound on the Lipschitz constant $L$ of the operator $T$ with respect to the RKHS norm.

## E.3 Closed-Form Solution to Optimization Problem

Finally, we show that the optimization problem that defines the estimator in Equation (4) can be computed in closed form. We present the results for the constrained estimator, but exact analogues also hold for the regularized version. The proof can be found in Appendix L.1.

**Proposition 9** (Closed-form maximization). *Suppose $\mathcal{F}$ is an RKHS with kernel $K$ equipped with the canonical RKHS norm $\|\cdot\|_{\mathcal{F}} = \|\cdot\|_K$. Then for any $h$*

$$\sup_{f \in \mathcal{F}} \Psi_n(h,f)^2 - \lambda \left( \|f\|_{K_{\mathcal{F}}}^2 + \frac{U}{n\delta^2} \|f\|_{2,n}^2 \right) = \frac{1}{4\lambda} \psi_n^\top K_n^{1/2} (\tfrac{U}{n\delta^2} K_n + I)^{-1} K_n^{1/2} \psi_n \quad (11)$$

$$= \frac{1}{4\lambda} \psi_n^\top K_n \, (\tfrac{U}{n\delta^2} K_n + I)^{-1} \psi_n$$

*where $K_n = (K(z_i, z_j))_{i,j=1}^n$ is the empirical kernel matrix and $\psi_n = (\frac{1}{n} \psi(y_i \,;\, h(x_i)))_{i=1}^n$.*

We note that if we did not enforce the extra $\ell_{2,n}$ norm constraint on $f$ (i.e. $\delta \to \infty$, then the above inner optimization problem simplifies to:

$$\sup_{f \in \mathcal{F}} \Psi_n(h,f)^2 - \lambda \|f\|_{K_{\mathcal{F}}}^2 = \frac{1}{4\lambda} \psi_n^\top K_n \psi_n = \frac{1}{4\lambda n^2} \sum_{i,j} \psi(y_i; h(x_i)) K(z_i, z_j) \psi(y_j; h(x_i)) \quad (12)$$

i.e. we get a pair-wise residual loss, weighted by a kernel matrix that is only a function of the conditioning set $z$.

Thus the solution $\hat{h}$ of the estimator in Equation (4) is equivalent to:

$$\hat{h} = \arg\min_{h \in \mathcal{H}} \frac{1}{4\lambda} \psi_n^\top M \psi_n + \mu \|h\|_{\mathcal{H}}^2 = \arg\min_{h \in \mathcal{H}} \psi_n^\top M \psi_n + 4\mu\,\lambda \|h\|_{\mathcal{H}}^2$$

where $M := K_n^{1/2} (\frac{U}{n\delta^2} K_n + I)^{-1} K_n^{1/2}$. Finally, we show that this outer maximization also has a closed form solution. See Appendix L.2 for the proof.

**Proposition 10** (Closed-form minimization). *Suppose that $\mathcal{H}$ and $\mathcal{F}$ are the RKHSes of the kernels $K_{\mathcal{H}}$ and $K_{\mathcal{F}}$, equipped with the canonical RKHS norms $\|\cdot\|_{\mathcal{H}} = \|\cdot\|_{K_{\mathcal{H}}}$ and $\|\cdot\|_{\mathcal{F}} = \|\cdot\|_{K_{\mathcal{F}}}$. Define the empirical kernel matrices $K_{\mathcal{H},n} = (K_{\mathcal{H}}(x_i, x_j))_{i,j=1}^n$ and $K_{\mathcal{F},n} = (K_{\mathcal{F}}(z_i, z_j))_{i,j=1}^n$. Then the following estimator is an optimizer of Equation (4):*

$$\hat{h} = \sum_{i=1}^n \alpha_{\lambda_*, i} K_{\mathcal{H}}(x_i, \cdot) \qquad \alpha_\lambda := (K_{\mathcal{H},n} M K_{\mathcal{H},n} + 4\,\lambda\,\mu K_{\mathcal{H},n})^\dagger K_{\mathcal{H},n} M y$$

*for $M = K_{\mathcal{F},n}^{1/2} (\frac{U}{n\delta^2} K_{\mathcal{F},n} + I)^{-1} K_{\mathcal{F},n}^{1/2} \equiv K_{\mathcal{F},n} (\frac{U}{n\delta^2} K_{\mathcal{F},n} + I)^{-1}$ and $A^\dagger$ is the Moore-Penrose pseudoinverse of a matrix $A$.*

**Hyper-parameter tuning** Observe that Theorem 1 states that as long as the regularization strength satisfies that $\lambda\mu = \Theta(\delta^4 L^2)$, then this estimator will provide results that automatically scale with the RKHS norm of true hypothesis $h_0$. Moreover, the regularization hyperparameter $\lambda \cdot \mu$ can also be tuned in practice by evaluating the loss function $\psi_n^\top M \psi_n$ on a left-out sample, with parameters $n, \delta$ set to the appropriate ones for the size of that sample.

**Low-Rank Approximation and Nystrom's Method** The solution to the empirical optimization problem requires inverting an $n \times n$ kernel matrix, which takes time $O(n^3)$. This can be prohibitive for moderate sample sizes of the order of tens of thousands. We note here that one can construct very good approximations to the solution in Proposition 10 by considering low-rank approximations of the kernel matrix $K$. We present here one such low-rank approximation, based on Nystrom's method, but we note that the plethora of recent literature on low-rank kernel approximation methods are applicable to our problem too (see e.g. Kumar et al. [2012], Bach and Jordan [2005], Musco and Musco [2017], Oglic and Gärtner [2017]).

Suppose that we can express our kernel matrices as $K_{\mathcal{H},n}$ and $K_{\mathcal{H},n}$ as $K_{\mathcal{H},n} = DD^\top$ and $K_{\mathcal{H},n} = VV^\top$, where $D$ and $V$ are of dimensions $n \times r$ and such that we can express the *kernel row* of any new test sample as:

$$(K_{\mathcal{H}}(x_1, x), \ldots, K_{\mathcal{H}}(x_n, x)) = V\phi(x)$$

for some $r$-dimensional vector $\phi(x)$. Then we can express $h(x) = \phi(x)^\top V^\top a_\lambda$. If we then define $\gamma = V^\top \alpha_\lambda$. Then we can re-write the closed form solutions to the min and max problems as follows:

$$\sup_{f \in \mathcal{F}} \Psi_n(h,f)^2 - \lambda \left( \|f\|_{K_{\mathcal{F}}}^2 + \frac{U}{n\delta^2} \|f\|_{2,n}^2 \right) = \frac{1}{4\lambda} \psi_n^\top D \left( \frac{U}{n\delta^2} D^\top D + I \right)^{-1} D^\top \psi_n$$

Figure 5: Estimated functions based on our minimax estimator for different true functions. We use an rbf kernel with parameter $\gamma = .1$ and $1000$ samples. We chose critical radius parameter $\delta = 5/n^{.4}$ and the regularization hyper-parameter $\tau$ is chosen via k-fold cross-validation. The data generating process was: $x = .6\,z + .4\,u + \delta$ and $y = h_0(x) + u + \epsilon$ and $z, u \sim N(0, 2)$ and $\epsilon, \delta \sim N(0, .1)$.

and if we let $Q := \left( \frac{U}{n\delta^2} D^\top D + I \right)^{-1}$ and $A = V^\top D$, then:

$$\gamma := \left( AQA^\top + 4\,\lambda\,\mu\,I \right)^{-1} AQD^\top y$$
$$\hat{h}(x) := \phi(x)^\top \gamma$$

Observe that every matrix calculation in the above expressions requires time at most $O(n\,r^2)$ to be computed. Thus if $r \ll n$, we have massively reduced the computation time from $\Theta(n^3)$ to $O(n\,r^2)$, making the method practical even very large data regimes.

Even though $r$ in the worst-case can be of size $n$, we can typically well-approximate the kernel matrices with $r \ll n$. One popular approach for achieving this is Nystrom's method, which essentially sub-samples a set of $r$ points and uses the normalized kernel distances with respect to this subset of points as $D$ and $V$, respectively. In particular, let $S$ denote an $n \times r$ matrix whose $i$-th column contains a 1 in position $j$ for some randomly sampled index $j$. Then $KS$ is an $n \times r$ sub-matrix of $K$, where a subset $S$ of the columns of $K$ are chosen at random.[5] Then we can approximate $K$ via $VV^\top$, where $V = KSM^{1/2}$ and $M = (S^\top KS)^+$ (i.e. $V$ is contains normalized kernel-based similarities to the subset $S$ of $r$ randomly chosen points). Moreover, for any new test point, we can set $\phi(x) = M^{1/2}(K_{\mathcal{H}}(x_i, x))_{i \in S}$.

### E.4 Bounds on Ill-Posedness Measure

The results so far in the section provide bounds on the projected RMSE. In this last section, we show that under further assumptions on the strength of the instrument (i.e. the correlation of $x$ and $z$), then the projected RMSE rates also imply rates for the RMSE. We give an example such set of conditions, mostly as an example of a sufficient set of assumptions that lead to RMSE rates and in order to provide qualitative insights on what RMSE rates one can expect in different regimes of the instrument strength and the eigendecay of the kernel. In this section we will assume that the space $\mathcal{H}$ is also augmented with a hard constraint on the RKHS norm, i.e. $\mathcal{H} = \mathbb{H}_K^B = \{h \in \mathbb{H}_K : \|h\|_K \leq B\}$. Assuming $\|h_0\|_K \leq B$ this does not change the statistical guarantees and moreover the closed form optimization theorems, can easily be amended to incorporate a hard constraint on top of the

(a) $\sin(x)$      (b) $1 + 1.5 \cdot 1\{x > 0\}$      (c) $-1.5 \cdot x + .9 \cdot x^2 + x^3$

Figure 6: Estimates based on Nystrom approximation, with 50 nystrom samples, for the same dgp and parameter setup as in Figure 5.

regularization (due to the equivalent between hard constraints and regularization). Imposing this hard constraint will simplify the analysis of this section.[6]

By Mercer's theorem we can express any function in the RKHS $\mathbb{H}_K^B$, in terms of the eigenfunctions of the kernel:

$$h = \sum_{j \in J} a_j e_j$$

with $e_j : \mathcal{X} \to \mathbb{R}$, such that $\mathbb{E}[e_j(x)^2] = 1$ and $\mathbb{E}[e_i(x)\, e_j(x)] = 0$ and $J$ a countable set. Moreover, we have $\|h\|_2^2 = \sum_{j \in J} a_j^2$ and $\|h\|_K = \sum_{j \in J} \frac{a_j^2}{\lambda_j} \leq B$. Thus we have that $\|h\|_{\mathcal{H}}^2 \leq B$ implies that for all $m \in \mathbb{N}_+$: $\sum_{j \geq m} a_j^2 \leq \lambda_m B$. Moreover, we have:

$$\|Th\|_2^2 = \sum_{i,j \in J} a_i a_j \mathbb{E}[\mathbb{E}[e_i(x) \mid z]\mathbb{E}[e_j(x) \mid z]].$$

For any $m \in \mathbb{N}_+$, let $I := \{1, \ldots, m\}$, $e_I = (e_1, \ldots, e_m)$, $a_I = (a_1, \ldots, a_m)$ and:

$$V_m := \mathbb{E}[\mathbb{E}[e_I(x) \mid z]\, \mathbb{E}[e_I(x) \mid z]^\top]$$

and suppose that $\lambda_{\min}(V_m) \geq \tau_m$, i.e. that these finite eigenfunctions maintain some fraction of their independent components, even when they are smoothened through the conditional expectation $p(x \mid z)$. Furthermore suppose that for all $i \leq m < j$: $|\mathbb{E}[\mathbb{E}[e_i(x) \mid z]\mathbb{E}[e_j(x) \mid z]]| \leq \gamma_m \leq c\,\tau_m$ (for some constant $c$), i.e. the smoothening performed by the conditional expectation does not ruin a lot the orthogonality of the first $m$ eigenfunctions with eigenfunctions for indices larger than $m$. Observe that if we had a perfect instrument, i.e. $z$ was perfectly correlated with $x$, then $V_m = I_m$ and $\mathbb{E}[\mathbb{E}[e_i(x) \mid z]\mathbb{E}[e_j(x) \mid z]] = \mathbb{E}[e_i(x)e_j(x)] = 0$. Thus for a perfect instrument $\tau_m = 1$ and $\gamma_m = 0$. Therefore the latter requirements are implicit assumptions on the strength of the instrument.[7] We show that under these assumptions, we can bound the measure of ill-posedness as follows.

**Lemma 11.** *Suppose that* $\lambda_{\min}(V_m) \geq \tau_m$ *and for some constant* $c > 0$, *for all* $i \leq m < j$,

$$|\mathbb{E}[\mathbb{E}[e_i(x) \mid z]\mathbb{E}[e_j(x) \mid z]]| \leq c\,\tau_m$$

*Then:*

$$\tau^*(\delta)^2 := \max_{h \in \mathbb{H}_K^B : \|Th\|_2 \leq \delta} \|h\|_2^2 \leq \min_{m \in \mathbb{N}_+} \left( \frac{4\delta^2}{\tau_m} + (4c^2 + 1)B\lambda_{m+1} \right)$$

The optimal choice of $m_*$ roughly solves the equation: $\tau_m \lambda_{m+1} = \delta^2/B$. If for instance $\lambda_m \leq m^{-b}$ for $b > 1$, and $\tau_m \geq m^{-a}$ for $a > 0$, then: $m_* \sim \delta^{2/(a+b)}$, leading to a rate of:

$$\|\hat{h} - h_*\|_2 = O\left( \delta^{b/(a+b)} \right)$$

We see that the RMSE rate is of a slower order than the projected MSE rate. If $\lambda_m$ has an exponential eigendecay, i.e. $\lambda_m \sim 2^{-m}$ (e.g. such as in the case of a Gaussian kernel), and $\tau_m \geq m^{-a}$, then $m_* \sim \log(1/\delta^2)$ and we get:

$$\|\hat{h} - h_*\|_2 = O\left(\delta\left(\log(1/\delta)\right)^{a/2}\right)$$

Thus we only get a logarithmic increase in the RMSE rate as compared to the Projected RMSE rate. However, we note that if also $\tau_m \sim 2^{-a\,m}$ and $\lambda_m \sim 2^{-b\,m}$, then we get rates of $O\left(\delta^{b/(a+b)}\right)$, by settings $m_* \sim \log(1/\delta^{2/(a+b)})$. Finally, in the severely ill-posed setup, where $\tau_m \sim 2^{-m}$ and $\lambda_m \sim m^{-b}$, then we have $m_* \sim \log(1/\delta^2)$ and:

$$\|\hat{h} - h_*\|_2 = O\left(\frac{1}{\log(1/\delta)^b}\right)$$

leading to a very slow rate of convergence that will typically be of the order of $1/\log(n)$.

Observe that we achieve the rate for the optimal choice of $m$, without the need to tune our algorithm. The RKHS norm penalty implicitly clips the weight that our functions can put on eigenfunctions with large index and hence controls the measure of ill-posedness for whatever is the decay rates of the eigenvalues $\lambda_m$ and $\tau_m$.

# F   Application: High-Dimensional Sparse Linear Function Spaces

In this section we deal with high-dimensional linear function classes, i.e. the case when $\mathcal{X}, \mathcal{Z} \subseteq \mathbb{R}^p$ for $p \gg n$ and $h_0(x) = \langle \theta_0, x \rangle$. We will address the case when the function $\theta_0$ is assumed to be sparse, i.e. $\|\theta_0\|_0 := \{j \in [p] : |\theta_j| > 0\} \leq s$. We will be denoting with $S$ the subset of coordinates of $\theta_0$ that are non-zero and with $S^c$ its complement. For simplicity of exposition we will also assume that $\mathbb{E}[x_i \mid z] = \langle \beta, z \rangle$, though most of the results of this section also extend to the case where $\mathbb{E}[x_i \mid z] \in \mathcal{F}_i$ for some $\mathcal{F}_i$ with small Rademacher complexity. We provide two sets of results, dependent on whether we make further minimum eigenvalue assumptions on the covariance matrix of the random variables $\mathbb{E}[x_i \mid z]$.

## F.1   Hard Sparsity Constraints without Minimum Eigenvalue

In the first result, we apply Theorem 1 to show that even without any further assumptions on the eigenvalues of the covariance matrix

$$V := \mathbb{E}[\mathbb{E}[x \mid z]\mathbb{E}[x \mid z]^\top],$$

we can attain fast rates of the order of $n^{-1/2}$ that are logarithmic in $p$ and only linear in the sparsity $s$ of $h_0$ and the sparsity $r$ of the conditional expectation functions $\mathbb{E}[x_i \mid z]$. Albeit the optimization problem we need to solve to get these rates is non-convex and has running time that is exponential in $r, s$. This setting covers and extends the linear moment case of the setting analyzed in [Fan and Liao, 2014]; albeit we only provide RMSE and projected RMSE rates.

**Corollary 12.** *Suppose that $h_0(x) = \langle \theta_0, x \rangle$ with $\|\theta_0\|_0 \leq s$ and $\mathbb{E}[x_i \mid z] = \langle \beta_0^i, z \rangle$ with $\|\beta_0^i\|_0 \leq r$. Then let $\mathcal{H}$ consist of all $s$-sparse linear functions of $x$ and $\mathcal{F}$ consist of all $(s \cdot r)$-sparse linear functions of $z$ with coefficients in $[-1, 1]$. in $p$ dimensions with only $s$ non-zero coefficients and $\mathcal{F}$ consists of linear functions in $q$ dimensions with $r$ non-zero coefficients. Then the estimator presented in Equation (4), satisfies that w.p. $1 - \zeta$:*

$$\|T(\hat{h} - h_0)\|_2 \leq O\left(\sqrt{\frac{r\,s\log(p\,n)}{n}} + \sqrt{\frac{\log(1/\zeta)}{n}}\right)$$

The proof follows immediately from the fact that the metric entropy of $r\,s$-sparse linear functions in $p$-dimensions, with coefficients in $[-1, 1]$ is of the order of $O\left(r\,s\log(p/\epsilon)\right)$. Thus we can invoke Corollary 5 to get a bound of $O\left(\sqrt{\frac{r\,s\,\log(p\,n)}{n}}\right)$ on the critical radii of classes $\mathcal{F}_{3U}$ and $\mathcal{G}_{B,U}$ and apply Theorem 1.

## F.2 $\ell_1$-Relaxation under Minimum Eigenvalue Condition

In the second set of results we assume a restricted minimum eigenvalue of $\gamma$ on the matrix $V$ and apply Theorem 2 to get fast rates of the order of $n^{-1/2}$, that also scale logarithmically in $p$, linearly in $r, s$ and $\gamma^{-1}$. Moreover, the optimization problem required is now a convex problem as we replace the hard sparsity constraint with an $\ell_1$ constraint. This dichotomy of computationally efficient vs computationally hard estimation dependent on whether we make minimum eigenvalue assumptions is a well established result in exogenous regression problems [Zhang et al., 2014] and hence we provide here analogous positive results for the endogenous regression setup. We also note that without the minimum eigenvalue condition, our Theorem 1 still provides slow rates of the order of $n^{-1/4}$, for computationally efficient estimators that replace the hard sparsity constraint with an $\ell_1$-norm constraint. Our results based on the $\ell_1$-constraint are also closely related to the work of Gautier et al. [2011], who analyzes an endogenous analogue of the Dantzig selector. Our work proposes an alternative to the Dantzig selector that enjoys similar estimation rate guarantees.

**Corollary 3.** *Suppose that $h_0(x) = \langle \theta_0, x \rangle$ with $\|\theta_0\|_0 \leq s$ and $\|\theta_0\|_1 \leq B$ and $\|\theta_0\|_\infty \leq 1$. Moreover, suppose that $\mathbb{E}[x_i \mid z] = \langle \beta_0^i, z \rangle$, with $\beta_0^i \in \mathbb{R}^p$ and $\|\beta_0^i\|_1 \leq U$ and that the co-variance matrix $V$ satisfies the following restricted eigenvalue condition:*

$$\forall \nu \in \mathbb{R}^p \text{ s.t. } \|\nu_{S^c}\|_1 \leq \|\nu_S\|_1 + 2\,\delta_{n,\zeta} : \nu^\top V \nu \geq \gamma \|\nu\|_2^2$$

*Then let $\mathcal{H} = \{x \to \langle \theta, x \rangle : \theta \in \mathbb{R}^p\}$, $\|\langle \theta, \cdot \rangle\|_{\mathcal{H}} = \|\theta\|_1$, $\mathcal{F}_U = \{z \to \langle \beta, z \rangle : \beta \in \mathbb{R}^p, \|\beta\|_1 \leq U\}$ and $\|\langle \beta, \cdot \rangle\|_{\mathcal{F}} = \|\beta\|_1$. Then the estimator presented in Equation (7) with $\lambda \leq \frac{\gamma}{8s}$, satisfies that w.p. $1 - \zeta$:*

$$\|T(\hat{h} - h_0)\|_2 \leq O\left( \max\left\{ 1, \frac{1}{\lambda} \frac{\gamma}{s} \right\} \sqrt{\frac{s}{\gamma}} \left( (B + U + 1) \sqrt{\frac{\log(p)}{n}} + \sqrt{\frac{\log(p/\zeta)}{n}} \right) \right)$$

*If instead we assume that $\|\beta_0^i\|_2 \leq U$ and $\sup_{z \in \mathcal{Z}} \|z\|_2 \leq R$ then by setting $\mathcal{F}_U = \{z \to \langle \beta, z \rangle : \|\beta\|_2 \leq U\}$ and $\|\langle \beta, \cdot \rangle\|_{\mathcal{F}} = \|\beta\|_2$, we have:*

$$\|T(\hat{h} - h_0)\|_2 \leq O\left( \max\left\{ 1, \frac{1}{\lambda} \frac{\gamma}{s} \right\} \sqrt{\frac{s}{\gamma}} \left( (B + 1) \sqrt{\frac{\log(p)}{n}} + \frac{U R}{\sqrt{n}} + \sqrt{\frac{\log(p/\zeta)}{n}} \right) \right)$$

**Second order influence from $\mathbb{E}[x_i \mid z]$ model complexity**   Notably, observe that in the case of $\|\beta_0^i\|_2 \leq U$, we note that if one wants to learn the true $\beta$ with respect to the $\ell_2$ norm or the functions $\mathbb{E}[x_i \mid z]$ with respect to the RMSE, then the best rate one can achieve (by standard results for statistical learning with the square loss), even when one assumes that $\sup_{z \in \mathcal{Z}} \|z\|_2 \leq R$ and that $\mathbb{E}[zz^\top]$ has minimum eigenvalue of at least $\gamma$, is: $\min\left\{ \sqrt{\frac{p}{n}}, \left( \frac{U R}{n} \right)^{1/4} \right\}$. For large $p \gg n$ the first rate is vacuous. Thus we see that even though we cannot accurately learn the conditional expectation functions at a $1/\sqrt{n}$ rate, we can still estimate $h_0$ at a $1/\sqrt{n}$ rate, assuming that $h_0$ is sparse. Therefore, the minimax approach offers some form of robustness to nuisance parameters, reminiscent of the type of robustness of Neyman orthogonal methods (see e.g. [Chernozhukov et al., 2018]).

## F.3 Solving the $\ell_1$-Relaxation Optimization Problem via First-Order Methods

The estimator presented in Corollary 3 require solving optimization problems of the form:

$$\min_{\theta : \|\theta\|_1 \leq B} \max_{\beta : \|\beta\| \leq U} \langle \mathbb{E}_n \left[ (y - \langle \theta, x \rangle) z \right], \beta \rangle + \mu \|\theta\|_1 \tag{13}$$

for some $R, \mu$ and for norm $\|\cdot\|$ either $\|\cdot\|_1$ or $\|\cdot\|_2$ (in the constrained estimator $\mu = 0$; while in the regularized $R = \infty$ - though in practice we can set it to some large value for stability of the optimization process). Observe that inner optimization simplifies to:

$$\min_{\theta : \|\theta\|_1 \leq B} \|\mathbb{E}_n \left[ (y - \langle \theta, x \rangle) z \right]\|_* + \frac{\mu}{U} \|\theta\|_1$$

where $\|\cdot\|_*$ is the dual norm of $\|\cdot\|$ (i.e. the $\ell_\infty$ norm in the case where $\|\cdot\|$ is the $\ell_1$ norm and the $\ell_2$ norm in the case where $\|\cdot\|$ is the $\ell_2$ norm). One approach to solving these optimization problems is using projected sub-gradient descent:

$$\beta_t = \underset{\beta:\|\beta\|\leq U}{\arg\max}\langle \mathbb{E}_n\left[(y-\langle\theta_t,x\rangle)z\right],\beta\rangle$$

$$\theta_{t+1} = \Pi\left(\theta_t + \eta\,\mathbb{E}_n\left[x\,z^\top\right]\beta_t - \frac{\mu}{U}\texttt{sign}(\theta_t)\right)$$

$$\Pi(\theta) = \underset{\theta':\|\theta'\|_1\leq B}{\arg\min}\|\theta-\theta'\|_2$$

Moreover, for both $\ell_1$ and $\ell_2$ norm, the solution to $\beta_t$ can be easily found in closed form.[8] After $O(1/\epsilon^2)$ iterations and for $\eta=\Theta(\epsilon)$, we will have that $\bar\theta = \frac{1}{T}\sum_{t=1}^T\theta_t$, is an $\epsilon$-approximate solution to the optimization problem.

**Improved Iteration Complexity with Optimistic FTRL Dynamics** The sub-gradient descent approach has two caveats: i) the rate of $1/\epsilon^2$ is considerably slow and would require a large number of iterations to converge to a reasonable solution, ii) the gradient does not admit an unbiased stochastic version (due to the non-linearity introduced by the $\arg\max$ operation that defines $\beta_t$), and therefore the algorithm does not admit a stochastic variant, which is useful for large samples. We can improve the error rate by invoking algorithms that address non-smooth optimization problems that take the form of a min-max objective of some underlying smooth loss.

First, we show that we can remove the non-smoothness of the $\ell_1$-regularization by lifting the parameter $\theta$ to a $2p$-dimensional positive orthant. Consider two vectors $\rho^+,\rho^-\geq 0$ and then setting $\theta = \rho^+ - \rho^-$, with $\rho = (\rho^+;\rho^-)$ and $\|\rho\|_1\leq B$. Observe that for any feasible $\theta$, the solution $\rho_i^+ = \theta_i 1\{\theta_i > 0\}$ and $\rho_i^- = \theta_i 1\{\theta_i \leq 0\}$ is still feasible and achieves the same objective. Moreover, any solution $\rho$, maps to a feasible solution $\theta$ (since $\|\theta\|_1 \leq \|\rho_+ - \rho_-\|_1 \leq \|\rho^+\|_1 + \|\rho^-\|_1 \leq B$) and thus the two optimization programs have the same optimal solutions. Then, if we define with $v = (x;-x)$, then the optimization problem can be re-stated as:

$$\min_{\rho\geq 0:\|\rho\|_1\leq B}\ \max_{\beta:\|\beta\|\leq U}\ \ell(\rho,\beta)$$

where:

$$\ell(\rho,\beta) := \beta^\top\mathbb{E}_n[zy] - \beta^\top\mathbb{E}_n[zv^\top]\rho + \mu\sum_{i=1}^{2p}\rho_i$$

This falls exactly into the class of problems analyzed in a line of work on bi-linear minimax optimization, starting from the seminal work of Nesterov [2005]. For instance, we can view the problem as a two-player bi-linear zero-sum game and invoke the Optimistic Follow-the-Regularized-Leader (OFTRL) or Optimistic Mirror Descent (OMD) paradigm of Rakhlin and Sridharan [2013], Syrgkanis et al. [2015], to find an $\epsilon$-approximate solution for $\rho$ in $O(1/\epsilon)$ iterations. The algorithm repeats for $T$ iterations the updates:

$$\rho_{t+1} = \underset{\rho\geq 0:\|\rho\|_1\leq B}{\arg\min}\sum_{\tau\leq t}\ell(\rho,\beta_\tau) + \ell(\rho,\beta_t) + \frac{1}{\eta}R_{\min}(\rho)$$

$$\beta_{t+1} = \underset{\beta:\|\beta\|_1\leq U}{\arg\max}\sum_{\tau\leq t}\ell(\rho_\tau,\beta) + \ell(\rho_t,\beta) - \frac{1}{\eta}R_{\max}(\beta)$$

and returns $\bar\rho = \frac{1}{T}\sum_{t=1}^T\rho_t$, $\bar\beta = \frac{1}{T}\sum_{t=1}^T\beta_t$.[9] We note that if we did not double count the last period's loss and we used $R_{\min}(x) = R_{\max}(x) = \frac{1}{2}\|x\|_2^2$, then this would correspond to running

simultaneous gradient descent dynamics for both parameters $\rho, \beta$. Moreover, the parameters $\bar\rho, \bar\beta$ can be thought as primal and dual solutions and we can use the duality gap as a certificate for convergence of the algorithm.[10]

$$\text{tol} = \max_{\beta: \|\beta\| \le U} \ell(\bar\rho, \beta) - \min_{\rho: \|\rho\|_1 \le B} \ell(\rho, \bar\beta)$$

This approach addresses both problems with projected sub-gradient descent: i) as we will show below, the iteration complexity is $O\left((B + U^2)\log(B\,p)/\epsilon\right)$, instead of $1/\epsilon^2$, ii) the per-iteration losses $\ell(\rho, \beta_t), \ell(\rho_t, \beta)$ in the FTRL formulation can be replaced with unbiased estimates, while still maintaining theoretical guarantees and therefore the algorithm admits a stochastic analogue which makes it scalable to very large data sets.[11]

To instantiate this paradigm we need to find appropriate regularizers for the strategy spaces of the two players. Below we outline two concrete such algorithms for the two cases of the norm of $\beta$ and provide worst-case convergence rates.

$\ell_1$-**ball adversary**     For the case when $\|\beta\| = \|\beta\|_1$, we can further simplify the problem by showing that the inner optimization can be performed over a $2p$-dimensional simplex. If we let $u = (z; -z)$, then we can re-write the optimization problem as:

$$\ell(\rho, w) := w^\top \mathbb{E}_n[uy] - w^\top \mathbb{E}_n[uv^\top]\rho + \frac{\mu}{U}\sum_{i=1}^{2p} \rho_i$$

$$\min_{\rho \ge 0: \|\rho\|_1 \le B} \max_{w: \|w\| = 1} \ell(\rho, w)$$

Since both player strategies $\rho, w$ are constrained to be in an $\ell_1$-ball, we can get iteration complexity that only grows logarithmically with the dimension $p$, if for each player we use OFTRL with an entropic regularizer: i.e. $R_{\min}(x) = R_{\max}(x) = \sum_{i=1}^{2p} x_i \log(x_i)$, denotes the negative entropy.

**Proposition 13.** *Consider the algorithm that for $t = 1, \ldots, T$, sets:*

$$\tilde\rho_{i,t+1} = \tilde\rho_{i,t} e^{-2\frac{\eta}{B}\left(-\mathbb{E}_n[v_i u^\top w_t] + \frac{\mu}{U}\right) + \frac{\eta}{B}\left(-\mathbb{E}_n[v_i u^\top w_{t-1}] + \frac{\mu}{U}\right)} \qquad \rho_{t+1} = \tilde\rho_{t+1} \min\left\{1, \frac{B}{\|\tilde\rho_{t+1}\|_1}\right\}$$

$$\tilde w_{i,t+1} = w_{i,t} e^{2\eta\,\mathbb{E}_n[(y-\rho_t^\top v)\,u_i] - \eta\,\mathbb{E}_n[(y-\rho_{t-1}^\top v)\,u_i]} \qquad w_{t+1} = \frac{\tilde w_{t+1}}{\|\tilde w_{t+1}\|_1}$$

*with $\tilde\rho_{i,-1} = \tilde\rho_{i,0} = 1/e$ and $\tilde w_{i,-1} = \tilde w_{i,0} = 1/(2p)$ and returns $\bar\rho = \frac{1}{T}\sum_{t=1}^{T} \rho_t$. Then for $\eta = \frac{1}{4\|\mathbb{E}_n[vu^\top]\|_\infty}$,[12] after*

$$T = 16\|\mathbb{E}_n[vu^\top]\|_\infty \frac{4B^2 \log(B \vee 1) + (B+1)\log(2p)}{\epsilon}$$

*iterations, the parameter $\bar\theta = \bar\rho^+ - \bar\rho^-$ is an $\epsilon$-approximate solution to the minimax problem in Equation (13).*

Moreover, every update step requires computation time $O(\min\{n\,p, p^2\})$.[13] Using techniques for sparse gradient updates, one could also potentially improve the iteration complexity to not depend linearly on the dimension $p$ (see e.g. Langford et al. [2009], Duchi et al. [2008], Duchi and Singer [2009], McMahan [2011]), but we defer such approaches to future work.

(a) true vs. est. $\theta$ ($n = 600$)  (b) true vs. est. $\theta$ ($n = 1000$)  (c) dual variables $w^+ - w^-$

Figure 7: Estimates based on minimax estimator proposed in Proposition 13. The left figure depicts the $p = 2000$ estimated coefficients compared to the true coefficients; we also include the coefficients of i) a direct lasso regression to portray the importance of dealing with the endogeneity problem (Lasso), ii) a two-stage lasso regression where we regress each $x_i$ on $z$ and then regress $y$ on $\mathbb{E}[x \mid z]$, all regressions performed with lasso where the first stage regularization was fixed to $0.01$ and the final stage was chosen via cross-validation (2SLasso), iii) the algorithm in Proposition 13 (SparseIV), iv) a stochastic variant of the algorithm in Proposition 13 where a mini-batch of 10 samples is used at each iteration (StochasticSparseIV). The right pictures depicts the coefficients of the dual test function learned by the adversary at equilibrium, which is of the form: $f(z) = \sum_{i=1}^{p}(w_i^+ - w_i^-)z_i$. The data generating process was: $x, z, u \in \mathbb{R}^p$, $x = z + u$, $y = \langle x + u, \theta \rangle$, $z, u \sim N(0, I_d)$, $\theta = (1, -1, 0, \ldots, 0)$, $p = 2000$.

$\ell_2$**-ball adversary** For the case when $\|\beta\| = \|\beta\|_2$, then we can use $R_{\max}(\beta) = \frac{1}{2}\|\beta\|_2^2$, which leads to an alternative update rule for the maximizing player. In this case, the update of the maximizing player is essentially optimistic gradient descent, modulo the normalization so as to respect the $\ell_2$-norm constraint.

**Proposition 14.** *Consider the algorithm that for $t = 1, \ldots, T$, sets:*

$$\tilde{\rho}_{i,t+1} = \tilde{\rho}_{i,t}e^{-2\frac{\eta}{B}\left(-\mathbb{E}_n[v_iz^\top\beta_t]+\frac{\mu}{U}\right)+\frac{\eta}{B}\left(-\mathbb{E}_n[v_iz^\top\beta_{t-1}]+\frac{\mu}{U}\right)} \quad \rho_{t+1} = \tilde{\rho}_{t+1}\min\left\{1, \frac{B}{\|\tilde{\rho}_{t+1}\|_1}\right\}$$

$$\tilde{\beta}_{t+1} = \tilde{\beta}_{t+1} + 2\eta\mathbb{E}_n[(y - \rho_t^\top v)\, z] - \eta\mathbb{E}_n[(y - \rho_{t-1}^\top v)\, z] \quad \beta_{t+1} = \tilde{\beta}_{t+1}\min\left\{1, \frac{U}{\|\tilde{\beta}_{t+1}\|_2}\right\}$$

*with $\tilde{\rho}_{i,-1} = \tilde{\rho}_{i,0} = 1/e$ and $\tilde{\beta}_{-1} = \tilde{\beta}_0 = 0$. Then for $\eta = \frac{1}{4\|\mathbb{E}_n[zv^\top]\|_{2,\infty}}$,[14] after*

$$T = 16\|\mathbb{E}_n[zv^\top]\|_{2,\infty}\frac{4B^2\log(B \vee 1) + B\log(2p) + U^2/2}{\epsilon}.$$

*iterations, the parameter $\bar{\theta} = \bar{\rho}^+ - \bar{\rho}^-$ is an $\epsilon$-approximate solution to the minimax problem in Equation* (13).

Observe that if $v_j \in [-H, H]$ then the quantity $\|\mathbb{E}_n[zv^\top]\|_{2,\infty}\|$ can be upper bounded by $H\sqrt{\mathbb{E}_n[\|z\|_2^2]}$, which under the assumptions of Corollary 3 is at most a constant.

### F.4 Bounds on Ill-Posedness Measure

Let $h(x) = \langle\theta, x\rangle$, $h_0(x) = \langle\theta_0, x\rangle$ and $\nu = \theta - \theta_0$. Then observe that we have:

$$\|T(h - h_0)\|_2^2 = \nu^\top\mathbb{E}\left[\mathbb{E}[x \mid z]\mathbb{E}[x \mid z]^\top\right]\nu = \nu^\top V\nu \geq \lambda_{\min}(V)\|\nu\|_2^2$$

where we remind that $V := \mathbb{E}\left[\mathbb{E}[x \mid z]\mathbb{E}[x \mid z]^\top\right]$ and $\lambda_{\min}(V)$ denotes the minimum eigenvalue of $V$. Moreover, if we let $\Sigma = \mathbb{E}\left[xx^\top\right]$ then:

$$\|h - h_0\|_2^2 = \nu^\top\mathbb{E}\left[xx^\top\right]\nu \leq \lambda_{\max}(\Sigma)\|\nu\|_2^2$$

Thus we see that the measure of ill-posedness can be upper bounded as:

$$\tau \leq \sqrt{\frac{\lambda_{\max}(\Sigma)}{\lambda_{\min}(V)}}$$

Thus assuming that these eigenvalues are upper and lower bounded correspondingly, then the results of this section extend also to RMSE guarantees for the recovered $\hat{h}$ and not just projected RMSE guarantees, at the cost of an extra multiplicative factor of $\tau$.

Moreover, we note that in both our hard sparsity and $\ell_1$-relaxed estimators we have further constraints on the vector $\nu$ and thus we only require the minimum and maximum eigenvalue to be bounded subject to these constraints. For instance, in the case of hard sparsity, we know that $\nu$ is a $2s$-sparse vector. Thus it suffices to require the minimum eigenvalue of $V$ and the maximum eigenvalue of $\Sigma$ to be bounded only for such $2s$-sparse vectors (i.e. they should hold for all $2s \times 2s$ square sub-matrices of $\Sigma$ and $V$). Similarly, for the $\ell_1$ based estimators we know that the vector $\nu$ falls in a restricted cone, such that most of the $\ell_1$ norm of $\nu$ is concentrated on the $s$ coordinates of the true coefficient $\theta_0$. Thus we solely need the $\lambda_{\min}$ and $\lambda_{\max}$ constraints to be valid only in this restricted cone of vectors.

# G   Application: Shape Constrained Functions

In this section, we consider the case when $x \in [0, 1]$ and we make shape constraints on $h_0$. We look at both monotonicity/total variation bound constraints and convexity constraints.

## G.1   Monotone functions and functions with small total variation

Consider the case when $h_0$ is a function with range in $[0, 1]$ and of bounded total variation, $BV(h_0) \leq 1$.[15] We let $\mathcal{H} := BV(1)$ denote the latter class of functions. Moreover, we assume that the operator $T$ satisfies that $Th$ is a monotone non-decreasing (or non-increasing) function of $z$ for any monotone non-decreasing (or non-increasing) function $h$ of $x$. Total variation function classes in linear inverse problems with a known linear operator have also been recently analyzed by del Álamo and Munk [2019] and a minimax loss based estimator was also considered, similar in spirit to our general framework.

Observe that any function $h$ with range in $[0, 1]$ and total variation at most 1 can be written as the difference of two non-decreasing functions $h_+, h_-$ with ranges in $[0, 1]$, i.e. $h = h_+ - h_-$. Thus we note that our assumption on $T$ implies that if $h \in BV(1)$, then $Th = Th_+ - Th_- = f_+ - f_-$, where $f_+$ and $f_-$ are monotone non-decreasing functions in $[0, 1]$. Thus $Th \in BV(1)$ and $T(h - h_0) \in BV(2)$. Thus in order to apply our main theorems, it suffices to take $\mathcal{F} = BV(2)$, i.e. the class of functions that can be expressed as the difference of two monotone non-decreasing functions with range in $[0, 2]$. Alternatively, we could also define the norm of a function in the function classes $\mathcal{F}$ and $\mathcal{H}$ as the total variation, which would enable the regularized estimator to adapt to the total variation of the true hypothesis. For simplicity, we assume a known upper bound.

Furthermore, we note that by standard results in statistical learning theory (see e.g. exercise 18, p.153 of Vaart and Wellner [1996] or excercise 3.6.7 of Gine and Nickl [2015]), that the class of monotone functions with range in $[0, 2]$ have metric entropy of the order of $O(1/\epsilon)$. Thus the same holds for the class $BV(2)$, leading to a critical radius of $\delta_n = O\left(n^{-1/3}\right)$, by invoking Corollary 5. Thus by applying our Theorem 1, we get that the corresponding estimators presented in these sections, when $\mathcal{H} = BV(1)$ and $\mathcal{F} = BV(2)$ (and no norm constraints, which can be emulated by setting $B = U = \infty$), satisfy w.p. $1 - \zeta$:

$$\|T(\hat{h} - h_0)\|_2 = O\left(\frac{1}{n^{1/3}} + \sqrt{\frac{\log(1/\zeta)}{n}}\right)$$

The latter rate matches known lower bounds on the achievable RMSE for monotone functions even in the case of exogenous regression problems Chatterjee et al. [2015].

**Efficiently solving the optimization problem**   We can solve the empirical optimization problem by using piece-wise constant monotone functions (or piece-wise linear), i.e. when running the estimator on $n$ samples, we can describe the function $h$ via a $2n$-dimensional vector $\theta = (\theta^+; \theta^-)$, such that $1 \geq \theta_1^+ \geq \ldots \geq \theta_n^+ \geq 0$ and $1 \geq \theta_1^- \geq \ldots \geq \theta_n^- \geq 0$.[16] Let $\Theta$ describe the set of $\theta$

| (a) Isotonic Regression $y \sim x$ | (b) Isotonic IV | (c) Lipschitz Isotonic IV |

Figure 8: Estimated functions based on our minimax estimator under monotonicity constraints. The first figure depicts a direct isotonic regression that ignores endogeneity. The second figure depics our isotonic IV regression, without any lipschitz constraints and the final figure depicts our isotonic IV regression with Lipschitzness constraints. The data generating process was: $h_0(x) = x^2 \, 1\{x > 0\}$, $x = .6\, z + .4\, u + \delta$ and $y = h_0(x) + u + \epsilon$ and $z, u \sim N(0, 2)$ and $\epsilon, \delta \sim N(0, .1)$. ($n = 1000$)

that satisfy these constraints. Similarly, we can describe $f$ via a vector $w = (w^+; w^-)$, such that $2 \geq w_1^+ \geq \ldots \geq w_n^+ \geq 0$ and $2 \geq w_1^- \geq \ldots \geq w_n^- \geq 0$. Let $W$ describe the set of $w$ that satisfy these constraints.

Then for every sample $i$, if we let $q_x(i)$ be the rank of sample $i$ (i.e. sample $i$ has the $q_x(i)$ highest $x$), when we order all samples based on $x$, we can set $h(x_i) = \theta_{q_x(i)}^+ - \theta_{q_x(i)}^-$. Similarly, if we let $q_z(i)$ be the rank of sample $i$, when we order all samples based on $z$, we can set $f(z_i) = w_{q_z(i)}^+ - w_{q_z(i)}^-$. For simplicity of exposition and w.l.o.g. we will assume that samples are ordered in terms of $x$, i.e. $q_x(i) = i$. Thus we can simplify the optimization problem in Theorem 1 as:

$$\min_{\theta \in \Theta} \max_{w \in W} \sum_i (y_i - (\theta_i^+ - \theta_i^-))(w_{q_z(i)}^+ - w_{q_z(i)}^-) - \lambda \sum_{i=1}^n (w_i^+ - w_i^-)^2$$

where the conclusions of the theorem hold if $\lambda \geq 1$. Since the loss:

$$\ell(\theta, w) = \sum_i (y_i - (\theta_i^+ - \theta_i^-))(w_{q_z(i)}^+ - w_{q_z(i)}^-) - \lambda \sum_{i=1}^n (w_i^+ - w_i^-)^2$$

is convex in $\theta$ and concave in $w$ and the spaces $\Theta, W$ are convex sets, we can solve this problem by running simultaneous projected gradient descent for $\theta$ and $w$ separately and returning the average solutions, i.e.: for $t = 1, \ldots, T$:

$$\theta_t = \Pi_\Theta(\theta_{t-1} - \eta \nabla_\theta \ell(\theta_{t-1}, w_{t-1}))$$
$$w_t = \Pi_W(w_{t-1} + \eta \nabla_w \ell(\theta_{t-1}, w_{t-1}))$$

and return $\bar{\theta} = \frac{1}{T} \sum_{t=1}^T \theta_t$. After $O(n/\epsilon^2)$ iterations this would return an $\epsilon$-approximate solution to the minimax problem. Each iteration step would require running a projection on the spaces $\Theta, W$. If we let $\tilde{\theta} \in \mathbb{R}^{2n}$, then we need to find a solution to the problem:

$$\min_{\theta \in \Theta} \frac{1}{2n} \sum_i (\tilde{\theta}_i^+ - \theta_i^+)^2 + (\tilde{\theta}_i^- - \theta_i^-)^2$$

Since the objective and the constraints decompose for the two parts of the vector, this corresponds to running two isotonic regressions for $\theta_i^+$ and $\theta_i^-$ with observations $\tilde{\theta}_i^+$ and $\tilde{\theta}_i^-$. Thus each problem can be solved via the well-known Pool-Adjacent-Violator (PAV) algorithm, which requires $O(n)$ computation time. Similarly, we can deal with the projection of $w$. Thus each iteration of the simultaneous projected gradient descent algorithm requires four calls to the PAV algorithm. If we further want to impose Lipschitzness constraints on our estimates, then we can instead use the Lipschitz-PAV algorithm (see Yeganova and Wilbur [2009], Kakade et al. [2011]) to project onto spaces $\Theta$ and $W$ that are augmented with lipschitzness constraints, e.g. $0 \leq \theta_i^+ - \theta_j^+ \leq L(x_i - x_j)$ for all $i \leq j$. Albeit the LPAV algorithm requires computation of $O(n^2)$.

**Generality of computational approach**    We note that the above approach of solving the endoge-
nous regression problem with shape constraints via our minimax estimator essentially applies to any
type of shape constraints and reduces the minimax problem to a standard square loss problem subject
to the same shape constraints (assuming that both $\mathcal{H}$ and $\mathcal{F}$ satisfy the same shape constraints; i.e.
that these constraints are invariant to the application of the operator $T$). Thus to solve the minimax
problem we simply require an oracle for the square loss problem. In the the setting described in this
section we used the PAV and LPAV algorithm as such oracles. In the next section we will be using a
quadratic optimization subject to linear constraints solver as our oracle.

**Ill-posedness**    We note that the recent work of Chetverikov and Wilhelm [2017], shows that when
$x, z \in [0, 1]$ and the distributions of $x$ and $z$ have full support and lower-bounded density, then for
any function $h$, that is $\alpha$-approximately monotone and continuously differentiable, then $\|Th\|_2 \geq$
$\frac{1}{\tau}\|h\|_{2,t}$, where $\|h\|_{2,t} = \int_{x_1}^{x_2} h(x)^2 dx$, for some $0 < x_1 < x_2 < 1$. The result requires several more
regularity conditions on the operator $T$ and the constant $\tau$ depends on constants in these regularity
conditions (e.g. the lower bound on the density, the quantities $x_1$ and $1-x_2$, the constant $\alpha$, etc). Thus
under these further regularity conditions, we have that for any $h_*$ that is $\alpha$-approximately constant
and for $h$ being a monotone function $\|T(h - h_*)\|_2 \geq \frac{1}{\tau}\|h\|_{2,t}$. Thus our bound on $\|T(h - h_*)\|_2$
also implies a bound on $\|h - h_*\|_{2,t}$. This claim, roughly recovers the main estimation rate result of
Chetverikov and Wilhelm [2017].

## G.2    Convex functions

In this section we consider the case when $h_0$ is assumed to be a convex function in $[0, 1]$, $\Gamma$-Lipschitz
and with range in $[0, 1]$. Moreover, we asusme that the linear operator $T$ satisfies that for any convex
$\Gamma$-Lipschitz function $h$, $Th$ is also convex and $\Gamma$-Lipschitz. Observe that if $T$ is a symmetric density,
i.e. $Th = h \star \rho$ (where $\star$ denotes the convolution operator), for some conditional density function
$\rho$, then we have $(Th)''(z) = (h'') \star \rho \geq 0$, since $h''(x) \geq 0$ and $\rho(x) \geq 0$ for all $x$. Thus any such
symmetric density satisfies our constraints.

The work of Bronshtein [1976] shows that the metric entropy this function class, even in the $d$-
dimensional hypercube, with respect to the $\ell_\infty$ norm, and therefore also with respect to the $\ell_{2,n}$ norm,
is of the order of $\epsilon^{-d/2}$ (see also the recent work of Guntuboyina and Sen [2012]). Thus we get
that by invoking Corollary 5, for $d = 1$, we can choose $\delta_n$ in Theorem 1 in the order of $O(n^{2/5})$,
leading to the corollary that the estimator in Theorem 1, for the case when $\mathcal{H}$ is the space of convex,
$\Gamma$-Lipscthiz functions with range in $[0, 1]$ and $\mathcal{F}$ is the space of differences of two convex functions,
each $\Gamma$-Lipschitz and with range in $[0, 1]$, then w.p. $1 - \zeta$:

$$\|T(\hat{h} - h_0)\|_2 = O\left(\frac{1}{n^{2/5}} + \sqrt{\frac{\log(1/\zeta)}{n}}\right)$$

**Solving the optimization problem**    Moreover, we can address the optimization problem in manner
similar to the previous section. We can choose estimators that optimize over piece-wise linear
functions and hence can be uniquely determined by their values on the $n$ samples, i.e. we can
describe $h$ by a $n$-dimensional vector $\theta$, such that $h(x_i) = \theta_{q_x(i)}$ (where $q_x(i)$ as defined in the
previous section). Similarly, we can descirbe $f \in \mathcal{F}$ via a $2n$-dimensional vector $w = (w^+; w^-)$,
such that $f(z_i) = w^+_{q_z(i)} - w^-_{q_z(i)}$. Subsequently, we can apply the simultaneous projected gradient
descent approach, which reduces the minimax optimization problem to solving the projection problem.
Observe that we can describe the constraints that describe the vectors $\theta$ and $w$ as linear constraints.
Using the same idea as the one described in Example 13.4 of Wainwright [2019], we can express the
convexity constraint as the existence of a subgradient, i.e. there must exist sub-gradients $u, \mu^+, \mu^- \in$
$\mathbb{R}^n$ such that for all $i, j \in [n]$:

$$\theta_j \geq \theta_i + \langle u_i, x_{q_x^{-1}(j)} - x_{q_x^{-1}(i)}\rangle$$
$$w^+_j \geq w^+_i + \langle \mu^+_i, z_{q_z^{-1}(j)} - z_{q_z^{-1}(i)}\rangle$$
$$w^-_j \geq w^-_i + \langle \mu^-_i, z_{q_z^{-1}(j)} - z_{q_z^{-1}(i)}\rangle$$

This is a set of linear constraints of $\theta, w^+, w^-, u, \mu^+, \mu^-$. Moreover, the lipschitz constraints
corresponds to another set of linear constraints, for all $i \in [n]$:

$$-\Gamma(x_{q_x^{-1}(i+1)} - x_{q_x^{-1}(i)}) \leq \theta_{i+1} - \theta_i \leq \Gamma(x_{q_x^{-1}(i+1)} - x_{q_x^{-1}(i)})$$

(a) Bounded TV      (b) Bounded TV and 1-Lipschitz      (c) Convex and 1-Lipschitz

Figure 9: Estimated functions based on our minimax estimator for different sets of shape constraints. In the last figure we also depict the direct regression estimate subject to the same constraints, i.e. if we regressed $y$ on $x$, ignoring endogeneity. The data generating process was: $h_0(x) = |x|$ and $x = .5\,z + .5\,u + \delta$ and $y = h_0(x) + u + \epsilon$ and $z, u \sim N(0, 2)$ and $\epsilon, \delta \sim N(0, .1)$. ($n = 1000$)

and similarly for $w^+, w^-$. Thus projecting onto onto $\Theta$ or $W$, corresponds to a convex quadratic optimization problem with $2n$ variables and $O(n^2)$ linear constraints. Therefore, we can compute such projections in polynomial time at every iteration of the simultaneous projected gradient descent algorithm. In practice, one can achieve substantial speedup by subsampling a set of $s \ll n$ points and restricting the curve to a piece-wise linear function in between these points. This would reduce the number of variables and constraints to $2s$ and $O(s^2)$, correspondingly.

# H   Neural Networks

In this section we describe how one can apply the theoretical findings from the previous sections to understand how to train neural networks that solve the conditional moment problem. We will consider the case when our true function $h_0$ can be represented (or well-approximated) by a deep neural network function of $x$, for some given domain specific network architecture, and we will represent it as $h_0(x) = h_{\theta_0}(x)$, where $\theta_0$ are the weights of the neural net. Moreover, we will assume that the linear operator $T$, satisfies that for any set of weights $\theta$, we have that $Th_\theta$ belongs to a set of functions that can be represented (or well-approximated) as another deep neural network architecture, and we will denote these functions as $f_w(z)$, where $w$ are the weights of the neural net.

**Adversarial GMM Networks (AGMM)**    Thus we can apply our general approach presented in Theorem 1 and consider the estimator:

$$\hat{\theta} = \arg\min_{\theta} \sup_{w} \mathbb{E}_n[\psi(y_i; h_\theta(x_i)) f_w(z)] - \lambda \left( \|f_w\|_{\mathcal{F}}^2 + \frac{U}{n\delta^2} \sum_i f_w(z_i)^2 \right) + \mu \|h_\theta\|_{\mathcal{H}}^2 \quad (14)$$

where $\lambda, \mu, U, \delta$ are hyperparameters that need to satisfy the conditions of the theorem. In particular, if we know that the neural nets $h_\theta, f_w$ output functions in $[0, 1]$, then we can choose $U = B = 1$, $\lambda = \delta^2$, $\mu = 2\delta^2(4L^2 + 27)$, where $L$ is a bound on the lipschitzness of the operator $T$ with respect to the two function space norms and $\delta$ is a bound on the critical radius of the function spaces $\mathcal{F}_3$ and $\hat{\mathcal{G}}_{1, L^2}$. Then problem takes the form:

$$\hat{\theta} = \arg\min_{\theta} \sup_{w} \mathbb{E}_n[\psi(y_i; h_\theta(x_i)) f_w(z)] - \delta^2 \|f_w\|_{\mathcal{F}}^2 - \frac{1}{n} \sum_i f_w(z_i)^2 + c\,\delta^2 \|h_\theta\|_{\mathcal{H}}^2$$

for some constant $c > 1$ that depends on the lipschitzness of the operator $T$. Moreover, theoretically we can set the critical radius $\delta$ by invoking Corollary 5, and using existing results on the pseudo-dimension of the neural network architecture, for which there exist known bounds Anthony and Bartlett [2009] that scale with the number of nodes and edges of the neural net. Moreover, one can also use the recent work of Bartlett et al. [2017], Golowich et al. [2018], to provide size independent bounds on the critical radius of these classes, that only depend on spectral properties of the learned weight matrices of the neural nets.

The work of Bennett et al. [2019] also proposed the use of second moment penalization of the test function, albeit from a different perspective. In particular, their approach stems from a reasoning based on the optimally weighted GMM estimator. In this work we show that second moment

penalization arises also when one wants to achieve fast rates of convergence in terms of mean squared error of the learned function. Moreover, the regularization presented in Bennett et al. [2019] is not a simple second moment penalization, but the second moment of each sample is re-weighted based on the moment evaluated at a preliminary estimate of $\theta$, i.e. $\sum_i f_w(z_i)^2 \psi(y_i; h_{\tilde{\theta}}(x_i))^2$. The preliminary estimate of $\tilde{\theta}$ is an extra burden and typically requires sample splitting and first stage estimation. Here we show that such re-weighting is not required if one simply wants fast projected MSE rates. Moreover, this alternative penalty has the property that as the model $h$ becomes very accurate, then $\psi(y_i; h(x_i)) \approx 0$ and hence the penalty vanishes as the model becomes accurate. This is a big qualitative difference of the two penalties and it is not clear that the penalty that rescales with the moment enjoys the same theoretical guarantees in terms of projected MSE as the simpler second moment penalty.

In the remainder of the section, we will mostly focus on the practical aspect of training neural networks, such as what would be appropriate architectures for the test function space, based on the intuition developed in the prior theoretical developments of the paper and what would be appropriate optimization algorithms for solving the optimization problem.

### H.1 MMD-GMM: A Neural Network Architecture for Adversarial GMM

**Maximum Mean Discrepancy GMM Networks (MMD-GMM).** Our results for RKHS function spaces, suggest that one class of test functions are functions that fall in an RKHS. Observe that Lemma 7 shows that, even when $h$ is an arbitrary function represented by a neural network, as long as $p(x \mid \cdot)$ is a function that belongs to an RKHS $\mathbb{H}_K$, with some kernel $K$, then $Th \in \mathbb{H}_K$. Thus we can choose test functions in $\mathbb{H}_K$.

In many neural network applications, we might have that $p(x \mid \cdot)$ is not in an RKHS (or might have very large RKHS norm), when we use the raw instrument $z$, as $z$ might be very high-dimensional and structured (e.g. an image). However, it might be natural to assume that there is some latent representation $g(z)$ of the instrument $z$, such that: $p(x \mid z) = \rho(x \mid g(z))$ and such that $\rho(x \mid \cdot)$ is in an RKHS.

Thus we will generalize our RKHS approach to augment the adversary with the ability to simultaneously learn the representation $g_w$ (represented as a neural network with weights $w$), and also choose the best function in the RKHS of the implied kernel $K_w(z, z') := K(g_w(z), g_w(z'))$. With this generalization, we are still guaranteeing that $T(h - h_0) \in \mathcal{F}$, whenever $p(x \mid \cdot) = \rho(x \mid g(\cdot))$ and $\rho(x \mid \cdot)$ is in $\mathbb{H}_K$.

Using the variational characterization of the best function in the RKHS presented in Equation (11) we get that the optimization of the adversary can be rephrased as optimizing over test functions of the form $f(z) = \frac{1}{n} \sum_{i=1}^{n} \beta_i K_w(z_i, z)$, leading to an objective for the adversary of the form:

$$\sup_{\beta, w} \frac{1}{n^2} \sum_{i,j} \left( \psi(y_i; h_\theta(x_i)) K_w(z_i, z_j) \beta_j - \delta^2 \beta_i \, K_w(z_i, z_j) \, \beta_j \right) - \frac{1}{n} \sum_i \left( \sum_j \frac{\beta_j}{n} K_w(z_i, z_j) \right)^2$$

which can be written as an average over triplets of samples:

$$\frac{1}{n^3} \sum_{i,j,k} \left( \psi(y_i; h_\theta(x_i)) K_w(z_i, z_j) \beta_j - \beta_i \left( \delta^2 K_w(z_i, z_j) + K_w(z_i, z_k) K_w(z_k, z_j) \right) \beta_j \right)$$

Kernels applied to learned representations have been applied in the context of distribution learning (see e.g. the work on MMD-GANs Li et al. [2017], Binkowski et al. [2018]) and distribution testing (see the recent work of Liu et al. [2020]).

**Unregularized MMD-GMM.** When we omit the $\ell_{2,n}$ regularization then the optimal solution for $\beta$ can be found in closed form (see Proposition 9) and the MMD-GMM simplifies to:

$$\arg\min_\theta \sup_w \frac{1}{n^2} \sum_{i,j} \psi(y_i; h_\theta(x_i)) K_w(z_i, z_j) \psi(y_j; h_\theta(x_j)) + c\delta^4 \|h_\theta\|_{\mathcal{H}}^2 \tag{15}$$

This version (without fixed kernel parameters $w$) was also independently analyzed from the perspective of testing by Muandet et al. [2020]. However, the $\ell_{2,n}$ penalty is crucial for obtaining fast

Figure 10: MMD-GMM architecture of adversary's test function.

rates (e.g. rates that adapt to the eigendecay in the case of RKHS spaces). On the other hand, the unregularized MMD-GMM admits a much easier implementation as we do not need to deal with the $n$ parameters $\beta$ and in the case where we use fixed kernel parameters $w$ we don't even need adversarial training.

**Kernel Approximation** Moreover, as we saw in the RKHS section, it can be beneficial from a computational perspective to approximate the kernel function by sampling a set of training points (either at random or more cleverly based on either leverage scores or k-means clustering) and restrict the space of functions to be supported only on this subset of the points, i.e. $f(z) = \frac{1}{s}\sum_{i=1}^{s}\beta_i K(g_w(z_i^*), g_w(z))$, where $z_i^*$ is a set of representative samples and approximating the RKHS norm penalty with $\sum_{i,j\in S}\beta_i K_w(z_i^*, z_j^*)\beta_j$. This has the benefit of only depending on an $|S|$-dimensional vector $\beta$, that the adversary needs to optimize over, as opposed to $n$-dimensional. Moreover, in practice, instead of constraining the centers to be of the form $g_w(z_i^*)$, we could instead consider arbitrary centers $c_i$ in the space of the output of $g_w$ and consider test functions of the form: $f(z) = \frac{1}{s}\sum_{i=1}^{s}\beta_i K(c_i, g_w(z))$, where $c_i$ are parameters that could also be trained via gradient descent. The latter essentially corresponds to adding what is known as an RBF layer at the end of the adversary neural net. This simplified architecture seems the most appealing from a practical point of view (as it does not require any pre-selection of representative samples $z_i^*$) and is depicted in Figure 11.

**Multi-Kernel MMD-GMM.** The case of sparse linear representations portrays that it might be important to test many different classes of functions, each potentially trained on a separate part of the input space, since different instruments might be correlated with different treatments and many of these treatments can be irrelevant.

$$\sup_{w_1,\ldots,w_m,t\in[m]} \mathbb{E}_n[\psi(y_i; h(x_i))f_{w_t}(z_{S_t})] - \delta^2\|f_{w_t}\|_{\mathcal{F}}^2 - \frac{1}{n}\sum_i f_{w_t}(z_{S_t,i})^2$$

where $S_t$ are pre-defined subsets of the instruments and $z_{S_t}$ corresponds to the sub-vector of instruments. Each of these functions $f_{w_t}$ corresponding to a neural net.

One can also combine the above approaches and set $f_{w_t}(z_{S_t}) = \frac{1}{n}\sum_j \beta_{tj}K_{w_t}(z_{S_t,j}, z_{S_t})$, i.e. allow for the test function that takes as input the subset of the instruments $S_t$ to be in an RKHS of a learned kernel $w_t$. This leads to taking a supremum over a set of kernels in the MMD-GMM objective, where each kernel calculates similarity based on a subset of the input instruments, i.e.:

$$\sup_{\beta,w,t} \frac{1}{n^2}\sum_{i,j}\left(\psi(y_i; h_\theta(x_i))K_w^t(z_i, z_j)\beta_{tj} - \delta^2\beta_{ti}\,K_w^t(z_i, z_j)\,\beta_{tj}\right) - \frac{1}{n}\sum_i\left(\sum_j\frac{\beta_{tj}}{n}K_w^t(z_i, z_j)\right)^2$$

where $K_w^t(z_i, z_j)$ is shorthand notation for $K_{w_t}(z_{S_t,i}, z_{S_t,j})$. The adversary's objective can also be written as choosing a distribution $p_t$ over the $t$ kernels, leading to an adversary objective of:

$$\frac{1}{n^3}\sum_{i,j,k}\left(\psi(y_i; h_\theta(x_i))\sum_t p_t K_w^t(z_i, z_j)\beta_j - \beta_i\,\beta_j\sum_t p_t(\delta^2 K_w^t(z_i, z_j) + K_w^t(z_i, z_k)K_w^t(z_k, z_j))\right)$$

Figure 11: Simplified MMD-GMM architecture of adversary's test function with kernel final activation layer.

We can again reduce the complexity of the optimization problem by restricting to a subset of samples to represent the test functions.

This combined method targets settings where different instruments are correlated with different latent "treatment factors", treatment factors are high-dimensional but only a small subset of them having a large and additively separable effect on the outcome and the relationship between the treatment factor and the instrument is non-linear. Thus it tackles several sources of high-dimensionality in the instrumental variable regression problem.

### H.2   Adversarial Training: Simultaneous Optimistic First-Order Stochastic Optimization

The optimization problem that we are facing is similar to the optimization problem that is encountered in training Generative Adversarial Networks, i.e. we need to solve a non-convex, non-concave zero-sum game, where the strategy of each of the two players are the parameters of a neural net. This is obviously a computationally intractable problem from a worst-case perspective. However, typical instances are far from worst-case and there has been a surge of recent work proposing iterative optimization algorithms inspired by the convex-concave zero-sum game theory (see, e.g. the Optimistic Adam algorithm of Daskalakis et al. [2017]). For instance, one can expect that in practice most early layers of a neural net will change very slowly or will not have a face transition in their non-linearities. In that case, the main parameters that matter are the parameters of the final layers of the two neural nets. However, the zero-sum game is convex-concave in these parameters. Hence, assuming that the features constructed in the final layer of the two neural nets, change slowly, then one should expect convex-concave zero-sum game optimization theory to apply. Such arguments have been recently exploited in the case of square loss minimization with deep over-parameterized neural networks (see e.g. Allen-Zhu et al. [2018], Du et al. [2018], Soltanolkotabi et al. [2019]). It is highly plausible and an interesting question for future research, whether such guarantees extend to the minimax problem that we are facing here. For instance, recent work of Lei et al. [2019], provides an instance of a minimax objective, related to training Wasserstein GANs, where stochastic iterative optimization of neural nets provably converges to an optimal solution.

In our implementation and experiments we used the optimistic Adam algorithm as was also proposed in Bennett et al. [2019]. Other algorithms that could prove useful for our problem are the extra-gradient or stochastic extra-gradient algorithm (see e.g. Hsieh et al. [2019], Mishchenko et al. [2019]).

## I   Random Forests via a Reduction Approach

In this section we deal with the problem of training random forests that solve the non-parametric IV problem. In particular, we aim to develop a learning procedure that learns a hypothesis $h$ that solves the Conditional Moment (1), that is represented as an ensemble of regression trees. Prior work on random forests for causal inference problems has primarily focused on learning forests that capture the heterogeneity of the treatment effect of a treatment, but did not account for non-linear relationships between the treatment and the outcome variable. We will provide a theoretical foundation of the proposed method by taking a reductions approach to the minimax problem defined by our estimator.

For simplicity, throughout this section we will assume that the hypothesis spaces $\mathcal{H}$ and $\mathcal{F}$ are bounded and have bound critical radius and will make no further norm constraints. Thus the estimator

(a) AGMM
($p = 1$, $n = 4000$)

(b) MMD-GMM
($p = 1$, $n = 4000$)

(c) Learned Kernel MMD-GMM
($p = 1$, $n = 4000$)

(d) AGMM
($p = 50$, $n = 4000$)

(e) MMD-GMM
($p = 50$, $n = 4000$)

(f) Learned Kernel MMD-GMM
($p = 50$, $n = 4000$)

Figure 12: Estimated function based on our minimax estimator with neural networks as a function of the relevant treatment. The $h_\theta$ function was a two layer neural net with 100 hidden units. In the first figure an two-layer neural net was used as a test function $f_w$. In the second and third, we used the MMD-GMM test functions with a low rank approximation. In the second we used test functions of the form: $f_\beta(z) = \sum_{i=1}^s \beta_i K_\gamma(c_i, z)$, with $c_i$ a fixed grid of test points in $[-3, 3]^p$ and $K$ is the rbf kernel with parameter $\gamma = .2$, i.e. $K(z, z') = \exp(-\gamma \|z - z'\|_2^2)$. In the third we learned the kernel, i.e. we used test functions of the form: $f_{w,\beta}(z) = \sum_{i=1}^s \beta_i K_\gamma(c_i, g_w(z))$ and $g_w(z) = \text{relu}(Az + b)$ (all the parameters $A, b, \beta, c_i, \gamma$ where trained). The networks were trained via the simultaneous Optimistic Adam algorithm. The data generating process was: $h_0(x) = |x[0]|$ and $x = .6\,z + .4\,u + \delta$ and $y = h_0(x) + u + \epsilon$ and $z \sim N(0, 2I_p)$, $u \sim N(0, 2)$ and $\epsilon, \delta \sim N(0, .1)$.

(a) Weak Instruments
($p = 2$, $n = 4000$)

Figure 13: Estimated function based on our minimax estimator with neural networks as a function of the relevant treatment. The setup is the same as in Figure 12, but we now made the instrument very weak. The data generating process was: $h_0(x) = |x[0]| \, 1\{x[0] > 0\}$ and $x = .05\,z + .95\,u + \delta$ and $y = h_0(x) + u + \epsilon$ and $z \sim N(0, 2I_p)$, $u \sim N(0, 2)$ and $\epsilon, \delta \sim N(0, .1)$.

proposed in Theorem 1[17] takes the simple form of:

$$\hat{h} = \arg\min_{h \in \mathcal{H}} \sup_{f \in \mathcal{F}} \mathbb{E}_n[\psi(y_i; h(x_i))f(z_i)] - \mathbb{E}_n[f(z_i)^2]$$

Since the statistical properties of random forests is an active area of investigation, we will solely focus on the optimization problem and leave the statistical properties (e.g. bounding the critical radius or bias of Random Forest methods) to future work. Our goal is to reduce the aforementioned optimization problem to classification and regression oracles over arbitrary hypothesis spaces. Subsequently in practice we can use random forests as oracles.

**Reducing the Optimization to Regression and Classification Oracles** To achieve this reduction we will make the assumption that the space $\mathcal{F}$ defines a convex image set on the samples, i.e. the set $A = \{(f(z_1), \ldots, f(z_n)) : f \in \mathcal{F}\}$ is a convex set. This can potentially be violated for tree based methods, but in practice will be alleviated when training a forest with a large set of trees.

We will show that we can reduce the problem to a regression oracle over the function space $\mathcal{F}$ and a classification oracle over the function space $\mathcal{B}$. We will assume that we have a regression oracle that solves the square loss problem over $\mathcal{F}$: for any set of labels and features $z_{1:n}, u_{1:n}$ it returns

$$\text{Oracle}_{\mathcal{F}}(z_{1:n}, u_{1:n}) = \arg\min_{f \in \mathcal{F}} \frac{1}{n} \sum_{i=1}^{n} (u_i - f(z_i))^2$$

Moreover, we will assume that we have a classification oracle that solves the weighted binary classification problem over $\mathcal{B}$: for any set of sample weights $w_{1:n}$, binary labels $v_{1:n}$ in $\{0, 1\}$ and features $x_{1:n}$:

$$\text{Oracle}_{\mathcal{H}}(x_{1:n}, v_{1:n}, w_{1:n}) = \arg\max_{h \in \mathcal{H}} \frac{1}{n} \sum_{i=1}^{n} w_i \Pr_{z_i \sim \text{Bernoulli}\left(\frac{1+h(x_i)}{2}\right)} [v_i = z_i]$$

Observe that the objective in the equation above is equivalent to a classification accuracy objective, assuming that $h$ outputs values in $[-1, 1]$ and it corresponds to an expected accuracy objective if one interprets $(h(x) + 1)/2$ as the probability of label 1 conditional on $x$. Having access to these oracles we can then show the following computational result:

**Theorem 4.** *Consider the algorithm where for $t = 1, \ldots, T$: let*

$$u_i^t = \frac{1}{2}\left(y_i - \frac{1}{t-1}\sum_{\tau=1}^{t-1} h_\tau(x_i)\right), \qquad f_t = \textit{Oracle}_{\mathcal{F}}\left(z_{1:n}, u_{1:n}^t\right)$$

$$v_i^t = 1\{f_t(z_i) > 0\}, w_i^t = |f_t(z_i)| \qquad h_t = \textit{Oracle}_{\mathcal{H}}\left(x_{1:n}, v_{1:n}^t, w_{1:n}^t\right)$$

*Then the ensemble hypothesis: $\bar{h} = \frac{1}{T}\sum_{t=1}^{T} h_t$, is a $\frac{8\,(\log(T)+1)}{T}$-approximate solution to the minimax problem in Equation* (6).

In practice, we will consider a random forest regression method as the oracle over $\mathcal{F}$ and a binary decision tree classification method as the oracle for $\mathcal{H}$.

Moreover, we observe that if the hypothesis space $\mathcal{H}$ can be expressed as linear span of base hypothesis, i.e. $\mathcal{H} = \{\sum_i w_i b_i : b_i \in B\}$, then observe that because the best-response problem of the learner is linear in the output of the hypothesis, it suffices to optimize only over the space of base hypothesis. Then the algorithm will return a linear span, supported on $T$ base hypothesis that solves the minimax problem over the whole linear span. This improvement can also lead to statistical rate improvements. For instance, if the base hypothesis $B$ is a VC class with VC dimension $d$ (e.g. a binary decision tree with small depth, see e.g. [Mansour and McAllester, 2000]), then the algorithm returns a convex combination of $T$ base hypothesis, which has VC dimension at most $d\,T$ [Shalev-Shwartz and Ben-David, 2014]. Thus the entropy integral of $\mathcal{H}$ is of the order of $\sqrt{\frac{T\,d\log(n)}{n}}$.

If we further have that the entropy integral of $\mathcal{F}$ is at most $\kappa(\mathcal{F})$, then we get a final rate of the order of:

$$\sqrt{\frac{T\,d\log(n)}{n}} + \kappa(\mathcal{F}) + \frac{\log(T)}{T}$$

Setting, $T = O(n^{1/4})$, one can achieve rates of the order of $n^{-1/4} + \kappa(\mathcal{F})$.

In practice, we will leverage the above observation and train a single binary classification tree at each period of the algorithm, as our Oracle$_{\mathcal{H}}$. In the end the final prediction will be the prediction of the random forest represented by the ensemble of the $T$ trees trained at each period. We refer to this algorithm as Random Forest IV (RFIV).

## J   Experimental Analysis

We consider the following data generating processes: for $n_x = 1$ and $n_z \geq 1$

$$
\begin{aligned}
y &= h_0(x[0]) + e + \delta, & \delta &\sim N(0, .1) \\
x &= \gamma\, z[0] + (1 - \gamma)\, e + \gamma, & z &\sim N(0, 2\, I_{n_z}), e \sim N(0, 2), \gamma \sim N(0, .1)
\end{aligned}
$$

While, when $n_x = n_z > 1$, then we consider the following modified treatment equation:

$$
x = \gamma\, z + (1 - \gamma)\, e + \gamma,
$$

We consider several ranges of the number of samples $n$, number of treatments $n_x$, number of instruments $n_z$ and instrument strength $\gamma$ and the following functional forms for $h_0$:

1. abs: $h_0(x) = |x|$
2. 2dpoly: $h_0(x) = -1.5\, x + .9\, x^2$
3. sigmoid: $h_0(x) = \frac{2}{1 + e^{-2x}}$
4. sin: $h_0(x) = \sin(x)$
5. frequentsin: $h_0(x) = \sin(3\, x)$
6. abssqrt: $h_0(x) = \sqrt{|x|}$
7. step: $h_0(x) = 1\{x < 0\} + 2.5\, 1\{x \geq 0\}$
8. 3dpoly: $h_0(x) = -1.5\, x + .9\, x^2 + x^3$
9. linear: $h_0(x) = x$
10. randpw: piece wise linear function drawn at random
11. abspos: $h_0(x) = x\, 1\{x \geq 0\}$
12. sqrpos: $h_0(x) = x^2\, 1\{x \geq 0\}$
13. band: $h_0(x) = 1\{-.75 \leq x \leq .75\}$
14. invband: $h_0(x) = 1 - 1\{-.75 \leq x \leq .75\}$
15. steplinear: $h_0(x) = 2\, 1\{x \geq 0\} - x$
16. pwlinear: $h_0(x) = (x + 1)\, 1\{x \leq -1\} + (x - 1)\, 1\{x >= 1\}$

We consider as classic benchmarks 2SLS with a polynomial features of degree 3 (2SLS) and a regularized version of 2SLS where ElasticNetCV is used in both stages (Reg2SLS). We have implemented several of the algorithms described in the paper:

1. NystromRKHS: The method described in Appendix E, with the Nystrom approximation described in Appendix E.3. We used 100 Nystrom samples for the approximation.

2. ConvexIV: The variant of the method described in Appendix G.2 with both lipscthiz and convexity constraints (lipschitz bound of $L = 2$).

3. TVIV: The variant of the method described in Appendix G.1 without a lipschitz constraint and only total variation constraint.

4. LipTVIV: The variant of the method described in Appendix G.1 with lipscthiz constraint and total variation constraint (lipscthiz bound of $L = 2$)

5. RFIV: The method described in Appendix I, where a Random Forest Regressor is used as an oracle for the adversary (with 40 trees, max depth 2, bootstrap sub-sampling enabled, and minimum leaf size of 40) and Random Forest Classifier (with 5 trees, max depth 2, minimum leaf size of 40 and bootstrap subsampling disabled) was used as an oracle for the learner. The optimization was run for $T = 200$ iterations.

6. SpLin: The method described in Appendix F.2 with the specific optimization method described in Proposition 13.

7. StSpLin: A stochastic gradient descent variant of SpLin, where a mini-batch of 100 samples is used at every step to calculate the co-variance matrices.

8. AGMM: The method described in Equation (14). A two-layer neural net with 100 hidden units at each layer and leaky ReLU units was used for both the learner and the adversary architecture. Optimization was done via the Optimistic Adam.

9. KLayerFixed: The variant of the method described in Appendix H.1, where an RBF layer is attached at the end of the adversary's architecture with fixed centers, i.e. testing functions of the form: $f(z) = \sum_{j=1}^{n_{\text{centers}}} K(c_j, g_w(z))\beta_j$, with $n_{\text{centers}} = 100$. The centers $c_j$ are placed in a 100 dimensional feature space and the function $g_w$ is a two-layer neural net with 100 hidden units in each layer.

10. KLayerTrained: The same as KLayerFixed, but the centers of the RBF layer are trained.

11. CentroidMMD: The version of the MMMD-GMM in Appendix H.1, where we select a subset of the data points to use as centers in the Kernel approximation, i.e. testing functions of the form: $f(z) = \sum_{j=1}^{n_{\text{centers}}} K(g_w(z_j^*), g_w(z))\beta_j$. $z_j^*$ are chosen as the centroids of a KMeans clustering and $n_{\text{centers}} = 100$. $g_w$ is the same architecture as in KLayerFixed.

12. KLossMMD: The method described in Equation (15), where no $\ell_{2,n}$ penalty is imposed on the adversary test function. $g_w$ is the same architecture as in KLayerFixed.

In addition to these regimes, we consider high-dimensional experiments with images, following the scenarios proposed in Bennett et al. [2019] where either the instrument $z$ or treatment $x$ or both are images from the MNIST dataset consisting of grayscale images of $28 \times 28$ pixels. We compare the performance of our approaches to that of Bennett et al. [2019], using their code. A full description of the DGP is given in Appendix J.1.

**Results.** The main findings are: i) for small number of treatments, the RKHS method with a Nystrom approximation (NystromRKHS), outperforms all methods (Figure 1) with only exception being functions that are highly non-smooth or non-continuous, in which case the methods that are based on shape constraints (ConvexIV, TVIV, LipTVIV) are better, ii) for moderate number of instruments and treatments, Random Forest IV (RFIV) significantly outperforms most methods, with second best being neural networks (AGMM, KLayerTrained) (Figure 2), iii) the estimator for sparse linear hypotheses can handle an ultra-high dimensional regime (Figure 3), iv) neural network methods (AGMM, KLayerTrained) outperform the state of the art in prior work [Bennett et al., 2019] for tasks that involve images (Figure 4). The figures below present the average MSE across 100 experiments (10 experiments for Figure 4) and two times the standard error of the average MSE. Note that for non-parametric IV there was no prior Random Forest (RF) algorithm, as we outline in the Random Forest section. We present the first algorithm for this setting. Prior Random Forest algorithms for IV setup only work when one makes the assumption of linearity w.r.t. to treatment and estimates heterogeneity with respect to exogenous features (such as the IV forest of [Wager and Athey, 2018]).

| | NystromRKHS | 2SLS | Reg2SLS | ConvexIV | TVIV | LipTVIV | RFIV |
|---|---|---|---|---|---|---|---|
| abs | **0.045 ± 0.010** | 0.100 ± 0.035 | 1.733 ± 2.981 | 0.054 ± 0.005 | 0.089 ± 0.005 | 0.047 ± 0.004 | 0.084 ± 0.007 |
| 2dpoly | 0.121 ± 0.014 | **0.036 ± 0.022** | 9.068 ± 16.071 | 0.060 ± 0.007 | 0.090 ± 0.009 | 0.069 ± 0.009 | 0.379 ± 0.022 |
| sigmoid | **0.016 ± 0.003** | 0.071 ± 0.037 | 0.429 ± 0.244 | 0.029 ± 0.005 | 0.067 ± 0.004 | 0.034 ± 0.003 | 0.044 ± 0.006 |
| sin | **0.023 ± 0.003** | 0.090 ± 0.042 | 0.801 ± 0.420 | 0.055 ± 0.005 | 0.074 ± 0.004 | 0.036 ± 0.003 | 0.057 ± 0.007 |
| frequentsin | 0.129 ± 0.005 | 0.193 ± 0.040 | 0.145 ± 0.017 | 0.143 ± 0.008 | 0.115 ± 0.005 | **0.106 ± 0.005** | 0.126 ± 0.010 |
| abssqrt | **0.033 ± 0.004** | 0.099 ± 0.039 | 0.117 ± 0.046 | 0.045 ± 0.007 | 0.096 ± 0.006 | 0.047 ± 0.004 | 0.064 ± 0.008 |
| step | **0.035 ± 0.003** | 0.103 ± 0.043 | 0.497 ± 0.276 | 0.054 ± 0.005 | 0.073 ± 0.004 | 0.044 ± 0.003 | 0.056 ± 0.007 |
| 3dpoly | 0.220 ± 0.037 | **0.004 ± 0.003** | 0.066 ± 0.014 | 0.396 ± 0.051 | 0.138 ± 0.028 | 0.190 ± 0.036 | 0.687 ± 0.069 |
| linear | 0.019 ± 0.003 | 0.038 ± 0.021 | 0.355 ± 0.189 | **0.017 ± 0.005** | 0.042 ± 0.002 | 0.027 ± 0.002 | 0.048 ± 0.005 |
| randpw | 0.067 ± 0.012 | 0.092 ± 0.024 | 3.810 ± 5.878 | 0.162 ± 0.032 | 0.073 ± 0.009 | **0.046 ± 0.006** | 0.121 ± 0.015 |
| abspos | **0.022 ± 0.003** | 0.060 ± 0.027 | 0.299 ± 0.157 | 0.022 ± 0.004 | 0.062 ± 0.004 | 0.033 ± 0.003 | 0.055 ± 0.006 |
| sqrpos | 0.064 ± 0.013 | **0.026 ± 0.015** | 0.490 ± 0.494 | 0.030 ± 0.006 | 0.034 ± 0.003 | 0.033 ± 0.005 | 0.181 ± 0.013 |
| band | **0.059 ± 0.003** | 0.125 ± 0.051 | 0.085 ± 0.017 | 0.086 ± 0.008 | 0.102 ± 0.006 | 0.059 ± 0.004 | 0.071 ± 0.008 |
| invband | **0.056 ± 0.003** | 0.130 ± 0.041 | 0.138 ± 0.051 | 0.075 ± 0.008 | 0.102 ± 0.006 | 0.059 ± 0.004 | 0.073 ± 0.008 |
| steplinear | 0.141 ± 0.009 | 0.231 ± 0.085 | 0.203 ± 0.063 | 0.138 ± 0.008 | 0.156 ± 0.009 | **0.100 ± 0.006** | 0.141 ± 0.011 |
| pwlinear | **0.032 ± 0.004** | 0.051 ± 0.024 | 0.058 ± 0.025 | 0.037 ± 0.006 | 0.061 ± 0.003 | 0.035 ± 0.003 | 0.068 ± 0.006 |

Figure 14: $n = 300, n_z = 1, n_x = 1, \gamma = .6$

| | NystromRKHS | 2SLS | Reg2SLS | ConvexIV | TVIV | LipTVIV | RFIV |
|---|---|---|---|---|---|---|---|
| abs | **0.010 ± 0.001** | 0.025 ± 0.001 | 0.025 ± 0.002 | 0.031 ± 0.001 | 0.031 ± 0.001 | 0.021 ± 0.001 | 0.026 ± 0.002 |
| 2dpoly | 0.022 ± 0.005 | **0.002 ± 0.000** | 0.043 ± 0.039 | 0.052 ± 0.004 | 0.034 ± 0.004 | 0.037 ± 0.004 | 0.286 ± 0.013 |
| sigmoid | **0.005 ± 0.001** | 0.007 ± 0.001 | 0.021 ± 0.017 | 0.011 ± 0.000 | 0.018 ± 0.001 | 0.008 ± 0.001 | 0.015 ± 0.001 |
| sin | **0.005 ± 0.001** | 0.013 ± 0.002 | 0.033 ± 0.025 | 0.035 ± 0.001 | 0.020 ± 0.001 | 0.009 ± 0.001 | 0.017 ± 0.001 |
| frequentsin | 0.118 ± 0.001 | 0.117 ± 0.001 | 0.115 ± 0.001 | 0.116 ± 0.001 | 0.089 ± 0.002 | 0.105 ± 0.002 | **0.087 ± 0.004** |
| abssqrt | **0.011 ± 0.001** | 0.018 ± 0.001 | 0.018 ± 0.001 | 0.020 ± 0.001 | 0.028 ± 0.001 | 0.016 ± 0.001 | 0.022 ± 0.002 |
| step | 0.022 ± 0.001 | 0.029 ± 0.001 | 0.043 ± 0.017 | 0.034 ± 0.001 | 0.026 ± 0.001 | **0.020 ± 0.001** | 0.026 ± 0.002 |
| 3dpoly | 0.028 ± 0.012 | **0.000 ± 0.000** | 0.010 ± 0.003 | 0.325 ± 0.026 | 0.086 ± 0.019 | 0.121 ± 0.020 | 0.375 ± 0.036 |
| linear | 0.004 ± 0.001 | **0.002 ± 0.000** | 0.022 ± 0.022 | 0.002 ± 0.000 | 0.013 ± 0.001 | 0.007 ± 0.001 | 0.012 ± 0.001 |
| randpw | 0.031 ± 0.006 | 0.057 ± 0.010 | 0.131 ± 0.111 | 0.150 ± 0.032 | 0.032 ± 0.004 | **0.029 ± 0.004** | 0.054 ± 0.010 |
| abspos | 0.006 ± 0.001 | 0.007 ± 0.001 | 0.015 ± 0.009 | **0.005 ± 0.000** | 0.016 ± 0.001 | 0.008 ± 0.001 | 0.016 ± 0.001 |
| sqrpos | 0.011 ± 0.003 | **0.004 ± 0.000** | 0.010 ± 0.006 | 0.011 ± 0.002 | 0.011 ± 0.001 | 0.012 ± 0.002 | 0.091 ± 0.007 |
| band | 0.031 ± 0.001 | 0.046 ± 0.001 | 0.046 ± 0.001 | 0.059 ± 0.001 | 0.039 ± 0.002 | **0.031 ± 0.002** | 0.032 ± 0.002 |
| invband | **0.031 ± 0.001** | 0.046 ± 0.001 | 0.046 ± 0.001 | 0.049 ± 0.001 | 0.039 ± 0.002 | 0.031 ± 0.001 | 0.032 ± 0.002 |
| steplinear | 0.066 ± 0.002 | 0.085 ± 0.003 | 0.089 ± 0.005 | 0.104 ± 0.001 | 0.074 ± 0.002 | **0.064 ± 0.002** | 0.066 ± 0.003 |
| pwlinear | **0.007 ± 0.001** | 0.009 ± 0.000 | 0.012 ± 0.001 | 0.017 ± 0.001 | 0.016 ± 0.001 | 0.009 ± 0.001 | 0.016 ± 0.001 |

Figure 15: $n = 2000$, $n_z = 1$, $n_x = 1$, $\gamma = .6$

| | NystromRKHS | 2SLS | Reg2SLS | ConvexIV | TVIV | LipTVIV | RFIV |
|---|---|---|---|---|---|---|---|
| abs | **0.008 ± 0.001** | 0.027 ± 0.001 | 0.027 ± 0.001 | 0.024 ± 0.000 | 0.016 ± 0.001 | 0.012 ± 0.001 | 0.017 ± 0.001 |
| 2dpoly | 0.009 ± 0.002 | **0.001 ± 0.000** | 0.016 ± 0.007 | 0.036 ± 0.003 | 0.018 ± 0.002 | 0.022 ± 0.003 | 0.151 ± 0.010 |
| sigmoid | **0.004 ± 0.000** | 0.007 ± 0.000 | 0.017 ± 0.005 | 0.013 ± 0.000 | 0.011 ± 0.001 | 0.007 ± 0.000 | 0.012 ± 0.001 |
| sin | **0.003 ± 0.000** | 0.023 ± 0.002 | 0.033 ± 0.006 | 0.055 ± 0.001 | 0.013 ± 0.001 | 0.009 ± 0.001 | 0.014 ± 0.001 |
| frequentsin | 0.114 ± 0.001 | 0.114 ± 0.001 | 0.113 ± 0.001 | 0.114 ± 0.001 | 0.048 ± 0.001 | 0.051 ± 0.001 | **0.024 ± 0.001** |
| abssqrt | **0.008 ± 0.001** | 0.017 ± 0.001 | 0.017 ± 0.001 | 0.017 ± 0.000 | 0.015 ± 0.001 | 0.011 ± 0.001 | 0.015 ± 0.001 |
| step | 0.021 ± 0.000 | 0.031 ± 0.001 | 0.039 ± 0.004 | 0.038 ± 0.000 | 0.015 ± 0.001 | **0.012 ± 0.001** | 0.018 ± 0.001 |
| 3dpoly | 0.030 ± 0.006 | **0.000 ± 0.000** | 0.001 ± 0.000 | 0.344 ± 0.025 | 0.081 ± 0.015 | 0.114 ± 0.016 | 0.366 ± 0.031 |
| linear | 0.003 ± 0.000 | **0.001 ± 0.000** | 0.016 ± 0.008 | 0.002 ± 0.000 | 0.009 ± 0.000 | 0.008 ± 0.000 | 0.010 ± 0.001 |
| randpw | 0.021 ± 0.004 | 0.055 ± 0.009 | 0.069 ± 0.010 | 0.157 ± 0.032 | 0.015 ± 0.002 | **0.013 ± 0.002** | 0.028 ± 0.004 |
| abspos | 0.004 ± 0.000 | 0.007 ± 0.000 | 0.013 ± 0.003 | **0.003 ± 0.000** | 0.010 ± 0.001 | 0.007 ± 0.000 | 0.013 ± 0.001 |
| sqrpos | 0.008 ± 0.002 | **0.004 ± 0.000** | 0.008 ± 0.003 | 0.025 ± 0.003 | 0.013 ± 0.002 | 0.018 ± 0.002 | 0.109 ± 0.008 |
| band | 0.026 ± 0.001 | 0.044 ± 0.001 | 0.044 ± 0.001 | 0.056 ± 0.001 | 0.018 ± 0.001 | **0.014 ± 0.001** | 0.020 ± 0.001 |
| invband | 0.026 ± 0.001 | 0.044 ± 0.001 | 0.044 ± 0.001 | 0.046 ± 0.001 | 0.018 ± 0.001 | **0.015 ± 0.001** | 0.020 ± 0.001 |
| steplinear | 0.042 ± 0.001 | 0.064 ± 0.001 | 0.066 ± 0.002 | 0.079 ± 0.001 | 0.036 ± 0.001 | **0.032 ± 0.001** | 0.032 ± 0.001 |
| pwlinear | **0.005 ± 0.000** | 0.010 ± 0.000 | 0.013 ± 0.002 | 0.019 ± 0.000 | 0.011 ± 0.001 | 0.008 ± 0.001 | 0.014 ± 0.001 |

Figure 16: $n = 2000$, $n_z = 1$, $n_x = 1$, $\gamma = .8$

| | NystromRKHS | 2SLS | Reg2SLS | RFIV |
|---|---|---|---|---|
| abs | 0.026 ± 0.010 | 0.025 ± 0.001 | 0.054 ± 0.007 | **0.023 ± 0.001** |
| 2dpoly | 0.033 ± 0.006 | **0.002 ± 0.000** | 0.361 ± 0.059 | 0.292 ± 0.012 |
| sigmoid | 0.015 ± 0.006 | **0.006 ± 0.000** | 0.096 ± 0.016 | 0.014 ± 0.001 |
| sin | 0.019 ± 0.007 | **0.012 ± 0.001** | 0.142 ± 0.024 | 0.016 ± 0.001 |
| frequentsin | 0.131 ± 0.007 | 0.117 ± 0.001 | 0.116 ± 0.003 | **0.069 ± 0.003** |
| abssqrt | 0.027 ± 0.010 | **0.018 ± 0.001** | 0.026 ± 0.004 | 0.019 ± 0.001 |
| step | 0.036 ± 0.006 | 0.028 ± 0.001 | 0.116 ± 0.017 | **0.021 ± 0.001** |
| 3dpoly | 0.018 ± 0.008 | **0.000 ± 0.000** | 0.021 ± 0.003 | 0.416 ± 0.041 |
| linear | 0.015 ± 0.005 | **0.002 ± 0.000** | 0.120 ± 0.019 | 0.012 ± 0.001 |
| randpw | **0.047 ± 0.010** | 0.057 ± 0.011 | 0.448 ± 0.185 | 0.050 ± 0.009 |
| abspos | 0.019 ± 0.007 | **0.007 ± 0.001** | 0.060 ± 0.010 | 0.014 ± 0.001 |
| sqrpos | 0.025 ± 0.005 | **0.004 ± 0.001** | 0.065 ± 0.010 | 0.092 ± 0.007 |
| band | 0.056 ± 0.012 | 0.046 ± 0.001 | 0.053 ± 0.003 | **0.027 ± 0.002** |
| invband | 0.051 ± 0.012 | 0.046 ± 0.001 | 0.052 ± 0.004 | **0.027 ± 0.002** |
| steplinear | 0.087 ± 0.006 | 0.084 ± 0.001 | 0.103 ± 0.005 | **0.059 ± 0.002** |
| pwlinear | 0.023 ± 0.008 | **0.010 ± 0.001** | 0.026 ± 0.004 | 0.014 ± 0.001 |

Figure 17: $n = 2000$, $n_z = 5$, $n_x = 1$, $\gamma = .6$

|  | NystromRKHS | 2SLS | Reg2SLS | RFIV |
|---|---|---|---|---|
| abs | $0.027 \pm 0.011$ | $0.035 \pm 0.002$ | $0.107 \pm 0.016$ | $\mathbf{0.021 \pm 0.001}$ |
| 2dpoly | $0.050 \pm 0.019$ | $\mathbf{0.006 \pm 0.000}$ | $0.545 \pm 0.080$ | $0.282 \pm 0.014$ |
| sigmoid | $0.017 \pm 0.009$ | $0.014 \pm 0.001$ | $0.115 \pm 0.023$ | $\mathbf{0.013 \pm 0.001}$ |
| sin | $0.023 \pm 0.009$ | $0.020 \pm 0.001$ | $0.181 \pm 0.045$ | $\mathbf{0.017 \pm 0.001}$ |
| frequentsin | $0.136 \pm 0.012$ | $0.126 \pm 0.001$ | $0.117 \pm 0.003$ | $\mathbf{0.065 \pm 0.003}$ |
| abssqrt | $0.026 \pm 0.008$ | $0.030 \pm 0.002$ | $0.038 \pm 0.006$ | $\mathbf{0.018 \pm 0.002}$ |
| step | $0.035 \pm 0.008$ | $0.036 \pm 0.001$ | $0.135 \pm 0.025$ | $\mathbf{0.021 \pm 0.002}$ |
| 3dpoly | $0.022 \pm 0.018$ | $\mathbf{0.001 \pm 0.000}$ | $0.035 \pm 0.005$ | $0.402 \pm 0.045$ |
| linear | $0.022 \pm 0.008$ | $\mathbf{0.007 \pm 0.001}$ | $0.123 \pm 0.020$ | $0.011 \pm 0.001$ |
| randpw | $\mathbf{0.047 \pm 0.009}$ | $0.061 \pm 0.010$ | $0.457 \pm 0.165$ | $0.051 \pm 0.011$ |
| abspos | $0.022 \pm 0.008$ | $0.015 \pm 0.001$ | $0.082 \pm 0.015$ | $\mathbf{0.013 \pm 0.001}$ |
| sqrpos | $0.042 \pm 0.017$ | $\mathbf{0.008 \pm 0.001}$ | $0.129 \pm 0.020$ | $0.086 \pm 0.006$ |
| band | $0.056 \pm 0.013$ | $0.056 \pm 0.001$ | $0.062 \pm 0.007$ | $\mathbf{0.027 \pm 0.002}$ |
| invband | $0.052 \pm 0.012$ | $0.058 \pm 0.002$ | $0.060 \pm 0.006$ | $\mathbf{0.026 \pm 0.002}$ |
| steplinear | $0.102 \pm 0.013$ | $0.097 \pm 0.002$ | $0.099 \pm 0.005$ | $\mathbf{0.059 \pm 0.003}$ |
| pwlinear | $0.031 \pm 0.008$ | $0.017 \pm 0.002$ | $0.033 \pm 0.006$ | $\mathbf{0.014 \pm 0.001}$ |

Figure 18: $n = 2000, n_z = 10, n_x = 1, \gamma = .6$

|  | NystromRKHS | 2SLS | Reg2SLS | RFIV |
|---|---|---|---|---|
| abs | $0.051 \pm 0.002$ | $0.262 \pm 0.076$ | $\mathbf{0.031 \pm 0.002}$ | $0.038 \pm 0.001$ |
| 2dpoly | $0.226 \pm 0.012$ | $0.106 \pm 0.033$ | $\mathbf{0.105 \pm 0.027}$ | $0.316 \pm 0.013$ |
| sigmoid | $0.025 \pm 0.002$ | $0.198 \pm 0.060$ | $0.056 \pm 0.002$ | $\mathbf{0.015 \pm 0.001}$ |
| sin | $0.035 \pm 0.002$ | $0.222 \pm 0.066$ | $0.077 \pm 0.006$ | $\mathbf{0.022 \pm 0.001}$ |
| frequentsin | $0.140 \pm 0.002$ | $0.386 \pm 0.084$ | $0.114 \pm 0.001$ | $\mathbf{0.108 \pm 0.002}$ |
| abssqrt | $0.037 \pm 0.002$ | $0.288 \pm 0.087$ | $0.025 \pm 0.001$ | $\mathbf{0.025 \pm 0.001}$ |
| step | $0.045 \pm 0.002$ | $0.234 \pm 0.064$ | $0.076 \pm 0.002$ | $\mathbf{0.025 \pm 0.001}$ |
| 3dpoly | $0.308 \pm 0.030$ | $\mathbf{0.009 \pm 0.003}$ | $0.027 \pm 0.004$ | $0.414 \pm 0.034$ |
| linear | $0.040 \pm 0.002$ | $0.124 \pm 0.039$ | $0.058 \pm 0.006$ | $\mathbf{0.014 \pm 0.001}$ |
| randpw | $0.131 \pm 0.015$ | $0.266 \pm 0.163$ | $0.161 \pm 0.028$ | $\mathbf{0.077 \pm 0.011}$ |
| abspos | $0.034 \pm 0.002$ | $0.185 \pm 0.057$ | $0.043 \pm 0.002$ | $\mathbf{0.017 \pm 0.001}$ |
| sqrpos | $0.111 \pm 0.008$ | $0.088 \pm 0.028$ | $\mathbf{0.029 \pm 0.002}$ | $0.097 \pm 0.006$ |
| band | $0.060 \pm 0.002$ | $0.327 \pm 0.085$ | $0.055 \pm 0.001$ | $\mathbf{0.038 \pm 0.001}$ |
| invband | $0.060 \pm 0.002$ | $0.311 \pm 0.089$ | $0.054 \pm 0.001$ | $\mathbf{0.039 \pm 0.001}$ |
| steplinear | $0.161 \pm 0.004$ | $0.457 \pm 0.115$ | $0.100 \pm 0.003$ | $\mathbf{0.090 \pm 0.002}$ |
| pwlinear | $0.052 \pm 0.003$ | $0.187 \pm 0.058$ | $\mathbf{0.017 \pm 0.001}$ | $0.018 \pm 0.001$ |

Figure 19: $n = 2000, n_z = 5, n_x = 5, \gamma = .6$

|  | NystromRKHS | 2SLS | Reg2SLS | RFIV |
|---|---|---|---|---|
| abs | $0.143 \pm 0.005$ | $10050.672 \pm 13267.141$ | $0.122 \pm 0.011$ | $\mathbf{0.049 \pm 0.001}$ |
| 2dpoly | $0.595 \pm 0.025$ | $5890.128 \pm 8261.553$ | $4.510 \pm 1.245$ | $\mathbf{0.346 \pm 0.014}$ |
| sigmoid | $0.045 \pm 0.003$ | $11712.144 \pm 16799.716$ | $0.091 \pm 0.005$ | $\mathbf{0.017 \pm 0.001}$ |
| sin | $0.058 \pm 0.003$ | $13769.428 \pm 20805.861$ | $0.114 \pm 0.006$ | $\mathbf{0.029 \pm 0.001}$ |
| frequentsin | $0.136 \pm 0.004$ | $12928.749 \pm 19554.361$ | $0.144 \pm 0.004$ | $\mathbf{0.120 \pm 0.002}$ |
| abssqrt | $0.062 \pm 0.004$ | $12764.707 \pm 17195.564$ | $0.079 \pm 0.005$ | $\mathbf{0.034 \pm 0.001}$ |
| step | $0.064 \pm 0.003$ | $12187.342 \pm 17814.756$ | $0.109 \pm 0.004$ | $\mathbf{0.027 \pm 0.001}$ |
| 3dpoly | $0.648 \pm 0.039$ | $432.572 \pm 596.731$ | $\mathbf{0.061 \pm 0.005}$ | $0.444 \pm 0.029$ |
| linear | $0.080 \pm 0.002$ | $6964.376 \pm 9566.774$ | $0.107 \pm 0.006$ | $\mathbf{0.016 \pm 0.001}$ |
| randpw | $0.272 \pm 0.029$ | $1882.000 \pm 1998.862$ | $0.682 \pm 0.539$ | $\mathbf{0.093 \pm 0.013}$ |
| abspos | $0.067 \pm 0.003$ | $8841.523 \pm 11921.282$ | $0.095 \pm 0.005$ | $\mathbf{0.020 \pm 0.001}$ |
| sqrpos | $0.243 \pm 0.010$ | $4250.312 \pm 5449.534$ | $0.126 \pm 0.014$ | $\mathbf{0.105 \pm 0.006}$ |
| band | $0.078 \pm 0.004$ | $20401.368 \pm 29655.000$ | $0.090 \pm 0.004$ | $\mathbf{0.049 \pm 0.002}$ |
| invband | $0.079 \pm 0.004$ | $11210.315 \pm 14271.847$ | $0.090 \pm 0.005$ | $\mathbf{0.048 \pm 0.002}$ |
| steplinear | $0.212 \pm 0.005$ | $22217.181 \pm 33274.806$ | $0.141 \pm 0.005$ | $\mathbf{0.110 \pm 0.002}$ |
| pwlinear | $0.075 \pm 0.003$ | $9280.655 \pm 12159.776$ | $0.041 \pm 0.004$ | $\mathbf{0.021 \pm 0.001}$ |

Figure 20: $n = 2000, n_z = 10, n_x = 10, \gamma = .6$

| | AGMM | KLayerFixed | KLayerTrained | CentroidMMD | KLossMMD |
|---|---|---|---|---|---|
| abs | **0.062 ± 0.003** | 0.190 ± 0.006 | 0.127 ± 0.007 | 0.114 ± 0.007 | 0.193 ± 0.007 |
| 2dpoly | **0.099 ± 0.006** | 0.971 ± 0.040 | 0.240 ± 0.014 | 0.204 ± 0.022 | 0.467 ± 0.023 |
| sigmoid | 0.040 ± 0.001 | 0.063 ± 0.002 | **0.024 ± 0.001** | 0.058 ± 0.003 | 0.043 ± 0.003 |
| sin | 0.074 ± 0.002 | 0.076 ± 0.002 | **0.057 ± 0.002** | 0.098 ± 0.003 | 0.083 ± 0.004 |
| frequentsin | 0.158 ± 0.002 | **0.120 ± 0.002** | 0.128 ± 0.002 | 0.181 ± 0.004 | 0.160 ± 0.007 |
| abssqrt | 0.060 ± 0.003 | **0.058 ± 0.004** | 0.060 ± 0.003 | 0.093 ± 0.004 | 0.090 ± 0.007 |
| step | 0.066 ± 0.002 | 0.076 ± 0.002 | **0.050 ± 0.001** | 0.088 ± 0.003 | 0.069 ± 0.003 |
| 3dpoly | **0.426 ± 0.027** | 0.716 ± 0.037 | 0.491 ± 0.029 | 0.496 ± 0.030 | 0.526 ± 0.032 |
| linear | 0.020 ± 0.001 | 0.142 ± 0.003 | **0.013 ± 0.001** | 0.029 ± 0.002 | 0.027 ± 0.001 |
| randpw | **0.127 ± 0.020** | 0.449 ± 0.051 | 0.165 ± 0.024 | 0.169 ± 0.025 | 0.218 ± 0.030 |
| abspos | **0.034 ± 0.002** | 0.090 ± 0.003 | 0.039 ± 0.002 | 0.057 ± 0.003 | 0.060 ± 0.003 |
| sqrpos | **0.059 ± 0.003** | 0.347 ± 0.013 | 0.131 ± 0.007 | 0.113 ± 0.009 | 0.178 ± 0.009 |
| band | 0.088 ± 0.003 | **0.068 ± 0.002** | 0.074 ± 0.003 | 0.117 ± 0.004 | 0.130 ± 0.037 |
| invband | 0.088 ± 0.003 | **0.073 ± 0.005** | 0.077 ± 0.003 | 0.114 ± 0.004 | 0.120 ± 0.026 |
| steplinear | 0.176 ± 0.003 | 0.197 ± 0.004 | **0.133 ± 0.003** | 0.218 ± 0.005 | 0.170 ± 0.010 |
| pwlinear | 0.049 ± 0.001 | 0.074 ± 0.002 | **0.033 ± 0.001** | 0.063 ± 0.002 | 0.049 ± 0.002 |

Figure 21: $n = 2000, n_z = 10, n_x = 10, \gamma = .6$

| $p =$ | 1000 | 10000 | 100000 | 1000000 |
|---|---|---|---|---|
| SpLin | 0.020 ± 0.003 | 0.021 ± 0.003 | - | - |
| StSpLin | 0.020 ± 0.002 | 0.023 ± 0.002 | 0.033 ± 0.002 | 0.050 ± 0.004 |

Figure 22: $n = 400, n_z = n_x := p, \gamma = .6, h_0(x[0]) = x[0]$

| | DeepGMM (Bennett et al. [2019]) | AGMM | KLayerTrained |
|---|---|---|---|
| $\text{MNIST}_z$ | 0.12 ± 0.07 | 0.04 ± 0.03 | 0.05 ± 0.02 |
| $\text{MNIST}_x$ | 0.34 ± 0.21 | 0.24 ± 0.08 | 0.36 ± 0.20 |
| $\text{MNIST}_{xz}$ | 0.26 ± 0.16 | 0.21 ± 0.07 | 0.26 ± 0.11 |

Figure 23: MSE on the high-dimensional DGPs

## J.1 Experiments with Image Data

In this section, we describe the experimental setup for our experiments with high-dimensional data using the MNIST dataset. We replicate the data-generating process of Bennett et al. [2019]. We present a full description here for completeness.

**The Data-Generating Process** We begin by describing a low-dimensional DGP which will define a mapping for $x$ or $z$ or both to be MNIST images. The data-generating process is:

$$y = g_0(x^{\text{low}}) + e + \delta$$
$$z^{\text{low}} \sim \text{Uniform}([-3, 3]^2)$$
$$x^{\text{low}} = z_1^{\text{low}} + e + \gamma$$
$$e \sim \mathcal{N}(0, 1), \delta, \gamma \sim \mathcal{N}(0, 0.1).$$

Let $\pi(x) = \text{round}\left(\min(\max(1.5x + 5, 0), 9)\right)$. $\pi$ is a transformation function that maps inputs to an integer between 0 and 9. Let RandomImage($d$) be a function which selects a random MNIST image from the class of images corresponding to digit $d$. The three high-dimensional scenarios are:

$$\text{MNIST}_Z : x = x^{\text{low}}, z = \text{RandomImage}(\pi(z_1^{\text{low}}))$$
$$\text{MNIST}_X : x = \text{RandomImage}(\pi(x^{\text{low}})), z = z^{\text{low}}$$
$$\text{MNIST}_{XZ} : x = \text{RandomImage}(\pi(x^{\text{low}})), z = \text{RandomImage}(\pi(z_1^{\text{low}})).$$

We use the function $g_0(x) = |x|$ to compare with Bennett et al. [2019] but in general, the other functional forms described above can also be used. Similar to Bennett et al. [2019] we normalize the data so that $y$ has zero mean and unit standard deviation.

We evaluate the performance of our AGMM and KLayerTrained estimators on these 3 data-generating processes with 20,000 train samples and 2,000 test samples and compare their performance to that achieved when we evaluate Bennett et al. [2019]'s code (performance is measured by the average mean squared error of the predictions on test data).

**Setup** We describe more details about our experimental setup for the MNIST experiments here. We run 10 Monte-Carlo runs of each experiment and report the average MSE and the standard deviation in the MSE achieved.

**Architectures** We use a 4-layer convolutional architecture in all cases where the input to the network is an image. This consists of 2 convolutional layers with a 3x3 kernel followed by two fully connected layers with 9216 and 512 hidden units respectively. A ReLU activation is applied after each layer. Along with that, a max-pooling operation is applied after the first two convolutional layers and a dropout operation (with dropout probability 0.1) is applied before each fully connected layer. When the instrument or treatment is low-dimensional we use a 2 layer fully connected neural network with 200 neurons in the hidden layer along with the dropout function as before. All networks use ReLU as the activation function.

**Early Stopping** We utilize the early stopping procedure proposed in Bennett et al. [2019] which works as follows. In addition to the 20,000 training samples, 10,000 samples are used for preparing a set of candidate adversary functions prior to training. During training at each epoch, the maximum error incurred by the learner against the candidates in this pre-computed list is recorded. The early stopping selects the model whose maximum error as computer above is the smallest.

**Hyper-Parameters** We use a batch size of 100 samples, and run for 200 epochs where an epoch is defined as one full pass over the train set. We have as hyper-parameters learning rates for the learner and adversary networks, the regularization terms for the weights of the learner and the adversary, and a regularization term on the norm of the output of the adversary network. For the MNIST$_x$ experiment, we saw best results when the weight penalizations on both the learner and the adversary were set to very small values as compared to the other two experiments.

## K   Proofs from Section 3 and Appendix C

### K.1   Preliminary Lemmas

**Lemma 15.** *Let $f_h$, be any test function that satisfies: $\|f_h - T(h_* - h)\|_2 \leq \epsilon$ and let*

$$\Psi(h, f) := \mathbb{E}[\psi(y\,;\, h(x))\, f(z)].$$

*Then:*

$$\frac{1}{\|f_h\|_2}\left(\Psi\left(h, f_h\right) - \Psi(h_*, f_h)\right) \geq \|T(h - h_*)\|_2 - 2\epsilon_n$$

*Proof.* Let $f_h^* = T(h_* - h)$ and observe that by the tower law of expectations:

$$\frac{1}{\|f_h\|_2}(\Psi(h, f_h) - \Psi(h_*, f_h)) = \frac{\mathbb{E}[(h_* - h)(x)\, f_h(z)]}{\|f_h\|_2} = \frac{\mathbb{E}[f_h^*(z)\, f_h(z)]}{\|f_h\|_2}$$

However, observe that by the Cauchy-Schwarz inequality we have:

$$
\begin{aligned}
\mathbb{E}[f_h^*(Z)\, f_h(Z)] &= \mathbb{E}[f_h(Z)^2] + \mathbb{E}[f_h(Z)(f_h^*(Z) - f_h(Z))] \geq \|f_h\|_2^2 - |\mathbb{E}[f_h(Z)(f_h^*(Z) - f_h(Z))]|\\
&\geq \|f_h\|_2^2 - \sqrt{\mathbb{E}[f_h(Z)^2]}\sqrt{\mathbb{E}[(f_h^*(Z) - f_h(Z))^2]}\\
&\geq \|f_h\|_2^2 - \|f_h\|_2 \|f_h^* - f_h\|_2\\
&\geq \|f_h\|_2^2 - \epsilon_n \|f_h\|_2
\end{aligned}
$$

Thus we have:

$$\frac{1}{\|f_h\|_2}(\Psi(h, f_h) - \Psi(h_*, f_h)) \geq \|f_h\|_2 - \epsilon_n$$

Finally, by a triangle inequality,

$$\|f_h\|_2 \geq \|f_h^*\|_2 - \|f_h^* - f_h\|_2 \geq \|f_h^*\|_2 - \epsilon_n.$$

Hence, we can conclude that:

$$\frac{1}{\|f_h\|_2}(\Psi(h, f_h) - \Psi(h_*, f_h)) \geq \|f_h^*\|_2 - 2\epsilon_n = \|T(h - h_*)\|_2 - 2\epsilon_n$$

$\square$

### K.2   Proof of Theorem 1

*Proof.* For convenience let:

$$\Psi(h, f) := \mathbb{E}[\psi(y\,;\, h(x))\, f(z)] = \mathbb{E}[T(h_0 - h)(z)\, f(z)] \quad \text{(by conditional moment restriction)}$$

$$\Psi_n(h, f) := \frac{1}{n}\sum_{i=1}^{n}\psi(y_i\,;\, h(x_i))\, f(z_i)$$

Moreover, for our choice of $\delta$ as described in the statement of the theorem, let:

$$
\begin{aligned}
\mathcal{H}_B &:= \left\{h \in \mathcal{H} : \|h\|_{\mathcal{H}}^2 \leq B\right\}\\
\mathcal{F}_U &:= \left\{f \in \mathcal{F} : \|f\|_{\mathcal{F}}^2 \leq U\right\}
\end{aligned}
$$

Moreover, let:

$$\Psi_n^\lambda(h, f) = \Psi_n(h, f) - \lambda\left(\|f\|_{\mathcal{F}}^2 + \frac{U}{\delta^2}\|f\|_{2,n}^2\right)$$

$$\Psi^\lambda(h, f) = \Psi(h, f) - \lambda\left(\frac{2}{3}\|f\|_{\mathcal{F}}^2 + \frac{U}{\delta^2}\|f\|_2^2\right)$$

Thus our estimate can be written as:

$$\hat{h} := \arg\min_{h \in \mathcal{H}} \sup_{f \in \mathcal{F}} \Psi_n^\lambda(h, f) + \mu\|h\|_{\mathcal{H}}^2$$

**Relating empirical and population regularization.** As a preliminary observation, we have that by Theorem 14.1 of Wainwright [2019], w.p. $1 - \zeta$:

$$\forall f \in \mathcal{F}_{3U} : \left|\|f\|_{n,2}^2 - \|f\|_2^2\right| \leq \frac{1}{2}\|f\|_2^2 + \delta^2$$

for our choice of $\delta := \delta_n + c_0\sqrt{\frac{\log(c_1/\zeta)}{n}}$, where $\delta_n$ upper bounds the critical radius of $\mathcal{F}_{3U}$ and $c_0, c_1$ are universal constants. Moreover, for any $f$, with $\|f\|_{\mathcal{F}}^2 \geq 3U$, we can consider the function $f\sqrt{3U}/\|f\|_{\mathcal{F}}$, which also belongs to $\mathcal{F}_{3U}$, since $\mathcal{F}$ is star-convex. Thus we can apply the above lemma to this re-scaled function and multiply both sides by $\|f\|_{\mathcal{F}}^2/(3U)$, leading to:

$$\forall f \in \mathcal{F} \text{ s.t. } \|f\|_{\mathcal{F}}^2 \geq 3U : \left|\|f\|_{n,2}^2 - \|f\|_2^2\right| \leq \frac{1}{2}\|f\|_2^2 + \delta^2 \frac{\|f\|_{\mathcal{F}}^2}{3U}$$

Thus overall, we have:

$$\forall f \in \mathcal{F} : \left|\|f\|_{n,2}^2 - \|f\|_2^2\right| \leq \frac{1}{2}\|f\|_2^2 + \delta^2 \max\left\{1, \frac{\|f\|_{\mathcal{F}}^2}{3U}\right\} \tag{16}$$

Thus we have that w.p. $1 - \zeta$:

$$
\begin{aligned}
\forall f \in \mathcal{F} : \|f\|_{\mathcal{F}}^2 + \frac{U}{\delta^2}\|f\|_{2,n}^2 &\geq \|f\|_{\mathcal{F}}^2 + \frac{U}{\delta^2}\left(\|f\|_2^2 - \delta^2 \max\left\{1, \frac{\|f\|_{\mathcal{F}}^2}{3U}\right\}\right) \\
&\geq \|f\|_{\mathcal{F}}^2 + \frac{U}{\delta^2}\|f\|_2^2 - \max\left\{U, \frac{1}{3}\|f\|_{\mathcal{F}}^2\right\} \\
&\geq \frac{2}{3}\|f\|_{\mathcal{F}}^2 + \frac{U}{\delta^2}\|f\|_2^2 - U
\end{aligned} \tag{17}
$$

**Upper bounding centered empirical sup-loss.** We now argue that the centered empirical sup-loss: $\sup_{f \in \mathcal{F}}(\Psi_n(\hat{h}, f) - \Psi_n(h_*, f))$ is small. By the definition of $\hat{h}$:

$$\sup_{f \in \mathcal{F}} \Psi_n^\lambda(\hat{h}, f) \leq \sup_{f \in \mathcal{F}} \Psi_n^\lambda(h_*, f) + \mu\left(\|h_*\|_{\mathcal{H}}^2 - \|\hat{h}\|_{\mathcal{H}}^2\right) \tag{18}$$

By Lemma 7 of Foster and Syrgkanis [2019], the fact that $\phi(y; h_*(x))f(z)$ is 2-Lipschitz with respect to $f(z)$ (since $y \in [-1, 1]$ and $\|h_*\|_\infty \in [-1, 1]$) and by our choice of $\delta := \delta_n + c_0\sqrt{\frac{\log(c_1/\zeta)}{n}}$, where $\delta_n$ is an upper bound on the critical radius of $\mathcal{F}_{3U}$, w.p. $1 - \zeta$:

$$\forall f \in \mathcal{F}_{3U} : |\Psi_n(h_*, f) - \Psi(h_*, f)| \leq 36\delta\|f\|_2 + 36\delta^2$$

Thus, if $\|f\|_{\mathcal{F}} \geq \sqrt{3U}$, we can apply the latter inequality for the function $f\sqrt{3U}/\|f\|_{\mathcal{F}}$, which falls in $\mathcal{F}_{3U}$, and then multiply both sides by $\|f\|_{\mathcal{F}}/\sqrt{3U}$ to get:

$$\forall f \in \mathcal{F} : |\Psi_n(h_*, f) - \Psi(h_*, f)| \leq 36\delta\|f\|_2 + 36\delta^2 \max\left\{1, \frac{\|f\|_{\mathcal{F}}}{\sqrt{3U}}\right\} \tag{19}$$

By Equations (17) and (19), we have that w.p. $1 - 2\zeta$:

$$
\begin{aligned}
\sup_{f \in \mathcal{F}} \Psi_n^\lambda(h_*, f) &= \sup_{f \in \mathcal{F}}\left(\Psi_n(h_*, f) - \lambda\left(\|f\|_{\mathcal{F}}^2 + \frac{U}{\delta^2}\|f\|_{2,n}^2\right)\right) \\
&\leq \sup_{f \in \mathcal{F}}\left(\Psi(h_*, f) + 36\delta^2 + \frac{36\delta^2}{\sqrt{3U}}\|f\|_{\mathcal{F}} + 36\delta\|f\|_2 - \lambda\left(\|f\|_{\mathcal{F}}^2 + \frac{U}{\delta^2}\|f\|_{2,n}^2\right)\right) \\
&\leq \sup_{f \in \mathcal{F}}\left(\Psi(h_*, f) + 36\delta^2 + \frac{36\delta^2}{\sqrt{3U}}\|f\|_{\mathcal{F}} + 36\delta\|f\|_2 - \lambda\left(\frac{2}{3}\|f\|_{\mathcal{F}}^2 + \frac{U}{\delta^2}\|f\|_2^2\right) + \lambda U\right) \\
&\leq \sup_{f \in \mathcal{F}} \Psi^{\lambda/2}(h_*, f) + 36\delta^2 + \lambda U \\
&\quad + \sup_{f \in \mathcal{F}}\left(\frac{36\delta^2}{\sqrt{3U}}\|f\|_{\mathcal{F}} - \frac{\lambda}{2}\frac{2}{3}\|f\|_{\mathcal{F}}^2\right) + \sup_{f \in \mathcal{F}}\left(36\delta\|f\|_2 - \frac{\lambda}{2}\frac{U}{\delta^2}\|f\|_2^2\right)
\end{aligned}
$$

Moreover, observe that for any norm $\|\cdot\|$ and any constants $a, b > 0$:

$$\sup_{f \in \mathcal{F}} \left( a\|f\| - b\|f\|^2 \right) \leq \frac{a^2}{4b}$$

Thus if we assume that $\lambda \geq \delta^2/U$, we have:

$$\sup_{f \in \mathcal{F}} \left( \delta^2 \frac{36}{\sqrt{3U}} \|f\|_{\mathcal{F}} - \frac{\lambda}{2} \frac{2}{3} \|f\|_{\mathcal{F}}^2 \right) \leq \frac{36^2}{4} \frac{\delta^4}{U\lambda} \leq 324\delta^2$$

$$\sup_{f \in \mathcal{F}} \left( 36\delta \|f\|_2 - \frac{\lambda}{2} \frac{U}{\delta^2} \|f\|_2^2 \right) \leq \frac{36^2 \delta^4}{2\lambda U} \leq 648\delta^2$$

Thus we have:

$$\sup_{f \in \mathcal{F}} \Psi_n^\lambda(h_*, f) \leq \sup_{f \in \mathcal{F}} \Psi^{\lambda/2}(h_*, f) + \lambda U + O(\delta^2)$$

Moreover:

$$\begin{aligned}
\sup_{f \in \mathcal{F}} \Psi_n^\lambda(\hat{h}, f) = & \sup_{f \in \mathcal{F}} \left( \Psi_n(\hat{h}, f) - \Psi_n(h_*, f) + \Psi_n(h_*, f) - \lambda \left( \|f\|_{\mathcal{F}}^2 + \frac{U}{\delta^2} \|f\|_{2,n}^2 \right) \right) \\
\geq & \sup_{f \in \mathcal{F}} \left( \Psi_n(\hat{h}, f) - \Psi_n(h_*, f) - 2\lambda \left( \|f\|_{\mathcal{F}}^2 + \frac{U}{\delta^2} \|f\|_{2,n}^2 \right) \right) \\
& + \inf_{f \in \mathcal{F}} \left( \Psi_n(h_*, f) + \lambda \left( \|f\|_{\mathcal{F}}^2 + \frac{U}{\delta^2} \|f\|_{2,n}^2 \right) \right) \\
= & \sup_{f \in \mathcal{F}} \left( \Psi_n(\hat{h}, f) - \Psi_n(h_*, f) - 2\lambda \left( \|f\|_{\mathcal{F}}^2 + \frac{U}{\delta^2} \|f\|_{2,n}^2 \right) \right) - \sup_{f \in \mathcal{F}} \Psi_n^\lambda(h_*, f)
\end{aligned}$$

Combining this with Equation (18) yields:

$$\begin{aligned}
\sup_{f \in \mathcal{F}} \left( \Psi_n(\hat{h}, f) - \Psi_n(h_*, f) - 2\lambda \left( \|f\|_{\mathcal{F}}^2 + \frac{U}{\delta^2} \|f\|_{2,n}^2 \right) \right) \leq & \ 2 \sup_{f \in \mathcal{F}} \Psi_n^\lambda(h_*, f) + \mu \left( \|h_*\|_{\mathcal{H}}^2 - \|\hat{h}\|_{\mathcal{H}}^2 \right) \\
\leq & \ O(\delta^2) + \lambda U + 2 \sup_{f \in \mathcal{F}} \Psi^{\lambda/2}(h_*, f) \\
& + \mu \left( \|h_*\|_{\mathcal{H}}^2 - \|\hat{h}\|_{\mathcal{H}}^2 \right)
\end{aligned}$$

**Lower bounding centered empirical sup-loss.** For any $h$, let

$$f_h := \arg\inf_{f \in \mathcal{F}_{L^2 \|h - h_*\|_{\mathcal{H}}^2}} \|f - T(h_* - h)\|_2.$$

and observe that by our assumption, for any $h \in \mathcal{H}$: $\|f_h - T(h_* - h)\|_2 \leq \eta_n$.

Suppose that $\|f_{\hat{h}}\|_2 \geq \delta$ and let $r = \frac{\delta}{2\|f_{\hat{h}}\|_2} \in [0, 1/2]$. Then observe that since $f_{\hat{h}} \in \mathcal{F}_{L\|h - h_*\|_{\mathcal{H}}}$ and $\mathcal{F}$ is star-convex, we also have that $r f_h \in \mathcal{F}_{L\|h - h_*\|_{\mathcal{H}}}$. Thus we can lower bound the supremum by its evaluation at $r f_h$:

$$\begin{aligned}
\sup_{f \in \mathcal{F}} \left( \Psi_n(\hat{h}, f) - \Psi_n(h_*, f) - 2\lambda \left( \|f\|_{\mathcal{F}}^2 + \frac{U}{\delta^2} \|f\|_{2,n}^2 \right) \right) \geq & \ r(\Psi_n(\hat{h}, f_{\hat{h}}) - \Psi_n(h_*, f_{\hat{h}})) \\
& - 2\lambda r^2 \left( \|f_{\hat{h}}\|_{\mathcal{F}}^2 + \frac{U}{\delta^2} \|f_{\hat{h}}\|_{2,n}^2 \right)
\end{aligned}$$

Moreover, since $\delta_n$ upper bounds the critical radius of $\mathcal{F}_{3U}$, $\|f_{\hat{h}}\|_{\mathcal{F}} \leq L\|\hat{h} - h_*\|_{\mathcal{H}}$ and by Equation (16):

$$\begin{aligned}
r^2 \left( \|f_{\hat{h}}\|_{\mathcal{F}}^2 + \frac{U}{\delta^2} \|f_{\hat{h}}\|_{2,n}^2 \right) \leq & \ \|f_{\hat{h}}\|_{\mathcal{F}}^2 + \frac{U}{\delta^2} r^2 \|f_{\hat{h}}\|_{2,n}^2 \\
\leq & \ \|f_{\hat{h}}\|_{\mathcal{F}}^2 + \frac{U}{\delta^2} r^2 \left( 2\|f_{\hat{h}}\|_2^2 + \delta^2 + \delta^2 \frac{\|f_{\hat{h}}\|_{\mathcal{F}}^2}{3U} \right) \\
\leq & \ \frac{4}{3} L^2 \|h - h_*\|_{\mathcal{H}}^2 + \frac{U}{2} + \frac{U}{4} \leq 2L^2 \|h - h_*\|_{\mathcal{H}}^2 + U
\end{aligned}$$

Thus we get:

$$\sup_{f \in \mathcal{F}} \left( \Psi_n(\hat{h}, f) - \Psi_n(h_*, f) - 2\lambda \left( \|f\|_{\mathcal{F}}^2 + \frac{U}{\delta^2} \|f\|_{2,n}^2 \right) \right) \geq r(\Psi_n(\hat{h}, f_{\hat{h}}) - \Psi_n(h_*, f_{\hat{h}}))$$
$$- 4\lambda L^2 \|h - h_*\|_{\mathcal{H}}^2 - 2\lambda U$$

Observe that:

$$\Psi_n(h, f_h) - \Psi_n(h_*, f_h) = \frac{1}{n} \sum_{i=1}^n (h_*(x_i) - h(x_i)) f_h(h_* - h)(z_i)$$
$$\Psi(h, f_h) - \Psi(h_*, f_h) = \mathbb{E}[(h_*(x_i) - h(x_i)) f_h(z_i)]$$

By Lemma 7 of Foster and Syrgkanis [2019], and by our choice of $\delta := \delta_n + c_0 \sqrt{\frac{\log(c_1/\zeta)}{n}}$, where $\delta_n$ upper bounds the critical radius of $\mathcal{G}$, we have that w.p. $1 - \zeta$: $\forall h$, such that $h - h_* \in \mathcal{H}_B$

$$|(\Psi_n(h, f_h) - \Psi_n(h_*, f_h)) - (\Psi(h, f_h) - \Psi(h_*, f_h))| \leq 18\delta \sqrt{\mathbb{E}[(h_*(X) - h(X))^2 f_h(Z)^2]} + 18\delta^2$$
$$\leq 18\delta \sqrt{\mathbb{E}[f_h(Z)^2]} + 18\delta^2$$
$$= 18\delta \|f_h\|_2 + 18\delta^2 \qquad (20)$$

where in the second inequality we used the fact that $h - h_*$ has range in $[-1, 1]$, when $\|h - h_*\|_{\mathcal{H}} \leq B$. If $h - h_*$ has $\|h - h_*\|_{\mathcal{H}}^2 \geq B$, we can apply the latter for $(h - h_*)\sqrt{B}/\|h - h_*\|_{\mathcal{H}}$ and multiply both sides by $\|h - h_*\|_{\mathcal{H}}^2/B$:

$$|(\Psi_n(h, f_h) - \Psi_n(h_*, f_h)) - (\Psi(h, f_h) - \Psi(h_*, f_h))| \leq 18\delta \|f_h\|_2 \frac{\|h - h_*\|_{\mathcal{H}}}{\sqrt{B}} + 18\delta^2 \frac{\|h - h_*\|_{\mathcal{H}}^2}{B}$$

Thus we have that for all $h \in \mathcal{H}$:

$$|(\Psi_n(h, f_h) - \Psi_n(h_*, f_h)) - (\Psi(h, f_h) - \Psi(h_*, f_h))| \leq \left( 18\delta \|f_h\|_2 + 18\delta^2 \right) \max \left\{ 1, \frac{\|h - h_*\|_{\mathcal{H}}^2}{B} \right\}$$

Applying the latter bound for $h := \hat{h}$ and multiplying by $r := \frac{\delta}{2\|f_{\hat{h}}\|_2} \in [0, 1/2]$, yields:

$$r(\Psi_n(\hat{h}, f_{\hat{h}}) - \Psi_n(h_*, f_{\hat{h}})) \geq r(\Psi(\hat{h}, f_{\hat{h}}) - \Psi(h_*, f_{\hat{h}})) - 18\delta^2 \max \left\{ 1, \frac{\|h - h_*\|_{\mathcal{H}}^2}{B} \right\}$$

Moreover, observe that by Lemma 15 and the fact that $\|f_{\hat{h}} - T(h_* - \hat{h})\|_2 \leq \eta_n$, we have:

$$r(\Psi(\hat{h}, f_{\hat{h}}) - \Psi(h_*, f_{\hat{h}})) \geq \frac{\delta}{2} \|T(h_* - \hat{h})\|_2 - \delta\eta_n$$

Thus we have:

$$\sup_{f \in \mathcal{F}} \left( \Psi_n(\hat{h}, f) - \Psi_n(h_*, f) - 2\lambda \left( \|f\|_{\mathcal{F}}^2 + \frac{U}{\delta^2} \|f\|_{2,n}^2 \right) \right) \geq \frac{\delta}{2} \|T(h_* - h)\|_2 - \delta\eta_n$$
$$- 27\delta^2 \max \left\{ 1, \frac{\|h - h_*\|_{\mathcal{H}}^2}{B} \right\}$$
$$- 4\lambda L^2 \|h - h_*\|_{\mathcal{H}}^2 - 2\lambda U$$

**Combining upper and lower bound.** Combining the upper and lower bound on the centered population sup-loss we get that w.p. $1 - 3\zeta$: either $\|f_{\hat{h}}\|_2 \leq \delta$ or:

$$\frac{\delta}{2} \|T(\hat{h} - h_*)\|_2 \leq O(\delta^2 + \delta\eta_n + \lambda U) + 2 \sup_{f \in \mathcal{F}} \Psi^{\lambda/2}(h_*, f)$$
$$+ 27\delta^2 \frac{\|\hat{h} - h_*\|_{\mathcal{H}}^2}{B} + 4\lambda L^2 \|\hat{h} - h_*\|_{\mathcal{H}}^2 + \mu \left( \|h_*\|_{\mathcal{H}}^2 - \|\hat{h}\|_{\mathcal{H}}^2 \right)$$

We now control the last part. Since $\lambda \geq \delta^2/U$, the latter is upper bounded by:

$$\lambda \left(\frac{27U}{B} + 4L^2\right) \|\hat{h} - h_*\|_{\mathcal{H}}^2 + \mu \left(\|h_*\|_{\mathcal{H}}^2 - \|\hat{h}\|_{\mathcal{H}}^2\right) \leq 2\lambda \left(\frac{27U}{B} + 4L^2\right) \left(\|\hat{h}\|_{\mathcal{H}}^2 + \|h_*\|_{\mathcal{H}}^2\right)$$
$$+ \mu \left(\|h_*\|_{\mathcal{H}}^2 - \|\hat{h}\|_{\mathcal{H}}^2\right)$$

Since $\mu \geq 2\lambda \left(\frac{27U}{B} + 4L^2\right)$, the latter is upper bounded by:

$$\left(2\lambda \left(\frac{27U}{B} + 4L^2\right) + \mu\right) \|h_*\|_{\mathcal{H}}^2$$

Thus as long as $\mu \geq 2\lambda \left(\frac{27U}{B} + 4L^2\right)$ and $\lambda \geq \delta^2/U$, we have:

$$\frac{\delta}{2}\|T(\hat{h} - h_*)\|_2 \leq O(\delta^2 + \delta\eta_n + \lambda U) + 2 \sup_{f \in \mathcal{F}} \Psi^{\lambda/2}(h_*, f) + \left(2\lambda \left(\frac{27U}{B} + 4L^2\right) + \mu\right) \|h_*\|_{\mathcal{H}}^2$$

Dividing over by $\delta$ and treating $L, U, B$ as constants, we get:

$$\|T(\hat{h} - h_*)\|_2 \leq O(\delta + \eta_n + \|h_*\|_{\mathcal{H}}^2 (\lambda/\delta + \mu/\delta)) + \frac{2}{\delta} \sup_{f \in \mathcal{F}} \Psi^{\lambda/2}(h_*, f)$$

Thus either $\|f_{\hat{h}}\| \leq \delta$ or the latter inequality holds. However, in the case when $\|f_{\hat{h}}\| \leq \delta$, we have by a triangle inequality that: $\|T(\hat{h} - h_*)\|_2 \leq \delta + \eta_n$. Thus in any case the latter inequality holds.

**Upper bounding population sup-loss at minimum.** Let $f_0 = T(h_0 - h_*)$ and observe that:

$$\sup_{f \in \mathcal{F}} \Psi^{\lambda/2}(h_*, f) = \sup_{f \in \mathcal{F}} \mathbb{E}[f_0(z) f(z)] - \frac{\lambda}{2}\left(\frac{2}{3}\|f\|_{\mathcal{F}}^2 + \frac{U}{\delta^2}\|f\|_2^2\right) \leq \sup_{f \in \mathcal{F}} \mathbb{E}[f_0(z) f(z)] - \frac{\lambda}{2}\frac{U}{\delta^2}\|f\|_2^2$$

Then by the Cauchy-Schwarz inequality and since $\lambda \geq \delta^2/U$:

$$\sup_{f \in \mathcal{F}} \mathbb{E}[f_0(z) f(z)] - \lambda\frac{U}{\delta^2}\|f\|_2^2 \leq \sup_{f \in \mathcal{F}} \|f_0\|_2\|f\|_2 - \frac{\lambda}{2}\frac{U}{\delta^2}\|f\|_2^2 \leq \frac{\|f_0\|_2^2}{2\lambda U}\delta^2 \leq \frac{\|f_0\|^2}{2}$$

**Concluding.** Concluding we get that w.p. $1 - 3\zeta$:

$$\|T(\hat{h} - h_*)\|_2 \leq O(\delta + \eta_n + \|h_*\|_{\mathcal{H}}^2 (\lambda/\delta + \mu/\delta)) + \frac{\|T(h_* - h_0)\|_2^2}{\delta}$$

By a triangle inequality:

$$\|T(\hat{h} - h_0)\|_2 \leq \|T(\hat{h} - h_*)\|_2 + \|T(h_* - h_0)\|_2$$
$$\leq O(\delta + \eta_n + \|h_*\|_{\mathcal{H}}^2 (\lambda/\delta + \mu/\delta)) + \frac{\|T(h_* - h_0)\|_2^2}{\delta} + \|T(h_* - h_0)\|_2$$

$\square$

### K.3   Proof of Theorem 2

*Proof.* By the definition of $\hat{h}$:

$$0 \leq \sup_f \Psi_n(\hat{h}, f) \leq \sup_f \Psi_n(h_0, f) + \lambda \left(\|h_0\|_{\mathcal{H}} - \|\hat{h}\|_{\mathcal{H}}\right)$$

Let $\mathcal{F}_U^i = \{f \in \mathcal{F}_i : \|f\|_{\mathcal{F}} \leq U\}$ and $\delta_{n,\zeta} = \max_{i=1}^d 2\,\mathcal{R}(\mathcal{F}_U^i) + c_0\sqrt{\frac{\log(c_1/\zeta)}{n}}$ for some universal constants $c_0, c_1$. By Theorem 26.5 and 26.9 of Shalev-Shwartz and Ben-David [2014], and since $\mathcal{F}_U^i$ is a symmetric class and $\sup_{y \in \mathcal{Y}, x \in \mathcal{X}} |y - h_0(x)| \leq 2$, w.p. $1 - \zeta$:

$$f \in \mathcal{F}_U^i \, |\Psi_n(h_0, f) - \Psi(h_0, f)| \leq \delta_{n,\zeta}$$

Since $\Psi(h_0, f) = 0$ for all $f$, we have that, w.p. $1 - \zeta$:

$$\|\hat{h}\|_{\mathcal{H}} \leq \|h_0\|_{\mathcal{H}} + \delta_{n,\zeta}/\lambda$$

Let $B_{n,\lambda,\zeta} = (\|h_0\|_{\mathcal{H}} + \delta_{n,\zeta}/\lambda)^2$. Then if we let $\epsilon_{n,\lambda,\zeta} = \max_i \mathcal{R}(\mathcal{H}_{B_{n,\lambda,\zeta}} \cdot \mathcal{F}_U^i) + c_0 \sqrt{\frac{\log(c_1/\zeta)}{n}}$ for some universal constants $c_0, c_1$.

$$\forall h \in \mathcal{H}_{B_{n,\lambda,\zeta}}, f \in \mathcal{F}_U^i \ |\Psi_n(h,f) - \Psi(h,f)| \leq \delta_{n,\zeta}$$

By a union bound over the $d$ function classes composing $\mathcal{F}$, we have that w.p. $1 - 2\zeta$:

$$\sup_{f \in \mathcal{F}_U} \Psi_n(h_0, f) \leq \sup_{f \in \mathcal{F}_U} \Psi(h_0, f) + \delta_{n,\zeta/d} = \delta_{n,\zeta/d}$$

and

$$\sup_{f \in \mathcal{F}_U} \Psi_n(\hat{h}, f) \geq \sup_{f \in \mathcal{F}_U} \Psi(\hat{h}, f) - \epsilon_{n,\zeta/d}$$

Since, by assumption, for any $h \in \mathcal{H}_{B_{n,\lambda,\zeta}}$, $\frac{T(h_0 - h)}{\|T(h_0 - h)\|_2} \in \text{span}_R(\mathcal{F}_U)$, we have $\frac{T(h_0 - h)}{\|T(h_0 - h)\|_2} = \sum_{i=1}^{p} w_i f_i$, with $p < \infty$, $\|w\|_1 \leq \kappa$ and $f_i \in \mathcal{F}_U$. Thus we have:

$$\begin{aligned}
\sup_{f \in \mathcal{F}_U} \Psi(\hat{h}, f) &\geq \frac{1}{\kappa} \sum_{i=1}^{p} w_i \Psi(\hat{h}, f_i) = \frac{1}{\kappa} \Psi\left(\hat{h}, \sum_i w_i f_i\right) \\
&= \frac{1}{\kappa} \frac{1}{\|T(h_0 - \hat{h})\|_2} \Psi(\hat{h}, T(h_0 - \hat{h})) \\
&= \frac{1}{\kappa} \frac{1}{\|T(h_0 - \hat{h})\|_2} \mathbb{E}[T(h_0 - \hat{h})(z)^2] \\
&= \frac{1}{\kappa} \|T(h_0 - \hat{h})\|_2
\end{aligned}$$

Combining all the above we have:

$$\|T(h_0 - \hat{h})\|_2 \leq \kappa \left(\epsilon_{n,\lambda,\zeta/d} + \delta_{n,\zeta/d} + \lambda \left(\|h_0\|_{\mathcal{H}} - \|\hat{h}\|_{\mathcal{H}}\right)\right)$$

Moreover, since functions in $\mathcal{H}$ and $\mathcal{F}$ are bounded in $[-1, 1]$, we have that the function $h \cdot f$ is 1-Lipschitz with respect to the vector of functions $(h, f)$. Thus we can apply a vector version of the contraction inequality Maurer [2016] to get that:

$$\mathcal{R}(\mathcal{H}_{B_{n,\lambda,z}} \cdot \mathcal{F}_U^i) \leq 2 \left(\mathcal{R}(\mathcal{H}_{B_{n,\lambda,z}}) + \mathcal{R}(\mathcal{F}_U^i)\right)$$

Finally, we have that since $\mathcal{H}$ is star-convex:

$$\mathcal{R}(\mathcal{H}_{B_{n,\lambda,z}}) \leq \sqrt{B_{n,\lambda,z}} \mathcal{R}(\mathcal{H}_1)$$

Leading the final bound of:

$$\|T(h_0 - \hat{h})\|_2 \leq \kappa \left(2 \left(\|h_0\|_{\mathcal{H}} + \delta_{n,\zeta}/\lambda\right) \mathcal{R}(\mathcal{H}_1) + 2 \max_{i=1}^{d} \mathcal{R}(\mathcal{F}_U^i) + c_0 \sqrt{\frac{\log(c_1 d/\zeta)}{n}} + \lambda \left(\|h_0\|_{\mathcal{H}} - \|\hat{h}\|_{\mathcal{H}}\right)\right)$$

Since $\|h_0\|_{\mathcal{H}} \leq R$ and $\lambda \geq \delta_{n,\zeta}$, we get the result. $\qquad\square$

### K.4 Proof of Theorem 6

The proof is identical to that of Theorem 1 with small modifications. Hence we solely mention these modifications and omit the full proof.

The only part that we change is instead of the set of Equations (20), we instead view $\psi(y; h(x)) f_h(z)$ as a function of the vector valued function $(x, z) \rightarrow (h(x), f_h(z))$. Then we note that since $h, f$ take values in $[-1, 1]$ and $y \in [-1, 1]$, we note that this function 2-Lipschitz with respect to this vector. Then we can apply Lemma 7 of Foster and Syrgkanis [2019], and by our choice of $\delta := \delta_n + c_0 \sqrt{\frac{\log(c_1/\zeta)}{n}}$, where $\delta_n$ upper bounds the critical radius of $\text{star}(\mathcal{H}_B - h_*)$ and $\text{star}(T(\mathcal{H}_B - h_*))$, we have that w.p. $1 - \zeta$: $\forall h \in \mathcal{H}_B$:

$$|(\Psi_n(h, f_h) - \Psi_n(h_*, f_h)) - (\Psi(h, f_h) - \Psi(h_*, f_h))| \leq 36\delta (\|h - h_*\|_2 + \|f_h\|_2) + 18\delta^2$$

Subsequently, we can follow identical steps to conclude that w.p. $1 - 3\zeta$, either $\|f_{\hat{h}}\|_2 \leq \delta$ or:

$$\|T(\hat{h} - h_0)\|_2 \leq O\left(\delta + \delta \frac{\|\hat{h} - h_*\|_2}{\|f_h\|_2} + \eta_n + \|h_*\|_{\mathcal{H}}^2 \left(\lambda/\delta + \mu/\delta\right) + \frac{\|T(h_* - h_0)\|_2^2}{\delta}\right)$$

Subsequently, by the measure of ill-posedness we have:

$$\|\hat{h} - h_*\|_2 \leq \tau \|T(\hat{h} - h_*)\|_2$$

Moreover, observe that when $\|f_{\hat{h}}\|_2 \geq \delta \geq 3\eta_n$, then we have by a triangle inequality that:

$$\|T(h - h_*)\|_2 \geq \|f_{\hat{h}}\|_2 - \eta_n \geq 2\eta_n$$

and:

$$\|f_{\hat{h}}\|_2 \geq \|T(h - h_*)\|_2 - \eta_n \geq \frac{1}{2}\|T(h - h_*)\|_2$$

Thus we get that:

$$\frac{\|\hat{h} - h_*\|_2}{\|f_h\|_2} \leq \tau \frac{\|T(h - h_*)\|_2}{\|f_{\hat{h}}\|_2} \leq 2\tau$$

Thus overall we have that either $\|f_{\hat{h}}\|_2 \leq \delta$ or:

$$\|\hat{h} - h_*\|_2 \leq O\left(\tau \left(\tau\delta + \eta_n + \|h_*\|_{\mathcal{H}}^2 \left(\lambda/\delta + \mu/\delta\right) + \frac{\|T(h_* - h_0)\|_2^2}{\delta}\right)\right)$$

$$\leq O\left(\tau \left(\tau\delta + \eta_n + \|h_*\|_{\mathcal{H}}^2 \left(\lambda/\delta + \mu/\delta\right) + \frac{\|h_* - h_0\|_2^2}{\delta}\right)\right) \qquad (21)$$

where the last inequality follows by that fact that Jensen's inequality implies that $\|T(h_* - h_0)\|_2 \leq \|h_* - h_0\|_2$. Moreover, if $\|f_{\hat{h}}\|_2 \leq \delta$, then by a triangle inequality that $\|T(\hat{h} - h_*)\|_2 \leq \delta + \eta_n$, which, subsequently implies by invoking the bound on the ill-posedness measure that: $\|\hat{h}_* - h\| \leq \tau(\delta + \eta_n)$. Thus in any case the bound in Equation (21) holds. Choosing $h_* := \arg\inf_{h \in \mathcal{H}_B} \|h - h_0\|_2$, yields the result.

## L  Proofs from Section 4 and Appendix E

### L.1  Proof of Proposition 9

*Proof.* Since $\|f\|_{2,n}$ depends on $f$ only through the values $f(z_1), \dots, f(z_n)$, and the maximization over $f$ in (11) is the penalized problem

$$\sup_{f \in \mathcal{F}} \frac{1}{n} \sum_{i=1}^{n} \psi(y_i\,;\,h(x_i))\, f(z_i) - \lambda(\tfrac{U}{\delta^2}\|f\|_{2,n}^2 + \|f\|_K^2)$$

for some choice of $\lambda \geq 0$, the generalized representer theorem of [Schölkopf et al., 2001, Thm. 1] implies that an optimal solution of the constrained problem in (11) takes the form

$$f^*(z) = \sum_{i=1}^{n} \alpha_i^* K(z_i, z)$$

for some weight vector $\alpha^* \in \mathbb{R}^n$. Now consider a function

$$f(z) = \sum_{i=1}^{n} \alpha_i K(z_i, z)$$

for any $\alpha \in \mathbb{R}^n$. We have $\|f\|_K^2 = \alpha^\top K_n \alpha$, $f(z_i) = e_i^\top K_n \alpha$, and

$$\|f\|_{2,n}^2 = \frac{1}{n} \sum_{i=1}^{n} f(z_i)^2 = \frac{1}{n} \sum_{i=1}^{n} \alpha^\top K_n e_i e_i^\top K_n \alpha = \frac{1}{n} \alpha^\top K_n^2 \alpha.$$

Thus the penalized problem is equivalent to the finite dimensional maximization problem:

$$\sup_{\alpha \in \mathbb{R}^n} \psi_n^\top K_n \alpha - \lambda \alpha^\top \left( \frac{U}{n\delta^2} K_n + I \right) K_n \alpha$$

by taking the first order condition, the latter has a closed form optimizer of:

$$\alpha^* = \frac{1}{2\lambda} \left( \frac{U}{n\delta^2} K_n + I \right)^{-1} \psi_n$$

and optimal value of:

$$\frac{1}{4\lambda} \psi_n^\top K_n \left( \frac{U}{n\delta^2} K_n + I \right)^{-1} \psi_n = \frac{1}{4\lambda} \psi_n^\top K_n^{1/2} \left( \frac{U}{n\delta^2} K_n + I \right)^{-1} K_n^{1/2} \psi_n$$

where in the last equality we used a classic matrix inverse identity for kernel matrices.[18] □

## L.2   Proof of Proposition 10

*Proof.* By Proposition 9,

$$\hat{h} = \arg\min_{h \in \mathcal{H}} \frac{1}{4\lambda} \psi_n^\top M \psi_n + \mu \|h\|_{K_\mathcal{H}}^2 = \arg\min_{h \in \mathcal{H}} \psi_n^\top M \psi_n + 4\lambda\mu \|h\|_{K_\mathcal{H}}^2 \qquad (22)$$

where $\psi_n = (\frac{1}{n}\psi(y_i \,;\, h(x_i)))_{i=1}^n$. Since the objective of (22) depends only on $h$ only through the values $h(x_1), \ldots, h(x_n)$, and the problem, the generalized representer theorem of [Schölkopf et al., 2001, Thm. 1] implies that an optimal solution of the problem (22) takes the form

$$h^*(x) = \sum_{i=1}^n \alpha_i^* K_\mathcal{H}(x_i, x)$$

for some weight vector $\alpha^* \in \mathbb{R}^n$. Now consider a function

$$h(z) = \sum_{i=1}^n \alpha_i K_\mathcal{H}(z_i, z)$$

for any $\alpha \in \mathbb{R}^n$. We have $\|h\|_{K_\mathcal{H}}^2 = \alpha^\top K_{\mathcal{H},n} \alpha$, $h(z_i) = e_i^\top K_{\mathcal{H},n}\alpha$, and $\psi_n = y - K_{\mathcal{H},n}\alpha$. The problem (22) is therefore equivalent to

$$\min_{\alpha \in \mathbb{R}^n} \; \alpha^\top K_{\mathcal{H},n} M K_{\mathcal{H},n} \alpha - 2y^\top M K_{\mathcal{H},n}\alpha + 4\lambda\mu\,\alpha^\top K_{\mathcal{H},n}\alpha.$$

By [Boyd and Vandenberghe, 2004, Ex. 4.22], this problem is solved by:

$$\alpha^* := \left( K_{\mathcal{H},n} \, M \, K_{\mathcal{H},n} + 4\lambda\,\mu\, K_{\mathcal{H},n} \right)^\dagger K_{\mathcal{H},n} My$$

□

### L.3 Proof of Lemma 11

*Proof.* Under these assumptions we have:

$$\|Th\|_2^2 = a_I^\top V_m a_I - 2 \sum_{i \le m < j} a_i a_j \mathbb{E}[\mathbb{E}[e_i(x) \mid z]\mathbb{E}[e_j(x) \mid z]] + \mathbb{E}\left[\left(\sum_{j>m} a_j \mathbb{E}[e_j(x) \mid z]\right)^2\right]$$

$$\ge a_I^\top V_m a_I - 2 \sum_{i \le m < j} |a_i a_j| \,|\mathbb{E}[\mathbb{E}[e_i(x) \mid z]\mathbb{E}[e_j(x) \mid z]]|$$

$$\ge a_I^\top V_m a_I - 2 \sum_{i \le m < j} |a_i a_j| c\,\tau_m$$

$$\ge \tau_m \|a_I\|_2^2 - 2c\,\tau_m \sum_{i \le m} |a_i| \sqrt{\sum_{j>m} a_j^2}$$

$$\ge \tau_m \|a_I\|_2^2 - 2c\,\tau_m \sqrt{\lambda_{m+1} B} \sum_{i \le m} |a_i|$$

$$\ge \tau_m \|a_I\|_2^2 - 2c\,\tau_m \sqrt{\lambda_{m+1} B} \sqrt{\sum_{i \le m} a_i^2}$$

$$\ge \tau_m \|a_I\|_2^2 - 2c\,\tau_m \sqrt{\lambda_{m+1} B} \|a_I\|_2$$

Thus if $\|Th\|_2 \le \delta$, then by solving the above quadratic inequality and using the fact that $(a+b)^2 \le 2a^2 + 2b^2$, we have for all $m$:

$$\|a_I\|_2^2 \le \frac{4\delta^2}{\tau_m} + 4c^2 \lambda_{m+1} B$$

Moreover, observe that by the RKHS norm bound:

$$\|h\|_2^2 = \sum_{j \in J} a_j^2 \le \|a_I\|_2^2 + \lambda_m B$$

Thus we can bound:

$$\tau^*(\delta)^2 = \min_{h:\|Th\|_2^2 \le \delta^2} \|h\|_2 \le \min_{m \in \mathbb{N}_+} \frac{4\delta^2}{\tau_m} + (4c^2 + 1)\lambda_{m+1} B$$

$\square$

## M    Proofs from Section 5 and Appendix F

### M.1    Proof of Corollary 3

*Proof.* Let $\mathcal{H} = \{\langle \theta, x \rangle : \theta \in \mathbb{R}^d\}$ and $\|h\|_{\mathcal{H}} = \|\theta\|_1$. Moreover, suppose that $h_0$ is $s$-sparse. Then if $h \in H_{B_n,\lambda,\varsigma}$, then:

$$\delta_{n,\varsigma}/\lambda + \|\theta_0\|_1 \ge \|\hat{\theta}\|_1 = \|\theta_0 + \nu\|_1 = \|\theta_0 + \nu_S\|_1 + \|\nu_{S^c}\|_1 \ge \|\theta_0\|_1 - \|\nu_S\|_1 + \|\nu_{S^c}\|_1$$

Thus:

$$\|\nu\|_1 \le 2\|\nu_S\|_1 + \delta_{n,\varsigma}/\lambda \le 2\sqrt{s}\|\nu_S\|_2 + \delta_{n,\varsigma}/\lambda \le 2\sqrt{s}\|\nu\|_2 + \delta_{n,\varsigma}/\lambda \le 2\sqrt{\frac{s}{\gamma}\nu^\top V \nu} + \delta_{n,\varsigma}/\lambda$$

Moreover, observe that:

$$\|T(h - h_0)\|_2 = \sqrt{\mathbb{E}[\langle \nu, \mathbb{E}[x \mid z]\rangle^2]} = \sqrt{\nu^\top V \nu}$$

Thus we have:

$$\frac{T(h - h_0)}{\|T(h - h_0)\|_2} = \sum_{i=1}^p \frac{\nu_i}{\sqrt{\nu^\top V \nu}} \mathbb{E}[x_i \mid z]$$

Thus we can write $\frac{T(h-h_0)}{\|T(h-h_0)\|_2}$ as $\sum_{i=1}^p w_i f_i$, with $f_i \in \mathcal{F}_U$ and:

$$\|w\|_1 = \frac{\|\nu\|_1}{\sqrt{\nu^\top V \nu}} \le 2\sqrt{\frac{s}{\gamma}} + \frac{\delta_{n,\varsigma}}{\lambda} \frac{1}{\|T(h-h_0)\|_2}.$$

Thus: $\frac{T(h-h_0)}{\|T(h-h_0)\|_2} \in \mathrm{span}_\kappa(\mathcal{F}_U)$ for $\kappa = 2\sqrt{\frac{s}{\gamma}} + \frac{\delta_{n,\varsigma}}{\lambda} \frac{1}{\|T(h-h_0)\|_2}$.

Moreover, observe that by the triangle inequality:

$$\|h_0\|_{\mathcal{H}} - \|\hat{h}\|_{\mathcal{H}} = \|\theta_0\|_1 - \|\hat{\theta}\|_1 \le \|\theta_0 - \hat{\theta}\|_1 = \|\nu\|_1 \le 2\sqrt{\frac{s}{\gamma}\nu^\top V \nu} + \delta_{n,\varsigma}/\lambda$$

Moreover, by standard results on the Rademacher complexity of linear function classes (see e.g. Lemma 26.11 of [Shalev-Shwartz and Ben-David, 2014]), we have $\mathcal{R}(\mathcal{H}_B) \le B\sqrt{\frac{2\log(2p)}{n}} \max_{x \in \mathcal{X}} \|x\|_\infty$ and $\mathcal{R}(\mathcal{F}_U) \le U\sqrt{\frac{2\log(2p)}{n}} \max_{z \in \mathcal{Z}} \|z\|_\infty$ for $\mathcal{F}_U = \{z \to \langle \beta, z \rangle : \beta \in \mathbb{R}^p, \|\beta\|_1 \le U\}$. Thus invoking Theorem 2:

$$\|T(\hat{h}-h_0)\|_2 \le \left(2\sqrt{\frac{s}{\gamma}} + \frac{\delta_{n,\varsigma}}{\lambda} \frac{1}{\|T(h-h_0)\|_2}\right) \cdot \left(2(B+1)\sqrt{\frac{\log(2p)}{n}} + \delta_{n,\varsigma} + \lambda\sqrt{\frac{s}{\gamma}}\|T(h-h_0)\|_2\right)$$

The right hand side is upper bounded by the sum of the following four terms:

$$Q_1 := 2\sqrt{\frac{s}{\gamma}}\left(2(B+1)\sqrt{\frac{\log(2p)}{n}} + \delta_{n,\varsigma}\right)$$

$$Q_2 := \left(\frac{\delta_{n,\varsigma}}{\lambda} \frac{1}{\|T(h-h_0)\|_2}\right)\left(2(B+1)\sqrt{\frac{\log(2p)}{n}} + \delta_{n,\varsigma}\right)$$

$$Q_3 := 2\lambda\frac{s}{\gamma}\|T(h-h_0)\|_2$$

$$Q_4 := \delta_{n,\varsigma}\sqrt{\frac{s}{\gamma}}$$

If $\|T(h-h_0)\|_2 \ge \sqrt{\frac{s}{\gamma}}\delta_{n,\varsigma}$ and setting $\lambda \le \frac{\gamma}{8s}$, yields:

$$Q_2 \le 8\frac{1}{\lambda}\sqrt{\frac{\gamma}{s}}\left(2(B+1)\sqrt{\frac{\log(2p)}{n}} + \delta_{n,\varsigma}\right)$$

$$Q_3 \le \frac{1}{4}\|T(h-h_0)\|_2$$

Thus bringing $Q_3$ on the left-hand-side and dividing by $3/4$, we have:

$$\|T(h-h_0)\|_2 \le \frac{4}{3}(Q_1 + Q_2 + Q_4) = \frac{4}{3}\max\left\{\sqrt{\frac{s}{\gamma}}, \frac{1}{\lambda}\sqrt{\frac{\gamma}{s}}\right\}\left(20(B+1)\sqrt{\frac{\log(2p)}{n}} + 11\delta_{n,\varsigma}\right)$$

The result for the case when $\sup_{z \in \mathcal{Z}} \|z\|_2 \le R$ and $\mathcal{F}_U = \{z \to \langle \beta, z \rangle : \|\beta\|_2 \le U\}$, follows along the exact same lines, but invoking the Lemma 26.10 of [Shalev-Shwartz and Ben-David, 2014], instead of Lemma 26.11, in order to get that $\mathcal{R}(\mathcal{F}_U) \le \frac{UR}{\sqrt{n}}$. $\qquad\square$

## M.2 Proof of Propositions 13 and 14

**Proposition 16.** *Consider an online linear optimization algorithm over a convex strategy space $S$ and consider the OFTRL algorithm with a 1-strongly convex regularizer with respect to some norm $\|\cdot\|$ on space $S$:*

$$f_t = \underset{f \in S}{\arg\min} f^\top \left(\sum_{\tau \le t} \ell_\tau + \ell_t\right) + \frac{1}{\eta}R(f)$$

*Let $\|\cdot\|_*$ denote the dual norm of $\|\cdot\|$ and $R = \sup_{f \in S} R(f) - \inf_{f \in S} R(f)$. Then for any $f^* \in S$:*

$$\sum_{t=1}^{T} (f_t - f^*)^\top \ell_t \leq \frac{R}{\eta} + \eta \sum_{t=1}^{T} \|\ell_t - \ell_{t-1}\|_*^2 - \frac{1}{4\eta} \sum_{t=1}^{T} \|f_t - f_{t-1}\|^2$$

*Proof.* The proof follows by observing that Proposition 7 in Syrgkanis et al. [2015] holds verbatim for any convex strategy space $S$ and not necessarily the simplex. $\qquad\square$

**Proposition 17.** *Consider a minimax objective: $\min_{\theta \in \Theta} \max_{w \in W} \ell(\theta, w)$. Suppose that $\Theta, W$ are convex sets and that $\ell(\theta, w)$ is convex in $\theta$ for every $w$ and concave in $\theta$ for any $w$. Let $\|\cdot\|_\Theta$ and $\|\cdot\|_W$ be arbitrary norms in the corresponding spaces. Moreover, suppose that the following Lipschitzness properties are satisfied:*

$$\forall \theta \in \Theta, w, w' \in W : \|\nabla_\theta \ell(\theta, w) - \nabla_\theta \ell(\theta, w')\|_{\Theta, *} \leq L \|w - w'\|_W$$

$$\forall w \in W, \theta, \theta' \in \Theta : \|\nabla_w \ell(\theta, w) - \nabla_w \ell(\theta', w)\|_{W, *} \leq L \|\theta - \theta'\|_W$$

*where $\|\cdot\|_{\Theta, *}$ and $\|\cdot\|_{W, *}$ correspond to the dual norms of $\|\cdot\|_\Theta, \|\cdot\|_W$. Consider the algorithm where at each iteration each player updates their strategy based on:*

$$\theta_{t+1} = \underset{\theta \in \Theta}{\arg\min} \, \theta^\top \left( \sum_{\tau \leq t} \nabla_\theta \ell(\theta_\tau, w_\tau) + \nabla_\theta \ell(\theta_t, w_t) \right) + \frac{1}{\eta} R_{\min}(\theta)$$

$$w_{t+1} = \underset{w \in W}{\arg\max} \, w^T \left( \sum_{\tau \leq t} \nabla_w \ell(\theta_\tau, w_\tau) + \nabla_w \ell(\theta_t, w_t) \right) - \frac{1}{\eta} R_{\max}(w)$$

*such that $R_{\min}$ is 1-strongly convex in the set $\Theta$ with respect to norm $\|\cdot\|_\Theta$ and $R_{\max}$ is 1-strongly convex in the set $W$ with respect to norm $\|\cdot\|_W$ and with any step-size $\eta \leq \frac{1}{4L}$. Then the parameters $\bar{\theta} = \frac{1}{T} \sum_{t=1}^{T} \theta_t$ and $\bar{w} = \frac{1}{T} \sum_{t=1}^{T} w_t$ correspond to an $\frac{2R_*}{\eta \cdot T}$-approximate equilibrium and hence $\bar{\theta}$ is a $\frac{4R_*}{\eta T}$-approximate solution to the minimax objective, where $R$ is defined as:*

$$R_* := \max \left\{ \sup_{\theta \in \Theta} R_{\min}(\theta) - \inf_{\theta \in \Theta} R_{\min}(\theta), \, \sup_{w \in W} R_{\max}(w) - \inf_{w \in W} R_{\max}(w) \right\}$$

*Proof.* The proposition is essentially a re-statement of Theorem 25 of Syrgkanis et al. [2015] (which in turn is an adaptation of Lemma 4 of Rakhlin and Sridharan [2013]), specialized to the case of the OFTRL algorithm and to the case of a two-player convex-concave zero-sum game, which implies that the if the sum of regrets of players is at most $\epsilon$, then the pair of average solutions corresponds to an $\epsilon$-equilibrium (see e.g. Freund and Schapire [1999] and Lemma 4 of Rakhlin and Sridharan [2013]). $\qquad\square$

**Proof of Proposition 13: $\ell_1$-ball adversary** Let $R_E(x) = \sum_{i=1}^{2p} x_i \log(x_i)$. For the space $\Theta := \{\rho \in \mathbb{R}^{2p} : \rho \geq 0, \|\rho\|_1 \leq B\}$, the entropic regularizer is $\frac{1}{B}$-strongly convex with respect to the $\ell_1$ norm and hence we can set $R_{\min}(\rho) = B \, R_E(\rho)$. Similarly, for the space $W := \{w \in \mathbb{R}^{2p} : w \geq 0, \|w\|_1 = 1\}$, the entropic regularizer is 1-strongly convex with respect to the $\ell_1$ norm and thus we can set $R_{\max}(w) = R_E(w)$. For this choice of regularizers, the update rules can be easily verified to have a closed form solution provided in Proposition 13, by writing the Lagrangian of each OFTRL optimization problem and invoking strong duality. Further, we can verify the lipschitzness conditions. Since the dual of the $\ell_1$ norm is the $\ell_\infty$ norm, $\nabla_\rho \ell(\rho, w) = \mathbb{E}_n[vu^\top]w + \frac{\mu}{W}$ and thus:

$$\|\nabla_\rho \ell(\rho, w) - \nabla_\rho \ell(\rho, w')\|_\infty = \|\mathbb{E}_n[vu^\top](w - w')\|_\infty \leq \|\mathbb{E}_n[vu^\top]\|_\infty \|w - w'\|_1$$

$$\|\nabla_w \ell(\rho, w) - \nabla_w \ell(\rho', w)\|_\infty = \|\mathbb{E}_n[uv^\top](\rho - \rho')\|_\infty \leq \|\mathbb{E}_n[vu^\top]\|_\infty \|\rho - \rho'\|_1$$

Thus we have $L = \|\mathbb{E}_n[uv^\top]\|_\infty$. Finally, observe that:

$$\sup_{\rho \in \Theta} B \, R_E(\rho) - \inf_{\rho \in \Theta} B \, R_E(\rho) = B^2 \log(B \vee 1) + B \log(2p)$$

$$\sup_{w \in W} R_E(w) - \inf_{w \in W} R_E(w) = \log(2p)$$

Thus we can take $R_* = B^2 \log(B \vee 1) + (B+1) \log(2p)$. Thus if we set $\eta = \frac{1}{4\|\mathbb{E}_n[vu^\top]\|_\infty}$, then we have that after $T$ iterations, $\bar\theta = \bar\rho^+ - \bar\rho^-$ is an $\epsilon(T)$-approximate solution to the minimax problem, with

$$\epsilon(T) = 16\|\mathbb{E}_n[vu^\top]\|_\infty \frac{4B^2 \log(B \vee 1) + (B+1) \log(2p)}{T}.$$

Combining all the above with Proposition 17 yields the proof of Proposition 13.

**Proof of Proposition 14: $\ell_2$-ball adversary** For the case when $W := \{\beta \in \mathbb{R}^p : \|\beta\|_2 \leq U\}$, then we have that the squared norm regularizer $R_{\max}(\beta) = \frac{1}{2}\|\beta\|_2^2$ is 1-strongly convex with respect to the $\ell_2$ norm and we can use $\|\cdot\|_W = \|\cdot\|_2$. The choice of $R_{\min}$ is the same as in the case of an $\ell_1$ adversary, as detailed in the previous paragraph. For this choice of regularizers, the update rules can be easily verified to have a closed form solution provided in Proposition 14, by writing the Lagrangian of each OFTRL optimization problem and invoking strong duality. Moreover, the Lipschitzness conditions become:

$$\|\nabla_\rho \ell(\rho, \beta) - \nabla_\rho \ell(\rho, \beta')\|_\infty = \|\mathbb{E}_n[vz^\top](\beta - \beta')\|_\infty \leq \|\mathbb{E}_n[vz^\top]\|_{\infty,2}\|\beta - \beta'\|_2$$
$$\|\nabla_\beta \ell(\rho, \beta) - \nabla_\beta \ell(\rho', \beta)\|_2 = \|\mathbb{E}_n[zv^\top](\rho - \rho')\|_2 \leq \|\mathbb{E}_n[zv^\top]\|_{2,\infty}\|\rho - \rho'\|_1$$

where $\|A\|_{\infty,2} = \max_i \sqrt{\sum_j A_{ij}^2}$ and $\|A\|_{2,\infty} = \sqrt{\sum_i \max_j A_{ij}^2}$. Thus we can take

$$L = \max \left\{ \max_i \sqrt{\sum_j \mathbb{E}_n[v_i z_j]^2} + \sqrt{\sum_i \max_j \mathbb{E}_n[z_i v_j]^2} \right\}$$
$$\leq \sqrt{\sum_i \max_j \mathbb{E}_n[z_i v_j]^2} = \|\mathbb{E}_n[zv^T]\|_{2,\infty}$$

Finally, we also have that:

$$\sup_{\beta \in W} R_{\max}(\beta) - \inf_{\beta \in W} R_{\max}(\beta) \leq \frac{1}{2}U^2$$

Thus we can take $R_* = B^2 \log(B \vee 1) + B \log(2p) + \frac{1}{2}U^2$. Thus if we set $\eta = \frac{1}{4\|\mathbb{E}_n[zv^\top]\|_{2,\infty}}$, then we have that after $T$ iterations, $\bar\theta = \bar\rho^+ - \bar\rho^-$ is an $\epsilon(T)$-approximate solution to the minimax problem, with

$$\epsilon(T) = 16\|\mathbb{E}_n[zv^\top]\|_{2,\infty} \frac{4B^2 \log(B \vee 1) + B \log(2p) + U^2/2}{T}.$$

Combining all the above with Proposition 17 yields the proof of Proposition 14.

# N  Proofs from Section 7 and Appendix I

## N.1  Proof of Theorem 4

Observe that we can view the minimax problem as the solution to a convex-concave zero-sum game, where the strategy of each player is a vector in an $n$-dimensional space, subject to complex constraints imposed by the corresponding hypothesis. In particular, let $A = \{(f(z_1), \ldots, f(z_n)) : f \in \mathcal{F}\}$ and $B = \{(h(x_1), \ldots, h(z_n)) : h \in \mathcal{H}\}$. Then the minimax problem can be phrased as:

$$\min_{b \in B} \max_{a \in A} \frac{1}{n} \sum_i ((y_i - b_i) a_i - a_i^2) = \max_{b \in B} \min_{a \in A} \frac{1}{n} \sum_i (a_i^2 - (y_i - b_i) a_i)$$

Moreover, we will denote with $\ell(a, b) := \frac{1}{n}\sum_i (a_i^2 - (y_i - b_i) a_i)$, which is a loss that is concave (in fact linear) in $b$ and convex in $a$. Moreover, our assumption on $\mathcal{F}$ implies that $A$ is a convex set.

Then the algorithm described in the statement of the theorem corresponds to solving this zero-sum game via the following iterative algorithm: at every period $t = 1, \ldots, T$, the adversary chooses a vector $a_t$ based on the the follow the leader (FTL) algorithm, i.e.:

$$a_t = \arg\min_{a \in A} \frac{1}{t-1} \sum_{\tau=1}^{t-1} \ell(a, b_\tau)$$

and the learner chooses $b_t$ by best-responding to the current test function, i.e.:

$$b_t = \arg\max_{b \in B} \ell(a_t, b)$$

The equivalent stems from the following two observations: First, for the adversary we can re-write the FTL algorithm by completing the square as:

$$
\begin{aligned}
a_t &= \arg\min_{a \in A} \frac{1}{n} \sum_i \frac{1}{t-1} \sum_{\tau=1}^{t-1} (a_i^2 - (y_i - b_{it})\, a_i) \\
&= \arg\min_{a \in A} \frac{1}{n} \sum_i \left( a_i^2 - \left( y_i - \frac{1}{t-1} \sum_{\tau=1}^{t-1} b_{it} \right) a_i \right) \\
&= \arg\min_{a \in A} \frac{1}{n} \sum_i \left( a_i^2 - \frac{1}{2} \left( y_i - \frac{1}{t-1} \sum_{\tau=1}^{t-1} b_{it} \right) \right)^2
\end{aligned}
$$

which then is equivalent to the oracle call described in the statement of the theorem. Second for the learner we have:

$$
\begin{aligned}
b_t &= \arg\max_{b \in B} \ell(a_t, b) \\
&= \arg\max_{b \in B} \frac{1}{n} \sum_i b_i a_{it} \\
&= \arg\max_{b \in B} \frac{1}{n} \sum_i b_i |a_{it}| \mathtt{sign}(a_{it}) \\
&= \arg\max_{b \in B} \frac{1}{n} \sum_i |a_{it}| \mathbb{E}_{z \sim \mathrm{Bernoulli}(\frac{b_i+1}{2})}[(2\,z_i - 1)\, \mathtt{sign}(a_{it})] \\
&= \arg\max_{b \in B} \frac{1}{n} \sum_i |a_{it}| \left( \Pr_{z \sim \mathrm{Bernoulli}(\frac{b_i+1}{2})}[(2\,z_i - 1) = \mathtt{sign}(a_{it})] - \Pr_{z \sim \mathrm{Bernoulli}(\frac{b_i+1}{2})}[(2\,z_i - 1) \neq \mathtt{sign}(a_{it})] \right) \\
&= \arg\max_{b \in B} \frac{1}{n} \sum_i |a_{it}| \left( 2\Pr_{z \sim \mathrm{Bernoulli}(\frac{b_i+1}{2})}[(2\,z_i - 1) = \mathtt{sign}(a_{it})] - 1 \right) \\
&= \arg\max_{b \in B} \frac{1}{n} \sum_i |a_{it}| \Pr_{z \sim \mathrm{Bernoulli}(\frac{b_i+1}{2})}[(2\,z_i - 1) = \mathtt{sign}(a_{it})] \\
&= \arg\max_{b \in B} \frac{1}{n} \sum_i |a_{it}| \Pr_{z \sim \mathrm{Bernoulli}(\frac{b_i+1}{2})}\left[ z_i = \frac{\mathtt{sign}(a_{it}) + 1}{2} \right] \\
&= \arg\max_{b \in B} \frac{1}{n} \sum_i |a_{it}| \Pr_{z \sim \mathrm{Bernoulli}(\frac{b_i+1}{2})}[z_i = 1\{a_{it} > 0\}]
\end{aligned}
$$

which is exactly the oracle call described in the statement of the theorem.

Thus it remains to show that the vector $\bar{b} = \frac{1}{T} \sum_{t=1}^T b_t$ is a solution to the minimax problem, which would imply that the corresponding ensemble hypothesis $\bar{h} = \frac{1}{T} \sum_{t=1}^T h_t$ is also a solution to the empirical minimax problem.

To achieve this it suffices to show that the FTL algorithm is a no-regret algorithm for the adversary. Then we can invoke classic results on solving zero-sum games via no-regret dynamics [Freund and Schapire, 1999]. Observe that the learner obviously has zero regret as it best-responds at each period. Thus if we show that the FTL algorithm has $\epsilon(T)$-regret after $T$ periods, then $\bar{b}$ is an $\epsilon(T)$-approximate solution to the minimax problem, invoking the results of [Freund and Schapire, 1999].

Hence, we now focus on the online learning problem that the adversary is facing and show that FTL is a no-regret algorithm with regret rate $\epsilon(T) = \frac{4\log(T)}{T}$. We will begin by invoking Lemma 2.1 of [Shalev-Shwartz and Singer, 2007], which states that the regret of the FTL algorithm is bounded by:

$$\epsilon(T) \leq \frac{1}{T} \sum_{t=1}^T (\ell(a_t, b_t) - \ell(a_{t+1}, b_t))$$

Thus it remains to bound the RHS.

Observe that the loss function $\ell(\cdot, b)$ is $\frac{2}{n}$-strongly convex with respect the $\|\cdot\|_2$ norm on the space $A$, since $a^\top \nabla^2_{aa}\ell(a,b)a = \frac{2}{n}\|a\|^2_2$. Moreover, observe that the loss function $\ell(\cdot, b)$ is also $\frac{4}{\sqrt{n}}$-Lispchitz with respect to the $\|\cdot\|_2$ norm on the space $A$, since

$$\nabla_{a_i}\ell(a,b) = \frac{1}{n}(2\,a_i - (y_i - b_i))$$

and therefore:

$$\|\nabla_a \ell(a,b)\|_2 = \sqrt{\frac{1}{n^2}\sum_i (y_i - b_i - 2\,a_i)^2} = \frac{1}{\sqrt{n}}\sqrt{\frac{1}{n}\sum_i (y_i - b_i - 2\,a_i)^2} \leq \frac{4}{\sqrt{n}}$$

In the last inequality we used the fact $|y_i|, |h(x_i)|, |f(z_i)| \leq 1$.

Since $\ell_t$ is $\frac{2}{n}$-strongly convex, we have that $L_t = \sum_{\tau=1}^t \ell(\cdot, b_\tau)$ is $\frac{2t}{n}$ strongly convex. Since $a_{t+1}$ is the minimizer of $L_t$ and the set $A$ is a convex set, we have by strong convexity and the first order condition that:

$$L_t(a_t) \geq L_t(a_{t+1}) + \langle a_t - a_{t+1}, \nabla_a L_t(a_{t+1})\rangle + \frac{t}{n}\|a_t - a_{t+1}\|^2_2 \geq L_t(a_{t+1}) + \frac{t}{n}\|a_t - a_{t+1}\|^2_2$$

Moreover, since $a_t$ is a minimizer of $L_{t-1}$ and invoking the first order condition, in a similar way as above, we have:

$$L_{t-1}(a_{t+1}) \geq L_{t-1}(a_t) + \frac{t}{n}\|a_t - a_{t+1}\|^2_2$$

Adding the two inequalities and re-arranging we get:

$$\ell(a_t, b_t) - \ell(a_{t+1}, b_t) \geq \frac{2t}{n}\|a_t - a_{t+1}\|^2_2$$

Invoking the lipschitzness of $\ell_t$:

$$\frac{4}{\sqrt{n}}\|a_t - a_{t+1}\|_2 \geq \ell(a_t, b_t) - \ell(a_{t+1}, b_t) \geq \frac{2t}{n}\|a_t - a_{t+1}\|^2_2$$

Thus we have:

$$\|a_t - a_{t+1}\|_2 \leq \frac{2\sqrt{n}}{t}$$

Moreover, by lipschitzness of $\ell(\cdot, b)$, we have:

$$\ell(a_t, b_t) - \ell(a_{t+1}, b_t) \leq \frac{4}{\sqrt{n}}\|a_t - a_{t+1}\|_2 \leq \frac{8}{t}$$

Thus we get:

$$\epsilon(T) \leq \frac{8}{T}\sum_{t=1}^T \frac{1}{t} \leq \frac{8(\log(T)+1)}{T}$$

## Footnotes

[3] The $\log(n)$ factor can also be saved with a more careful analysis of the critical radius for finite dimensional linear function spaces (see Section D).

[4] Observe that: $K(x, x) = \sum_{j=1}^{\infty} \lambda_j e_j(x)^2$ and therefore: $\sum_{j=1}^{\infty} \lambda_j = \sum_{j=1}^{\infty} \lambda_j \mathbb{E}_x[e_j(x)^2] = \mathbb{E}_x[\sum_{j=1}^{\infty} \lambda_j e_j(x)^2] = \mathbb{E}_x[K(x, x)] \leq \max_{x \in \mathcal{X}} K(x, x)$. Thus in the worst case, when $\lambda_j \geq \delta^2$ for most $j$, we still recover the non-localized from the localized bounds.

[5]Several sampling strategies have been proposed in the literature to improve upon pure uniform sampling (see e.g. Kumar et al. [2012], Musco and Musco [2017], Oglic and Gärtner [2017]). One popular practical and simple method is to perform some version of unsupervised clustering of the samples, such as kmeans clustering, and choosing the points as the cluster centroids.

[6]We note that the proof of Theorem 1 implies that even without a hard constraint, with high probability $\|\hat{h}\|_K^2 \leq \|h_0\|_K^2 + \frac{\delta^2 + \lambda U}{\mu}$. Thus the results of this section hold for $B = \|h_0\|_K^2 + \frac{\delta^2 + \lambda U}{\mu}$ even without the extra hard constraint.

[7]Potentially the strongest assumption of these is that $\gamma_m \leq \tau_m$. This could be avoided by restricting the hypothesis space $\mathcal{H}_B$ to only be supported on the first $m$ eigenfunctions. However, this would require being able to diagonalize the kernel and also to tune the estimator to the unknown parameters $\tau_m$.

[8]For the case of the $\ell_1$ norm: $\beta_t = U e_{i_t}\texttt{sign}(\mathbb{E}_n[y-\langle\theta_t,x\rangle)z_{i_t}])$, with $i_t = \arg\max_i|\mathbb{E}_n[(y-\langle\theta_t,x\rangle)z_i]|$. For the case of the $\ell_2$ norm: $\beta_t = \mathbb{E}_n[(y-\langle\theta_t,x\rangle)z]\cdot U/\|\mathbb{E}_n[(y-\langle\theta_t,x\rangle)z]\|_2$

[9]Finally, if we want to compare with $s$-sparse solutions and we want to enhance sparsity of the returned solution, then we can always truncate to zero at the end of training any coordinate of $\bar\theta = \bar\rho^+ - \bar\rho^-$ that was smaller than $1/(s\,n^{1/2+\epsilon})$. This can introduce an extra lower order approximation error of at most $1/n^{1/2+\epsilon}$ in our projected MSE theorem, since by this shrinkage procedure, the error with respect to a sparse solution $\theta_0$ can only increase on the non-zero entries of $\theta_0$ and it can only increase by at most $1/(sn^{1/2+\epsilon})$ on every such entry.

[10]In particular, $\bar\rho$ and $\bar\beta$ are an $\epsilon$-equilibrium of the zero-sum game.

[11]We note that the fast rate of $1/\epsilon$ will deteriorate with the size of the mini-batch, but a $1/\epsilon^2$ rate is always achievable and the step-size $\eta$ should be appropriately tuned to account for the mini-batch sampling noise.

[12]For a matrix $A$, we denote with $\|A\|_\infty = \max_{i,j}|A_{ij}|$

[13]If $p \ge n$, then at every iteration we can calculate $m^{(j)} = v^{(j)} \cdot w_t$, for each sample $v^{(j)}$; which takes $O(n \cdot p)$ time; and then update each $\tilde\rho_{i,t+1}$ based on the quantity $\mathbb{E}_n[v_i u^\top w_t] = \frac{1}{n}\sum_j v_i^{(j)} m^{(j)}$. If $p < n$, then we can calculate $\Sigma_n = \mathbb{E}_n[vu^\top]$ ahead of time and at each period calculate $\mathbb{E}_n[v_i u^\top w_t] = (\Sigma w_t)_i$; which would require $O(p^2)$ time.

[14]For a matrix $A$, we denote with $\|A\|_{2,\infty} = \sqrt{\sum_i \max_j A_{ij}^2}$

[15]Our results easily extend to arbitrary intervals $x \in [a, b]$ and ranges $[-H, H]$, though we restrict to $[0, 1]$ for simplicity of exposition.

[16]If we want to enforce a monotone non-decreasing $h$, then we can set $\theta^- = 0$ and similarly, for a monotone non-increasing algorithm $\theta^+ = 0$.

[17] By setting $\lambda = \delta^2/U$, $\mu = 2\lambda\left(4L^2 + 27U/B\right)$ using an $\ell_\infty$ norm in both function spaces and taking $U, B \to \infty$. Observe that we can also take $L = 1$, since $\|Th\|_\infty \leq \|h\|_\infty$ for any $T$.

[18]The fact that for any matrix $X$: $X(X^\top X + \lambda I)^{-1} = XX^\top (X^\top X + \lambda I)$, and that $K_n = K_n^{1/2} K_n^{1/2}$ and $K_n^{1/2}$ is symmetric.