[Reviews · NeurIPS 2020]

Review 1

Summary and Contributions: The paper studies nonparametric instrumental variable regression through generalized method of moments. It takes the model that y = h0(x) + epsilon where E[epsilon | z] = 0 for outcome y, treatment x, instrument z, and true causal effect h0. It proposes a new estimator for the nonparametric causal effect of treatment x on outcome y based on minimizing the largest correlation between y - h(x) and a function of the instrument variable z. It analyzes the sample complexity of this estimator in terms of the complexity of model classes of functions of x and of z.

Strengths: The main contribution of the paper is theoretical and the results appear sensible. I have not verified the correctness of the proofs. The sample complexity results are based on well-established existing techniques but it is nontrivial to deploy them in this new estimation setting. The results are quite general. There is a significant and growing interest in understanding how machine learning can be applied to causal inference. This is an important area and this paper makes a theoretical contribution towards it.

Weaknesses: 1. The paper does not assume completeness, that is, that E[ g(x) | z ] = 0 if and only if g = 0 and hence, h0 is not identifiable. In this case, I am not sure whether IV regression still makes sense since the causal effect cannot be identified. 2. The conditional moment restriction E[ y - h(x) | z ] = 0 holds if and only if E[ (y-h(x)) f(x) ] = 0 for all L1 integrable function f. The paper instead considers a weaker criterion where E[ (y - h(x)) f(x) ] = 0 for all f in a smaller function class F. This may excerbate the identifiability issue that I described in the first point. One can address this by using a smaller function class for h in the optimization. This interplay between function class H and F seems important for determining which model class to use and I think this is not addressed by the paper. 3. I have trouble digesting the conditions and assumptions used in the theory. For instance, there is an assumption in Theorem 1 involving eta_n (line 148) that is not really discussed. I am not sure how reasonable this is. 4. It does not seem easy to choose the tuning parameters in this estimationg framework. I would guess that any cross-validation procedure would have to take the supremum over f so it is not clear how the parameter for the penalty of f can be chosen. 5. There seems to be no way to perform statistical tests on whether h0 = 0.

Correctness: The results look reasonably but I cannot vouch for their correctness.

Clarity: The paper explains the mathematical results quite clearly but does not sufficiently explain the motivation behind the optimization.

Relation to Prior Work: Although the paper cites many prior work, I do not have a very clear idea on where and how this paper fits into existing literature.

Reproducibility: Yes

Additional Feedback: The title of the paper is very misleading. The word "minimax estimation" generally refers to minimax optimality of an estimation procedure. I would recommend the phrase "min-max estimation" or "saddlepoint estimation". Typo in Theorem 2: should be R_n instead of R. Edit: author feedback is satisfactory.


Review 2

Summary and Contributions: The authors feedback successfully answered my concerns about the experimental section of the paper, and the revisions and additional experiments that they propose to add should be easily carried out. For that reason, I plan to raise my evaluation. ---------------------------------------------------------------------------------------- This paper describes how to estimate models that can be described as conditional moment restrictions with a min-max criterion function, and analyzes the estimation rate. Given proper regularization the estimation rate scales with the critical radius of the hypothesis space. The analysis is applied to a number of important hypothesis spaces, including RKHS, random forests, neural networks, etc. They also provide experimental results.

Strengths: The problem (instrumental variable estimation) considered is an extremely important one with many potential applications. The authors have proposed a variety of new estimators, and derived theorems about the rates of convergence for them.

Weaknesses: The paper lists a number of papers on the topic of non-parametric estimation of instrumental variable models. However, it appears that an estimator from only one of these papers is used in the experimental section as a comparison of the estimators proposed in this paper. In the simulated data in Figure 2, the new estimators are tested only against 2SLS and REG2SLS. But they don't test the new version of a neural network against existing neural networks, or the new version of random forests against existing versions of random forests, and similarly for RKHS, etc. There are estimators of these kinds for instrumental variable models. They would seem to provide a fairer test of what has been contributed by this paper, that is the objective function, and not just whether neural networks etc. are better than 2SLS. Also, the algorithms in Figure 3 aren't tested against any existing algorithms. The algorithms in Figure 4 is tested against one existing algorithm, but nothing is said about why that one was selected.

Correctness: I did not check the correctness of the claims. I had some comments about the experimental section discussed in the weaknesses section.

Clarity: This paper is quite difficult to read, although that may be due to the fact that they are trying to cover a tremendous amount of material, and the fact that while I am familiar with the basics of the background knowledge needed to understand the paper, I am not an expert in some of those areas. In terms of reproducibility, the main body of the paper does not contain enough details to be reproducible, although many more details are provided in the supplementary material. The abstract makes some claims that are not explicitly mentioned later in the paper, e.g. that the estimation problem can be thought of as a zero-sum game between a modeler and an adversary. The article repeatedly refers to Theorem 6, but there is no Theorem 6 in the paper (just the supplementary material). The expression of instrumental variable models as a conditional moment restriction in equation (1), although used in econometrics is very different than the usual expression as conditional independence or graphical constraints on the model usually used in computer science and other fields. It would be helpful if the relationship between those two expressions were clarified - does equation (1) assume additive noise for example? Estimating h from equation (1) is generally going to be ill-posed without the addition of further constraints. I gather (partially from consulting some of the articles that are cited in the paper) that the further constraints are the regularization terms that were added in equation 2. Is that correct? If so, this should be made more explicit in the paper. (There is a place in the body of the paper that mentions ill-posedness, but with respect to the correlation between Z and X). The regularization terms that were introduced in equation 2, are not really given an intuitive description of what they represent, and aren't instantiated until Theorem 1. Are they generally a penalty for complex hypotheses? Are there some properties that the regularization terms have to satisfy in general? The variable U is introduced in Theorem 1, but no intuitive explanation of what role it is playing is given.

Relation to Prior Work: The authors are clear about what contributions they have made.

Reproducibility: Yes

Additional Feedback:


Review 3

Summary and Contributions: This paper proposes an approach for estimating models characterized by conditional moment restrictions. The proposed approach minimizes a loss using an adversarial test function, leading to a minimax optimization problem. The authors prove several forms of theoretical guarantees depending on the conditions that the function classes satisfy and show interesting consequences including the cases for reproducing kernel Hilbert spaces and sparse linear functions. The paper also presents interesting algorithms within the proposed approach for neural networks and random forests. Their synthetic experiments demonstrate the effectiveness of the proposed approach using different types of function classes.

Strengths: The paper is excellently written and very easy to follow despite its formality and density. They provide rigorous analysis on the error rate of their estimator in a general form to show that the error is bounded by the critical radii of the function classes. This suggests that the proposed approach is promising for many function classes with small local Rademacher complexities. Indeed, the paper presents several instances for more concrete function classes with detailed analyses indicating their theoretically good performance. In particular, the fast rate for the sparse model is an appealing property. The authors also provide algorithms and heuristics for practically popular models such as neural networks and random forests.

Weaknesses: The proposed approach has a loss function that depends on the class of test functions. This means that one cannot simply compare different losses for different function classes, which makes it difficult to select the function class or tune hyper-parameters based on loss scores with validation data. The paper does not seem to present experiments with real-world data.

Correctness: I do not find any critical flaws although I did not have time to read the whole proofs of the theorems.

Clarity: The paper is well-written and nicely organized, and I enjoyed reading it. I only had a quick look at the supplementary material, but it also looks great.

Relation to Prior Work: I could not find any discussion about the different approach based on the Neyman orthogonality such as "Machine Learning Estimation of Heterogeneous Treatment Effects with Instruments" by Syrgkanis et al. (2019). The authors should discuss the difference between their work and "Adversarial Generalized Method of Moments" by Lewis and Syrgkanis (2018) since it takes a very similar approach. "Uplift Modeling from Separete Labels" by Yamane et al. (2018) may be also related.

Reproducibility: Yes

Additional Feedback: - In Theorem 1, is \eta_n supposed to depend on the sample size n and approach zero as n tends to infinity? - Where is the definition for L in Line 151? - Is there any practical way to tune hyper-parameters (e.g., regularization parameters and kernels)? - Line 176, 178: Is R the same as R_n? - The last term of the bound below Line 178 is a negative quantity, which feels a little strange to me. Is its absolute value guaranteed to be small enough not to dominate the bound? ===== After the authors' feedback ===== I have read the other reviews and the authors' response, and most of my concerns have been clearly addressed. I believe the paper makes great contribution to the methodology and the theory of this important estimation problem, and I'd like to keep my score to 8. I encourage the authors to add more intuitive explanations and interpretations for the theory part and provide experimental results on real-world data.

[Author Response · NeurIPS 2020]

We thank all reviewers for their careful reading, insightful comments and feedback. We apologize for typos and the lack
of clarity and heavy notation in some places. We will take all comments into account and fix typos and improve clarity.

**R1:** Our convergence is robust even in the absence of completeness or point identifiability (PI). Having PI allows us to
argue stronger convergence. Hence our results are simply stronger than assuming PI from the get-go as is typically
done. Our bounds are applicable to weak instruments where PI exists in the limit but is brittle with finite $n$. Moreover
(as noted in prior work) projected RMSE convergence suffices to estimate a nuisance parameter in a semiparametric
model and making good predictions does not require parameter identification. We could assume completeness and
together with a bound on ill-posedness (Apx C.4) our bounds imply identifiability. We will add a Corollary to this affect.
Note that we present a general criterion (lines 62-65) for test function selection to ensure equivalence of our objective
to $E[y - h(x)|z] = 0$. We also give many general function classes in our paper where this criterion yields a natural
function class. $\eta_n$ in line 148 appears in our bound on RMSE in line 152. It measures how good of an approximation
our norm-constrained test function family is to the optimal test function family by measuring the maximum error of the
best function in our norm-constrained family over all hypotheses in the space $\mathcal{H}$. We provide theoretical lower bounds
on the regularization hyper-parameters which can be used to select them. In addition to perform cross-validation for
hyper-parameter selection we have found that we can approximate the supremum over $\mathcal{F}$ by the supremum over the set
of functions $f$ encountered in a pre-training phase where we store the set of test functions $f$ we evaluate against at each
point in the pre-training. We can use this set later on for cross-validation to effectively simulate a supremum over $\mathcal{F}$.
Lastly, we do not address the question of hypothesis testing in this work ,but primarily address estimation. Testing and
estimation are two orthogonal tasks that are typically complementary but are also orthogonal and each of own interest.

**R2:** We apologize for the terse presentation of our experimental results in the main body which appears to have
caused an impression that we are not comparing to the recent works in IV regression. We will present a clearer, more
comprehensive experimental comparison in future versions. First, we would like to point that we do contain a more
comprehensive presentation of our experimental results in the Apx. For non-parametric IV there is no prior Random
Forest (RF) algorithm, as we outline in the RF section. We present the first RF algorithm for this setting. Prior RF
algorithms for IV setup only work when one makes the assumption of linearity w.r.t. to treatment and estimates
heterogeneity with respect to exogenous features (such as the IV forest of Athey and Wager). In Fig 2, for neural nets
we compare with AGMM which is reported to outperform DeepIV (hence we exclude DeepIV in the table). Fig 3, deals
with a sparse linear setting where the dimension of the input can be much larger than the number of available samples.
Many of the prior works cited do not have an explicit focus on handling sparsity and without such a focus would not
scale well in the high-dimensional setting. That being said, for the sake of completeness, we will add a comparison of
their performance with our approach in the Apx. In Fig 4, we compare with DeepGMM for two reasons. Many of the
other works implementations do not scale computationally to such high-dimensional instrument and treatment spaces.
Primarily neural-net based approaches scale well. DeepIV is one previous approach which works when the instrument
space is high-dimensional but since the DeepGMM paper reported a better performance of their estimator compared to
DeepIV in this setting, and since we outperform DeepGMM in this setting we left out a comparison with DeepIV in our
table. We apologize for the lack of clarity in some places. We will fix all of them in the paper. We will 1) make explicit
the zero-sum aspect of our min-max formulation, 2) fix references to Theorems in Apx, and 3) clarify the dependence
structure for IV regression. We do need bounds on ill-posedness for good RMSE and we provide explicit bounds for
some of the function classes we consider. However these constraints are different from the regularization terms of eq
(2). The regularization terms represent bounds on the norms of the function classes (and consequently their complexity)
and are necessary to get convergence. We will make this more clear. $U$ in Theorem 1 denotes a bound on the norm
of functions in $\mathcal{F}_U$ which is essential for controlling the complexity of the class of test functions and thereby getting
convergence in RMSE.

**R3:** Selecting test function family is indeed important for our approach. We provide strong theoretical guidance to
do this depending on the richness of our hypothesis class (for e.g. lines 62-65). Once we have selected a class of test
functions, we show how hyper-parameters for regularized estimators can be picked in many instantiations. Experiments
on real-world data is indeed an important direction and we leave it for future work. We do demonstrate the robustness
of our approach to partial real-world data by showing its efficacy on data comprised of MNIST images. Prior work: The
work on ML estimation for hetero effects assumes that the function $h(T, X)$ is linear in the endogenous treatment $T$ and
only heterogeneous wrt to the exogenous variables $X$. The linearity is the main assumption that enables the results of
that work and makes a significant qualitative and technical difference. The unpublished arXiv work of AGMM does not
provide statistical guarantees of the resulting estimator apart from a fully non-parametric rate that grows exponentially
with dimension. A crucial difference is they don't penalize the objective with the norm of the test function which is the
key idea that enables our fast rates (based on critical radius of $\mathcal{F}$). Finally, AGMM only provides experimental results
for neural nets, while here we provide experimental and theoretical results for many other function classes of interest.

[Meta-Review · NeurIPS 2020]

All the reviewers are in favor of accepting the paper. The rebuttal has answered all the questions. I would suggest revising the paper to address the questions and clarification that reviewers had.